


Earth **Surface** Dynamics
Discussions

# Hybrid modeling on 3D hydraulic features of a step-pool unit

Chendi Zhang[1], Yuncheng Xu[1,2], Marwan A Hassan[3], Mengzhen Xu[1], Pukang He[1]

[1]State Key Laboratory of Hydroscience and Engineering, Tsinghua University, Beijing, 100084, China.
[2]College of Water Resources and Civil Engineering, China Agricultural University, Beijing, 100081, China.
[3]Department of Geography, University of British Columbia, 1984 West Mall, Vancouver BC, V6T1Z2, Canada.

*Correspondence to*: Chendi Zhang (chendinorthwest@163.com) and Mengzhen Xu (mzxu@mail.tsinghua.edu.cn)

**Abstract.** Step-pool systems are common bedforms in mountain streams and have been utilized in river restoration projects around the world. Step-pool units exhibit highly non-uniform hydraulic characteristics which have been reported to closely
interact with the morphological evolution and stability of step-pool features. However, detailed information of the three-dimensional hydraulics for step-pool morphology has been scarce due to the difficulty of measurement. To fill in this knowledge gap, we established a hybrid model based on the technologies of Structure from Motion (SfM) and computational fluid dynamics (CFD). The model used 3D reconstructions of bed surfaces with an artificial step-pool unit built by natural stones at six flow rates as inputs for CFD simulations. The hybrid model succeeded in providing high-resolution visualization
of 3D flow structures for the step-pool unit. The results illustrate the segmentation of flow regimes below the step, i.e., the integral jump at the water surface, streaky wake vortexes near the bed, and high-speed jets in between. The highly non-uniform distribution of turbulence energy in the pool has been revealed and two energy dissipaters with comparable capacity are found to co-exist in the pool. Pool scour development under flow increase leads to the expansion of the jump and wake vortexes but this increase stops for the jump at high flows close to the critical condition for step-pool failure. The micro-bedforms as grain
clusters developed on the negative slope affect the local hydraulics significantly but this influence is suppressed at pool bottom. The drag forces on the step stones increase with discharge before the highest flow is used while the lift force has a larger magnitude and wider varying range. Our results highlight the feasibility and great potential of the hybrid model approach combining physical and numerical modeling in investigating the complex flow characteristics of step-pool morphology.

## 1 Introduction

Step-pool morphology is commonly formed in high-gradient headwater streams (Montgomery, and Buffington, 1997; Lenzi, 2001; Church and Zimmermann, 2007; Zimmermann et al., 2020). This bed structure has shown numerous benefits in providing diverse habitats for aquatic organisms (Wang et al., 2009; O' Dowd and Chin, 2016), efficiently dissipating flow energy (Wilcox et al., 2011; D'Agostino et al., 2015; Zhang et al., 2020) and enhancing channel stability (Abrahams et al. 1995; Wang et al., 2012). With these advantages, artificial step-pool systems mainly composed of boulders mimicking natural
channel morphology have been applied in restoration projects in steep channels with the objectives of improving local ecology and riverbed stability (e.g., Chin et al., 2009; Wang et al., 2012; Smith et al., 2020). To facilitate the application of artificial





step-pool systems, advanced understanding of the morphology, hydraulics, stability of step-pool features, and the interaction between these dimensions is needed. The high-resolution information of both topography and hydraulics for step-pool features is the key to fully reveal and describe these characteristics.

Recently, advanced information on the morphological evolution of step-pool features has been obtained by the rapidly developing technology Structure from Motion with Multi View Stereo (SfM-MVS, together referred to as SfM in this paper) photogrammetry (e.g., Golly et al., 2017; Zhang et al., 2018, 2020; Smith et al., 2020). SfM photogrammetry provides products with high spatial resolution and precision by using easily accessible customer-grade cameras or unmanned aerial vehicles (UAV) systems (Morgan et al., 2017). Although detailed topographic information has been available, access to high-resolution

hydraulic information remains limited for step-pool features. This incompatibility in the spatial resolution between morphological and hydraulic data hinders advancements in understanding how these two aspects interact with each other.

Different from topography, detailed measurements of the 3D flow properties of a step-pool unit are rarely accessible due to the appearance of highly non-uniform, aerated and turbulent flow regimes which result in the oscillation between supercritical (jet) and subcritical (jump) flow conditions (Church and Zimmermann, 2007; Wang et al., 2012; Zhang, 2017; Zimmermann

et al., 2020). Salt or rhodamine-dilution and tracer-based techniques (e.g., Waldon et al., 2004; Zimmermann et al., 2010) were used to characterize reach-scale flow properties in step-pool morphology, which however can hardly reflect the non-uniform features of hydraulics along the sequence. Point measurements for flow velocity around step-pool features could be achieved by using an Acoustic Doppler Velocimeter (Wilcox et al., 2011; Li et al., 2014) or electromagnetic current meter (Wohl and Thompson, 2000; Wilcox et al., 2011). Such measurements have the merit of high temporal resolution but with limited spatial

resolution as the arrangement of measured points is significantly affected by the rough beds and shallow water depths (Wilcox et al., 2011). These techniques are also more suitably used at low to moderate flows rather than at high flows during which significant sediment transport may occur and threaten the safety of such equipment. Particle tracking velocimetry (PTV, Maas et al., 1993) and particle image velocimetry (PIV, Adrian, 2005) techniques have been applied to measure the flow field for step-pool units in flume experiments (Zhang et al., 2018, 2020). The PTV method managed to visualize the recirculation at the

step toe and jet impinging at the pool bottom near flume side walls, while the PIV method presented the strong contrast of surface flow velocities at the step and pool areas. However, these measurements were limited at the side walls and water surface. Another problem was that the highly non-uniform flow characteristics led to uneven distribution of tracer particles over step-pools, leading to reduced accuracy in areas with low density of tracer particles (e.g., Zhang et al., 2020).

Nevertheless, the challenges in directly measuring the non-uniform hydraulic features of a step-pool unit present opportunities

for 3D computational fluid dynamics (CFD) modeling. CFD simulations have been applied in research addressing flow dynamics with highly turbulent free surfaces generated by complex structures in the channel (e.g., Thappeta et al., 2017; Xu and Liu, 2016, 2017; Lai et al., 2021; Zeng et al., 2021) or irregular boundaries of the channel (e.g., Roth et al., 2020). This numerical approach has shown great promise in characterizing and visualizing complex 3D hydraulic features at high spatial



and temporal resolutions. Furthermore, the flow forces on structures or topography which directly drive the interaction between
hydraulics and morphology can also be captured by CFD modeling (e.g., Xu and Liu, 2016; Chen et al., 2019).

The CFD approach has been applied in some numerical studies involving step-pool features which conceptualized a step-pool
sequence with highly simplified geometry mimicking the stepped spillway with flat surfaces (e.g., Thappeta et al., 2021).
Although this simplification reflects the unit-scale geometric properties of step-pools (e.g., step length and height, pool
inclination), it fails to characterize the sub-unit geometry including the transverse variability in the topography of step crests
(Wilcox et al., 2011), the shape of the scour hole (Comiti et al., 2005), and the development of grain clusters in the pool which
also affect the flow regimes in step-pool features (Zhang et al., 2020). Furthermore, to our knowledge flow forces on step-
pools have not been simulated in CFD models and have only been analyzed theoretically (Weichert, 2005; Zhang et al., 2016).
Therefore, we see great potential of CFD simulations using configurations that reconstruct natural step-pool morphology by
the SfM method in capturing the high-resolution hydraulic properties and flow forces of step-pool features.

The objective of this study is to first establish a hybrid model as a combination of the SfM and CFD methods to acquire the
high-resolution three-dimensional hydraulics for a step-pool unit built with natural stones, and then examine the 3D distribution
of flow velocity, turbulence, coherent structures, and flow forces on the bed surface. To address our objectives, we first
processed the topographic models of the bed surface derived from SfM photogrammetry in the flume experiment of Zhang et
al. (2020) and employed them as the input geometry for the CFD simulation. After the CFD simulation was verified by the
hydraulic measurements in the experiment, we conducted analysis on the 3D distribution of hydraulics and flow forces. The
three-dimensionality of flow characteristics, mechanisms of energy dissipation and interaction between hydraulics and
morphological evolution for a step-pool unit are discussed while insights for the stability and failure of step-pool units are also
provided. Finally, the limitations of the hybrid modeling approach are summarized.

## 2 Methods

The general workflow to establish the hybrid model is presented in Fig. 1. The 3D topographic models of a step-pool unit were
obtained by the SfM method in the flume experiment of Zhang et al. (2020) and were used as inputs for the CFD simulations
which were verified with the measurements of the water surfaces. Details of the flume measurements and CFD simulation are
presented in Section 2.1 and 2.2 respectively, followed by the model verification in Section 2.3 and the processing methods
for the outputs of the hybrid model in Section 2.4.





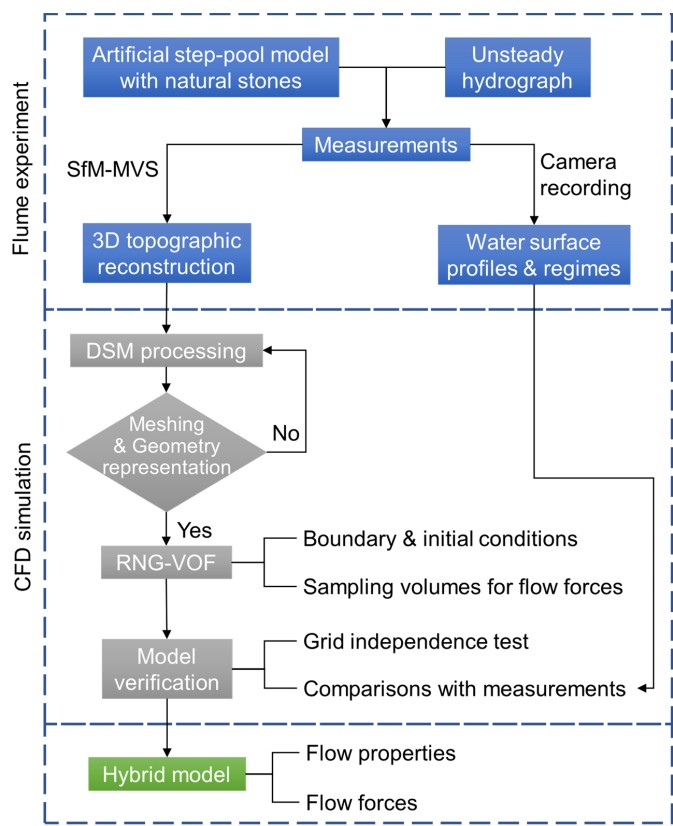


**Figure 1: Workflow of the hybrid modeling. SfM-MVS refers to the technology of Structure from Motion with Multi View Stereo. DSM is short for digital surface model. RNG-VOF is short for Renormalized Group (RNG) *k-ε* turbulence model coupled with Volume of Fluid method.**

### 2.1 Flume experiment

We used the measurements of bed topography and hydraulics in one of the runs from the flume experiments by Zhang et al. (2020) to establish the hybrid model. Since details of the flume system and experimental settings have been reported in detail in Zhang et al., (2018, 2020), only a brief description of the experimental setup is presented here.

The glass-steel-walled flume was 0.5 m wide and 0.6 m deep with a working length of 7.0 m. The initial slope of the sediment mixture was set at 3.2%. A top mounted camera ($1920 \times 1080$ px$^2$, with maximum frequency of 60 fps) was installed above

the flume to capture images of the surface flow regime, together with bed surface texture. Two side cameras were used to capture the longitudinal profiles of the bed and water surface near the flume walls. A step-pool model was manually constructed





by arranging six natural stones (Fig. 2a, Zhang et al., 2018). The step model was designed based on a Froude-scaled model ratio of 1:8, simulating the step-pool units formed in the reach with a channel width of 4.0 m (e.g., Chartrand et al., 2011; Recking et al., 2012). Another step called the guardian step was also built at the downstream of the step model (Fig. 3a) to

protect the step model from retrogressive erosion in each run. We did not manually build any pool features but allowed local scouring to form the pool morphology during each run.

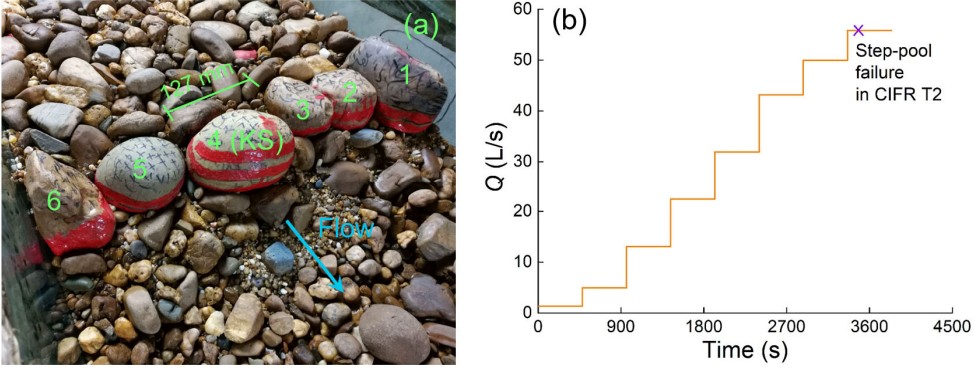

**Figure 2: Flume experiment settings in Zhang et al., (2020): (a) the artificially built-up step-pool model using natural stones, with stone number labelled; (b) the unsteady hydrograph of the run of CIFR (continually-increasing-flow-rate) T2 used in this study.**


Three CIFR (continually-increasing-flow-rate) T runs were conducted under designed unsteady hydrographs with step-by-step increase of flow to simulate the rising limbs of flood events in mountain streams. The flow was stopped to measure bed topography by SfM photogrammetry before it was increased to the next level in these runs. CIFR T2 (Fig. 2b) was chosen from the three runs as this run utilized a constant and relatively long discharge change interval (8 min) and showed prominent

pool features at high flows (Zhang et al., 2020). The outputs of SfM photogrammetry obtained in CIFR T2 were used in building the hybrid model. The designed discharge peak in CIFR T2 was 56.1 L/s, downscaled from the critical flow condition to destabilize natural step-pools (Lenzi, 2001; Turowski et al., 2009). The topographic measurements of the bed surface at the end of six flow conditions (5 L/s, 12.8 L/s, 22.8 L/s, 32.1 L/s, 43.6 L/s and 49.9 L/s) in this run were available before the step model collapsed (Zhang et al., 2020). The topography of the step structure remained stable while pool scour continued to

develop as the flow increased in CIFR T2.

During SfM measurements, image overlap > 80% in forward and side directions between two continuous photographs was used to guarantee the reconstruction quality (Javernick et al. 2014; Morgan et al. 2017). Four ground control points (GCPs) fixed at the side steel frames of the flume around the step-pool-step model were measured by a laser distance meter with a precision of 2 mm. The SfM measurements mainly covered the area taken up by the step-pool model (Fig. 3a). The digital



surface models (DSMs) established by the SfM workflow showed relatively low quality for the upstream area of the step model as the steel frames and other facilities here restricted image collecting sometimes. The DSMs at different flow rates were cropped for this area and had different distances (from 25 to 45 cm) between the upstream ends and the KS in the step. The quality of the DSMs near the side walls was also relatively poor as the corresponding photographs recorded the reflections of the bed surface on the glass. This resulted in incorrect feature matching for SfM and thus marginal areas of the DSMs were

cut and cleaned in Meshlab (version 2016.12, Cignoni et al., 2008). The widths of the DSMs were generally about 1.5-2 cm smaller than the flume width. The surface flow regime together with the surface grain size distributions (GSDs) in the pool was recorded by the top camera (see details in Zhang et al., 2018, 2020). The longitudinal profiles of the bed and water surfaces near the side walls were captured in the photographs taken by the side cameras every 2 seconds.

### 2.2 CFD simulation

The DSMs of the bed surface were further processed in the open-sourced software Blender (https://www.blender.org/) to fill holes and remove spikes and self-intersections, and then the model was remeshed with relatively uniform grids sized of 3.3-3.9 mm. This gridding method provided spatial resolutions high enough to characterize the detailed geometric features of the step-pool model used in the experiment and reduced the requirements for computing resources of the numerical simulations within the capacity of our workstation (CPU: Intel Xeon Gold 6230R × 2; Memory: 16 GB × 12).

The commercial solution FLOW-3D (v11.2) was utilized as the computational platform which applies the finite-volume method on a Cartesian coordinate system (Flow science, 2016). FLOW-3D has shown good performance to trace the free surface of water by the TruVOF technique, a special Volume of Fluid (VOF) method (Hirt and Nichols, 1981). Structured rectangular gridding incorporated with the FAVOR™ (Fractional Area Volume Obstacle Representation) technique is employed in FLOW-3D for meshing of the computational domain. FAVOR™ is a powerful discrete method to incorporate

geometry into the governing equations at the computational rectangular grids and enables the highly efficient characterization of complex geometric shapes (Flow science, 2016). 3D solid entities rather than 3D surfaces are required to build the terrain boundary in model setup (Flow science, 2016). Hence, the DSMs of bed surface (Fig. 3a) were extruded into solid entities in Blender first as the main geometry component (Fig. 3b) and then tested by the FAVOR™ technique (Fig. 3c).

The limited lengths of the bed surface captured in topographic models resulted in the negative slope in the pool located near

the downstream ends of the SfM reconstructions (Fig. 3a). If we set the downstream end as the outlet boundary, the effects of backwater would emerge near the outlet and cause a significant deviation of numerical results from experimental observations. To solve this problem, we extended the outlet by adding cubic components connecting to the reconstructed bed surface at downstream (Fig. 3b). These downstream components had a length of 30-50 cm, the same width with the step-pool component, and similar slopes with the bed surface measured by the side cameras. When leaks emerged between the DSMs of bed surfaces

and computation domains near both flume sides due to the cropping of DSMs, rectangular columns were added to fill in these leaks (Fig. 3b).

Earth **Surface**
**Dynamics**
Discussions

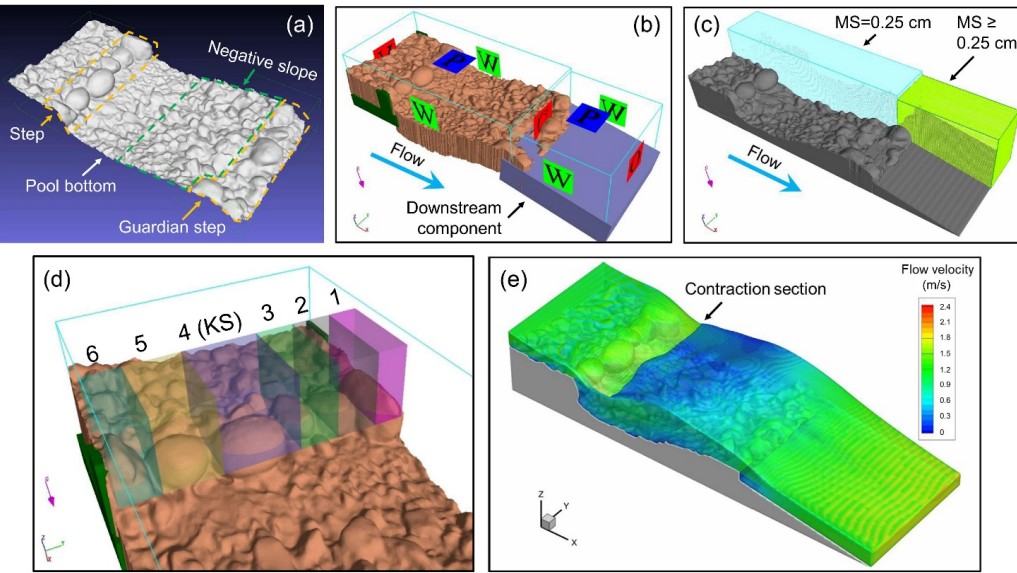

**Figure 3: Setup of the CFD model: (a) three-dimensional digital surface model (DSM) of the step-pool unit by structure from motion with multi view stereo (SfM-MVS) method as the input to the 3D computational fluid dynamics (CFD) modeling; (b) extruded bed**
**surface model connected to the extra downstream component (in purple blue) and rectangular columns to fill leaks (in green), with the boundary conditions shown on mesh planes; (c) recognized geometry with mesh grids of two mesh blocks shown where MS is short for mesh size; (d) sampling volumes to capture the flow forces acting on each step stone at X, Y, and Z directions; and (e) an example for the simulated 3D flow over the step-pool unit colored by velocity magnitude at the discharge of 49.9 L/s. The abbreviations for boundary conditions in (b) are: V for specified velocity; C for continuative; P for specific pressure; and W for wall**
**condition. The contraction section in Figure (e) refers to the edge between the jet and jump at water surface.**

The gravity model was activated and the gravitational acceleration was set at -9.81 m/s$^2$ along the vertical direction, i.e., Z axis in FLOW-3D. The Renormalized Group (RNG) $k$-$\varepsilon$ turbulence model was employed for turbulence simulation to account for the effects of smaller eddies compared to standard $k$-$\varepsilon$ turbulence model (Flow science, 2016). The VOF technique was
activated to accurately capture the free surface dynamics of the water flow.

We used 2-3 structured mesh blocks to define the total computational domain (Fig. 3c). One mesh block with a uniform grid size of 2.5 mm was used to cover the step-pool component acting as the main computational mesh block. The inlet boundary of the main computational domain was located about 24-37 cm of the upstream of the KS in the step, depending on the coverage of cropped DSM for the upstream area of the step. The setting of grid size achieved mesh independence (see details in Section
S1 in the supplemental materials) and the total grid number of the main computational domain ranged from 6.5 to 9.4 million



units. Non-uniform meshes sized from 2.5 to 5 mm covered the downstream areas connected to the step-pool features to save computational resources.

The boundary condition settings as exhibited in Fig. 3b were as follows: we used a specified velocity boundary with a fixed flow velocity and depth at the inlet cross sections to match the inflow discharge and water depth (measured by the side cameras) with the experimental conditions; no-slip wall boundary conditions were applied for the bed surface model and side walls; continuative boundary conditions were used for the interface between the mesh blocks and the outlet of the computational domain; specified pressure boundaries for the top faces of all the mesh blocks were applied and the fluid fraction was set at 0 for the air phase.

A still fluid region simulating the ponded water in the pool area was set as the initial condition to submerge the complex morphological features of the bed surface which efficiently accelerated the pressure convergence in the calculation in our study. We set one sampling volume for each step stone in which the components of flow forces including drag and lift forces on the bed surface were traced (Fig. 3d). To note, the lower boundary of the sampling volumes was set at elevations similar to the bed surface at the upstream of step stones (Fig. 3d) rather than at the elevations lower than the bed surface in the pool. This stems from the fact that the bed surface was impermeable in the CFD model (see Point 1 in Section 4.5). The simulation results (e.g., Fig. 3e) were collected after the solution was steady, with the variation from the mean less than 0.5% at each flow rate. A period of 30 seconds of the outputs (e.g., flow velocity, pressure) were extracted at a frequency of 2 Hz to obtain the time-averaged values for further processing and analysis.

### 2.3 Model verification

We both conducted the grid independence test and compared between the simulated and experimental results for model verification. The grid independence was reached when the grid size of 0.25 cm was used for the main computation domain in modeling. Two measurements (Fig. A3) in the previous flume experiments (Zhang et al., 2018, 2020) were used to validate the numerical models: (i) longitudinal water surface profiles extracted from the side cameras; and (ii) water surface regime recorded in pictures by the top view camera. The mean error (*ME*), mean absolute error (*MAE*), mean square error (*MSE*), root mean square error (*RMSE*) and standard deviation (*SD*) were calculated for the differences between the simulations and measurements from the side views (Table 1) and the top views (Table A1). The max *RMSE* of the simulated water surface is below 2 cm for side views (Table 1) and smaller than 3 for the boundaries between the jet and jump regimes from the top views (Table A1). The comparisons between simulated results and the measurements show that the hybrid model succeeded in capturing the flow characteristics for a step-pool feature built in the physical flume. Detailed descriptions of the model verification are presented in Appendix A.





**Table 1: Error indices of the simulated water surface elevations at both sides**

|  | Q (L/s) | ME (cm) | MAE (cm) | MSE (cm) | RMSE (cm) | SDE (cm) |
|---|---|---|---|---|---|---|
| Left side | 5 | 0.07 | 0.21 | 0.10 | 0.32 | 0.31 |
|  | 12.4 | 0.50 | 0.51 | 0.36 | 0.60 | 0.00 |
|  | 22.8 | 0.33 | 0.44 | 0.27 | 0.52 | 0.22 |
|  | 32.1 | 0.37 | 0.71 | 0.72 | 0.85 | 0.68 |
|  | 43.6 | 0.33 | 1.16 | 1.64 | 1.28 | 1.19 |
|  | 49.9 | 0.53 | 0.76 | 0.70 | 0.84 | 0.39 |
| Right side | 5 | 0.11 | 0.29 | 0.12 | 0.34 | 0.30 |
|  | 12.4 | 0.07 | 0.38 | 0.22 | 0.47 | 0.46 |
|  | 22.8 | -0.09 | 0.40 | 0.44 | 0.67 | 0.65 |
|  | 32.1 | 0.35 | 1.23 | 2.64 | 1.63 | 1.55 |
|  | 43.6 | 0.53 | 1.42 | 3.81 | 1.95 | 1.80 |
|  | 49.9 | 0.31 | 1.14 | 1.70 | 1.30 | 1.23 |

**2.4 Data processing**

The kinetic energy (*KE*), turbulent kinetic energy (*TKE*), and turbulent dissipation ($\varepsilon_T$) were used in the analysis of turbulent

210    features in the step-pool unit. The turbulent dissipation was obtained when solving the RNG *k-ε* turbulence model, whereas

the kinetic energy and turbulent kinetic energy were calculated by Eqs. 1 and 2.

$$KE = \frac{1}{2}\left(u_x^2 + u_y^2 + u_z^2\right),$$    (1)

where *u* denotes the instantaneous velocity in three directions.

$$TKE = \frac{1}{2}\left(u_x'^2 + u_y'^2 + u_z'^2\right),$$    (2)

215    where $u'$ denotes the instantaneous velocity fluctuation in three directions.

The Q-criterion (Hunt et al., 1988; Flow science, 2016) was used to calculate and visualize the coherent flow structures in the

step-pool unit and the $Q_{criterion}$ was calculated by Eq. 3. We used a threshold value of 1200 for $Q_{criterion}$ to isolate coherent

vortexes in this study.

$$Q_{criterion} = \frac{1}{2}\left(\Omega_{ij}\Omega_{ij} - S_{ij}S_{ij}\right),$$    (3)

220    where $\Omega_{ij}$ and $S_{ij}$ are the antisymmetric and symmetric parts of the velocity gradient tensor, respectively.



The shear stress and total pressure for the mesh grids on the bed surface were extracted from simulations. The shear stress was used directly in the analysis while the total pressure was further processed to obtain the dynamic pressure, which stemmed from the kinetic energy of the flow. The dynamic pressure ($P_d$) working on each mesh grid in bed surface was calculated by subtracting the static water pressure ($P_s$) from the total pressure ($P_t$).

$$P_d = P_t - P_s = P_t - \rho gh \tag{4}$$

where $\rho$ is the water density; $g$ is gravity acceleration; and $h$ is the water depth at the mesh grid in bed surface.

Drag ($C_D$) and lift ($C_L$) coefficients of the drag ($F_D$) and lift ($F_L$) forces acting on the step stones in the sampling volumes (Fig. 1) were calculated by using Eqs. 5 and 6.

$$C_D = \frac{2F_D}{\rho U_\infty^2 A_\perp} \tag{5}$$

$$C_L = \frac{2F_L}{\rho U_\infty^2 A_\perp} \tag{6}$$

where $U_\infty$ is the approach velocity and $A_\perp$ is the upstream projected area of the step stone in each sampling volume. The sectional-averaged flow velocity at the upstream face of the sampling volume was used as the approach velocity.

When calculating the section-averaged turbulent kinetic energy ($TKE$) for the jump and wake vortexes separately, we used the threshold method to distinguish the areas taken by them. Since the $TKE$ in the jet was far lower than that of the jump and wake (see details in Section 3.1.2), the threshold slightly higher than the maximum of $TKE$ in the jet was used to detect the boundaries of jump and wake vortexes in each vertical line in a cross section. The areas taken by the jump and wake vortexes in each cross section were then obtained, together with the integral of $TKE$ in these areas. These two parameters were used to calculate the section-averaged $TKE$.

## 3 Results

The spatial distributions of both hydraulic characteristics and flow forces in a step-pool unit are exhibited in this section. To clearly present these distributions, only the scenarios under the largest two discharges ($Q$=43.6 and 49.9 L/s) are shown in most of the analysis, while the rest are exhibited in Appendix B. These two discharges were chosen mainly for two reasons: (i) well-defined pool morphology showed up under the two flow conditions, and (ii) the largest discharge (49.9 L/s) recorded the topographic and hydraulic characteristics closest to the failure of this step-pool unit in the experiment and may present clues to the failure mechanism of a step-pool feature.


Earth **Surface** Dynamics
Discussions

### 3.1 Flow properties

#### 3.1.1 Flow velocity

The distribution of time-averaged flow velocity magnitude in three longitudinal sections is presented in Fig. 4, as well as the distribution of Froude number in Fig. A8. Flow accelerates and water depth decreases over the step stones before plunging

into the pool in the jet regime. As a result, the Froude number reaches its maximum at the step crest (Fig. A8). The highest flow velocity in the vertical profile at the crests of step stones mainly exists near the stone surface (Fig. 4), rather than near the water surface as it appears at the upstream of the step.

The pool area under the two flow conditions exhibits highly non-uniform flow fields in all the three longitudinal sections before the flow starts to accelerate on the negative slope (Fig. 4): low velocity magnitudes close to 0 in the hydraulic jump

near the water surface; high flow velocities (generally > 1 m/s) in the jet as the main flow; and low flow velocities at the step toe and along the bed surface in the pool. Worth noting is that the jet impinges at the bed surface in the pool in the section Y = 0 and 13.5 cm but does not hit the bed in the longitudinal section Y = -18 cm even though distinct scour also occurs near this section. The jet is deviated from the bed by the vortex formed at the step toe as a result of wake turbulence in the section Y = -18 cm. The comparison among the three sections indicates that highly three-dimensional flow structures in step-pool

feature exist. The larger discharge and water depth at $Q$ = 49.9 L/s result in the limitation of jump regime in the three sections but expansion of wake zone in the section Y = -18 cm comparing with the case at $Q$ = 43.6 L/s.

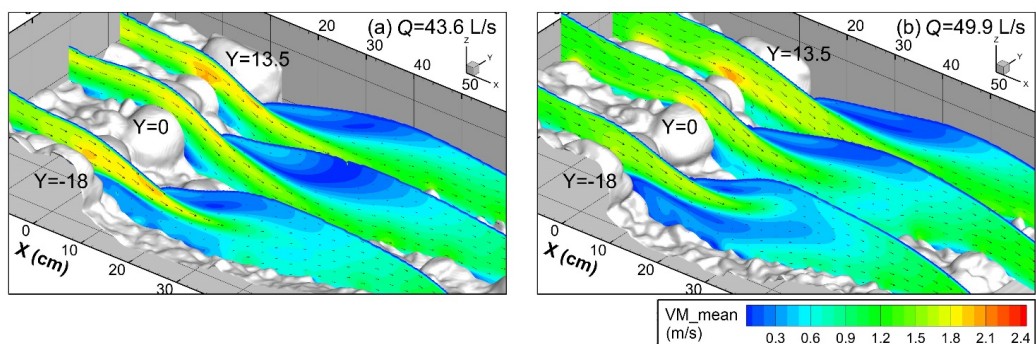

**Figure 4: Distribution of time-averaged velocity magnitude (VM_mean) and vectors in three longitudinal sections. The section at Y = 0 cm goes across the keystone while the other two (Y = -18 and 13.5 cm) are located at the step stones beside the keystone with**
**lower top elevations. $Q$ refers to the discharge at the inlet of the computational domain. The spacing for X, Y, and Z axes are all 10 cm in the plots.**

The transverse distribution of flow velocity magnitude is presented in Fig. 5, with five cross sections from the upstream to downstream side of the step-pool model exhibited. Section x0-18 is located at the upstream area of the step where no distinct


Earth **Surface**
Dynamics
Discussions

bed structures have developed. The water surface is relatively flat and velocity magnitude is relatively uniformly distributed
in this section. The x0-6 section, which is located at the step crest near the detaching point of the jet, shows that flow
concentrates at the lower top elevations of the step. The section at x0+2 cm is located at the upstream of the contraction section
for flow rates > 12.4 L/s and shows the existence of discrete vortexes near the bed surface whose dimensions expand with an
increase in discharge (Fig. 5 and A10). The centers of the vortexes follow the lower top elevations of the step, i.e., the

connecting points between step stones. The gaps between the wake vortexes near the bed are filled with high-speed flows. The
locations of these gaps correspond to the higher top elevations of the four step stones between the bank stones. In the section
at x0+15 cm near pool bottom at $Q$ = 32.1, 43.6 and 49.9 L/s, the wake vortexes shrink and show reduced velocity differences
with the jet if compared with the section at x0+2 cm. The jump regime with flow velocities close to 0 covers almost the entire
flume width at this section. As a result, high velocity magnitude appears in the middle of the vertical profile in most areas of

this section. The section at x0+40 cm is located on the negative slope and near the pool tail when the pool scour is fully
developed and shows no sign of the vortexes near the bed. The surface jump extends to this section but influences only part of
the flume width and flow velocity becomes relatively uniform beneath the surface jump. As the water depth increased in all
the five cross sections from 43.6 L/s to 49.9 L/s, the drop of flow velocity can be found in the sections x0-6 and x0+2, as well
as the enlargement of wake vortexes in the sections x0+2 and x0+15.

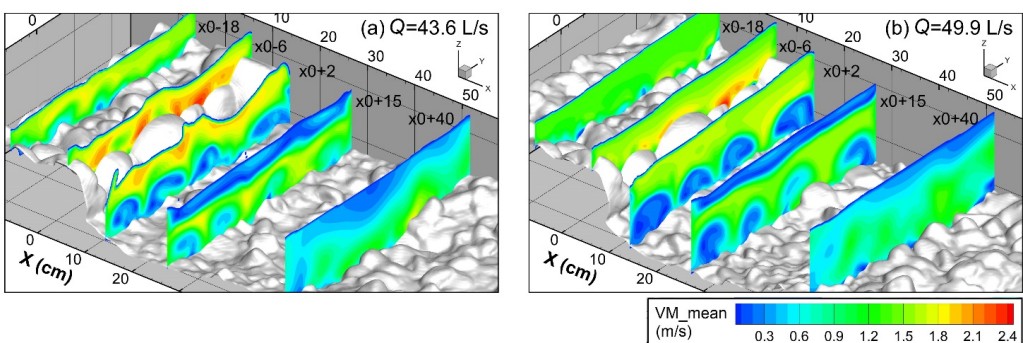


**Figure 5: Distribution of time-averaged flow velocity at five cross sections which are set according to the reference section (x0). The
reference cross section x0 is located at the downstream end of the keystone (KS). The five sections are located at 18 cm and 6 cm
upstream of the reference section (x0-18 and x0-6), and 2 cm, 15 cm and 40 cm downstream of the reference section (x0+2, x0+15,
x0+40). The spacing for X, Y, and Z axes are all 10 cm in the plots.**


### 3.1.2 Turbulence

Figure 6 presents the transverse distribution of turbulence kinetic energy (*TKE*) in the same cross sections with Fig. 5. The
*TKE* at the upstream of the step (section x0-18) and at the step (section x0-6) are generally at a much lower level if compared





Earth **Surface**
**Dynamics**
Discussions

with the pool. The area of low turbulence intensity overlaps with the area of high flow velocity (Fig. 5), indicating that high

flow velocities in the upstream area of the step limits the development of turbulence.

The distribution of *TKE* in the pool also exhibits high non-uniformity at the highest flow conditions. At the upstream of the contraction section (section x0+2), high *TKE* is only located at the wake turbulence of the step stones above bed surface while the jet with high flow velocity shows low turbulent energy near the water surface. Around the deepest area in the pool (section x0+15), both the jump at the water surface and wake vortexes show high turbulent energy, and much higher *TKE* is contained

in the jump above the jet if we further compare the *TKE* level of both. In the section x0+40 on the negative slope, the jump and wake have been mixed up and turbulent energy decreases from water surface to bed surface in the vertical direction. It is worth noting that for both the jump and wake vortexes, the highest dissipation occurs near the interfaces with the jets, i.e., at the bottom of the surface jump and the top edges of wake vortexes respectively (e.g., section x0+15). The increase of water depth and decrease of flow velocity from 43.6 L/s to 49.9 L/s lead to the significant limitation of *TKE* level near the bed surface

in the two sections (x0-18 and x0-6) at the upstream of the step and in the high-speed jets in the pool (sections x0+2 and x0+15).

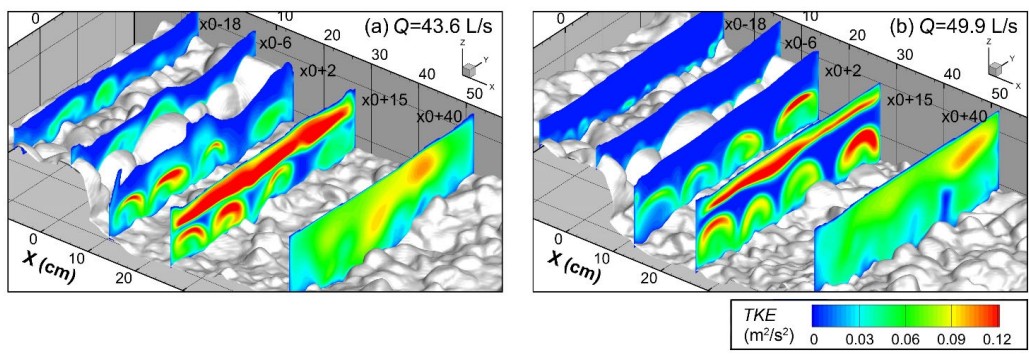

**Figure 6: Distribution of the time-averaged turbulence kinetic energy (*TKE*) at the five cross sections same with Figure 3.**

To present the transformation of flow energy in the pool, we plot the distribution of mass-averaged *KE*, *TKE* and $\varepsilon_T$ at the downstream area of the reference cross section x0 with a length of 50 cm in Fig. 7. The key findings are as follows:

First, at all the discharges examined, the kinetic energy of flow decreases after flow plunges into the pool but shows a slightly increasing trend on the negative slope (Fig. 7a to f). Worth noting is that at the two highest discharges (Fig. 7e and f), the flow kinetic energy remains at a high level at a distance of 5-6 cm at the downstream of x0 as the regime of jet before it decreases

dramatically. Second, the *TKE* first increases in the pool and reaches the maximum around the pool bottom, and then decreases on the negative slope (Fig. 7g to l). The location where the maximum of *TKE* shows up moves to the downstream as flow



increases, during which pool scour keeps developing and the pool bottom area also moves to the downstream (Zhang et al., 2020). Third, the turbulent dissipation increases sharply at the downstream area of x0 and reaches the maximum earlier than the *TKE* in the pool. The turbulent dissipation rate on the negative slope remains at a low level, even lower than that near the
step toe (Fig. 7m to r). Fourth, the maximum value of flow kinetic energy, *TKE* and turbulent dissipation in the pool increases during a flow increase from 5.0 to 43.6 L/s, but decreases when the flow further increases to 49.9 L/s with further occurrence of pool scour.



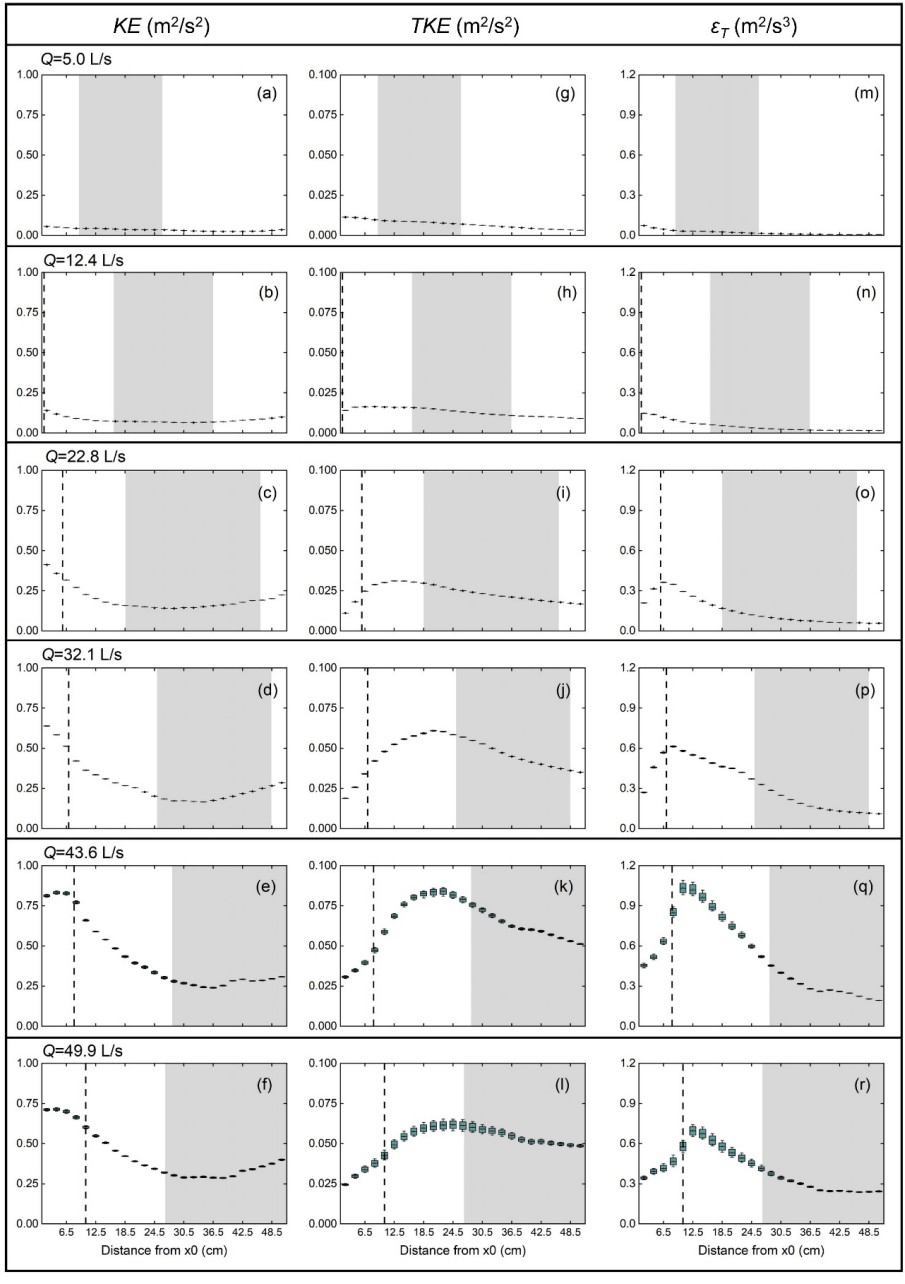



**Figure 7: Boxplots for the distributions of the mass-averaged flow kinetic energy (*KE*, panels a-f), turbulence kinetic energy (*TKE*, panels g-l), and turbulent dissipation ($\varepsilon_T$, panels m-r) in the pool for all the six tested discharges (the plots at the same discharge are in the same row). The mass-averaged values were calculated every 2 cm in the streamwise direction. The flow direction is from left to right in all the plots. The general locations of the contraction section for all the flow rates are marked by the dashed lines, except for $Q = 5$ L/s when the jump is located too close to the step. The longitudinal distance taken up by negative slope in the pool for the inspected range is shown by shaded area in each plot.**

### 3.1.3 Coherent flow structure

We present the instantaneous vortex structures in Fig. 8 (also showing the front view in Fig. A12). In the upstream area of the step, streamwise coherent structures are mainly located near the bed. When the flow rate is larger than 32.1 L/s, the flow structures show streaky features near the bed surface at the downstream of protruding grains.

Rich coherent structures exist at the downstream area of the step as a combination of vortexes stretched across the entire channel width near the water surface and discrete streamwise streaky vortexes close to the bed. The dimensions of both the surface jump and wake vortexes expand as the flow rate increases and pool scour develops. No clear coherent structures are visualized in the high-speed jet in the pool, indicating low vorticity. A wake vortex starts at the contacting point of two neighboring step stones, and its width and height keep decreasing to the downstream direction until the vortex vanishes near the start of the negative slope. The thickness of the hydraulic jump reaches the maximum near the pool bottom where water depth is the largest in the pool and then decreases as the jump regime fades away on the negative slope. The configuration of the surface jump is significantly affected by the distribution of wake vortexes: upper bends exist above the wake vortexes while downward bends appear at the gaps between two neighboring wake vortexes (e.g., Fig. 8d to f). On the negative slope, coherent structures mainly follow protruding grains in the micro-scale bed structures but do not show streaky features as at the upstream area of the step, where the grain sizes are similar with those on the negative slope.





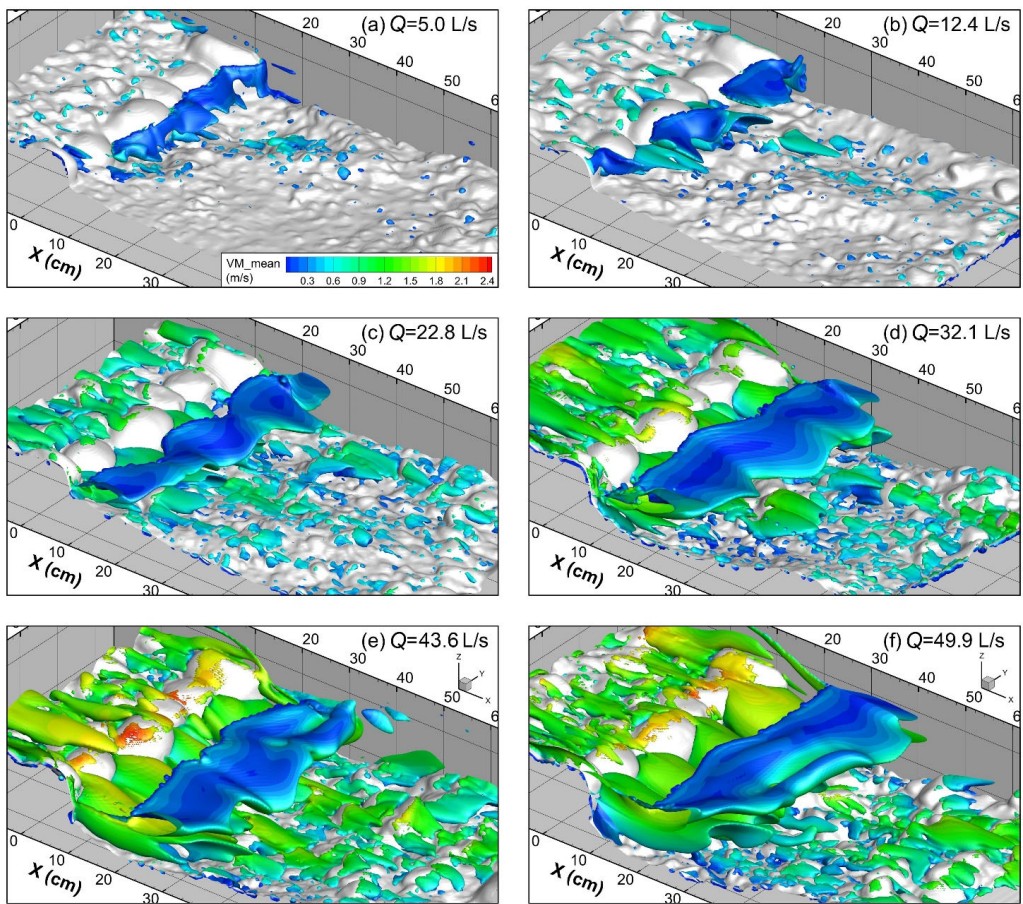

**Figure 8: Instantaneous flow structures extracted using the Q-criterion ($Q_{criterion}$=1200) and colored by the magnitude of flow velocity.**

### 3.2 Flow forces

**3.2.1** Dynamic pressure

For all the flow conditions, the dynamic pressure is at a relatively low level on the step stones and becomes even lower at the crests of step stones where the departure of the jet from step stones occurs. The dynamic pressure on the step stones generally decreases with flow rate and the development of pool scour. The minimum of dynamic pressure appears at the connection





between No. 2 and 3 stones at high flows with the existence of the highest flow velocity on the step (Fig. 9). Relatively high
dynamic pressure exists near the impinging point in the pool and its magnitude generally increases with flow rate (Fig. 9 and
Fig. A13). It is noteworthy that the relatively high values of dynamic pressure appear at the locations more downstream at $Q$
= 49.9 L/s than 43.6 L/s owing to the deposition of fine sediment at the step toe (Fig. 9b, Zhang et al., 2020). The dynamic
pressure at the pool bottom also shows higher values at $Q$ = 43.6 L/s than $Q$ = 49.9 L/s owing to lower water depth in the pool
at $Q$ = 43.6 L/s (Fig. 4-5) although the scour depth is larger at $Q$ = 49.9 L/s. The front sides of the protruding grains or grain
clusters to the flow on the negative slope show significantly lower dynamic pressures than on the back sides and surrounding
grains.

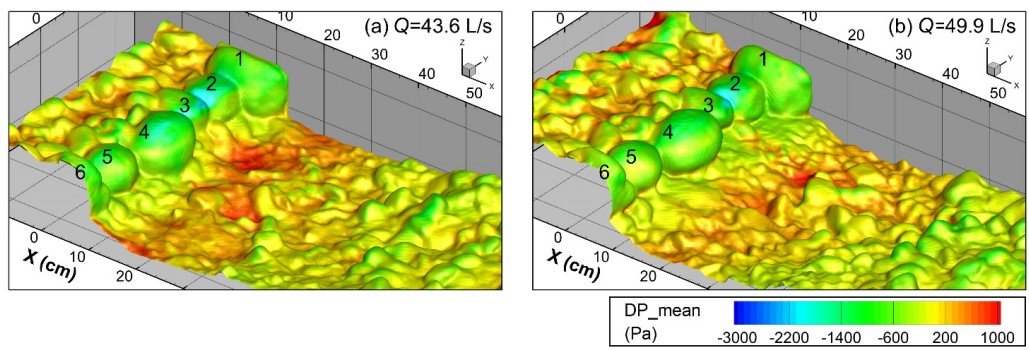

**Figure 9: Time-averaged dynamic pressure (DP_mean) on the bed surface in the step-pool model under the two highest discharges,
with the step numbers marked. The negative values in the plots result from the setting of standard atmospheric pressure = 0 Pa,**
**whose absolute value is 1.013×10⁵ Pa.**

### 3.2.2 Shear stress

The high-resolution distribution of shear stress on the bed surface at the highest flows is shown in Fig. 10. The magnitude of
shear stress along the step-pool model is generally two orders smaller than that of the dynamic water pressure. The step stones
bear the highest level of shear stress in the step-pool unit. Shear stress is further concentrated on the crests of the step stones.
The shape and top elevation of the step stones influence the distribution of shear stress significantly. No. 2 and 3 stones with
flat tops and lower top elevations show higher shear stress at the connection of these two stones but have quite low (close to
0) shear stress in almost the whole downstream faces of these two stones. In contrast, the shear stress on the No. 4 (KS) and 5
stones, which have an ellipsoid configuration, reaches a maximum near the highest elevation of each stone. The edges of high
shear stress zone in the back sides of these step stones show clear downstream curvature. Shear stress also shows higher values
where the bed is impinged by the flow and some protruding clusters/grains on the negative slope comparing with surrounding
grains. However, the highest shear stress in the pool only reaches about 50-70% of that on the step stones at high flows.





Earth **Surface**
**Dynamics**
Discussions

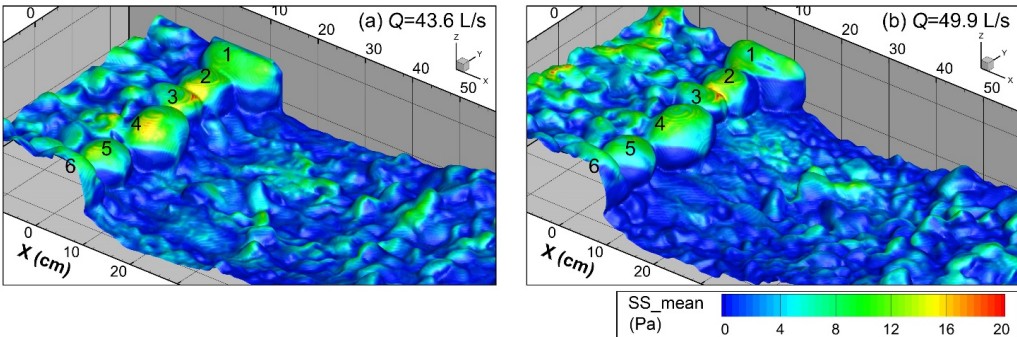

**Figure 10: Time-averaged shear stress (SS_mean) on bed surface in the step-pool model, with the step numbers marked. The**
**standard atmospheric pressure is set as 0 Pa.**

### 3.2.3 Flow forces on step stones

The variations of the components of flow forces on each step stone reveal the following patterns (Fig. 11). First, the component in the X direction, i.e., the drag force, on all the step stones keeps increasing until the flow rate reaches 43.6 L/s but decreases
when the flow is further enhanced to 49.9 L/s. The keystone (stone 4), which was the first stone to move and triggered the step failure in the experiment (Zhang et al., 2018), has the largest drag force at high flows. Second, the component in the Z direction of flow force, i.e., the lift force, generally has a larger magnitude than the drag force on step stones before the flow rate reaches 43.6 L/s. The lift force on the stones 1-4 turns the direction from downward to upward at $Q$ = 43.6 L/s, when flow velocity significantly increases at the step, but the water depth is similar with that at $Q$ = 32.1 L/s (Fig. 4 and Fig. A9). When the
discharge is further increased to 49.9 L/s and water depth shows a clear increase (Fig. 4-5), the lift force turns downward again. Third, the Y component on the step stones between the bank stones is about 2-3 orders smaller than the components in the other two directions. In contrast, the Y component of flow force has the largest magnitude for the two bank stones at the highest flow. This indicates that the transverse interaction between the step and the flow mainly occurs at the banks. Lastly, the magnitude of the resultant flow force increases when the discharge is enhanced to 49.9 L/s for the step stones except for stones
2-3, where the high flow velocity concentrates.





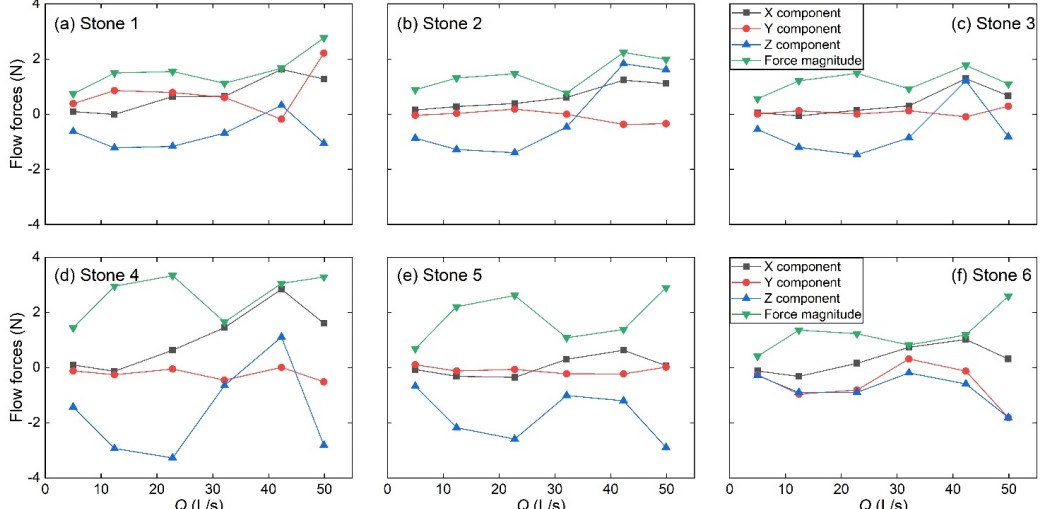

**Figure 11: Variation of fluid force components and magnitude of resultant flow force acting on step stones with flow rate. The stone 4 is the keystone. Stone numbers are consistent with those in Fig. 9-10. The upper limit of the sampling volumes for flow force calculation is higher than water surface while the lower limit is set at 3 cm lower than the keystone crest.**


We further show the non-dimensional drag and lift coefficients for each step stone at different flow conditions in Fig. 12. A generally increasing trend for drag coefficient is found for all the step stones when the discharge is larger than 12.4 L/s, although a slight drop is observed for all the stones except for stones 1 and 2 when the discharge is increased to 49.9 L/s from 43.6 L/s (Fig. 12c to f). In contrast to the drag force (Fig. 11d), the drag coefficient of the KS (stone 4) is amongst the lowest

of the step stones (Fig. 12d) while stone 2 shows the largest $C_D$ at all flow rates (Fig. 12b). The lift coefficient also shows an increasing trend after the discharge is larger than 12.4 L/s and decreases at $Q$ = 49.9 L/s, and the largest magnitude of $C_L$ appears at stone 2 for all flow rates. There is no significant change in the $C_D$ of the KS at the discharge of 22.8-49.9 L/s while the $C_L$ shows much more prominent variation.





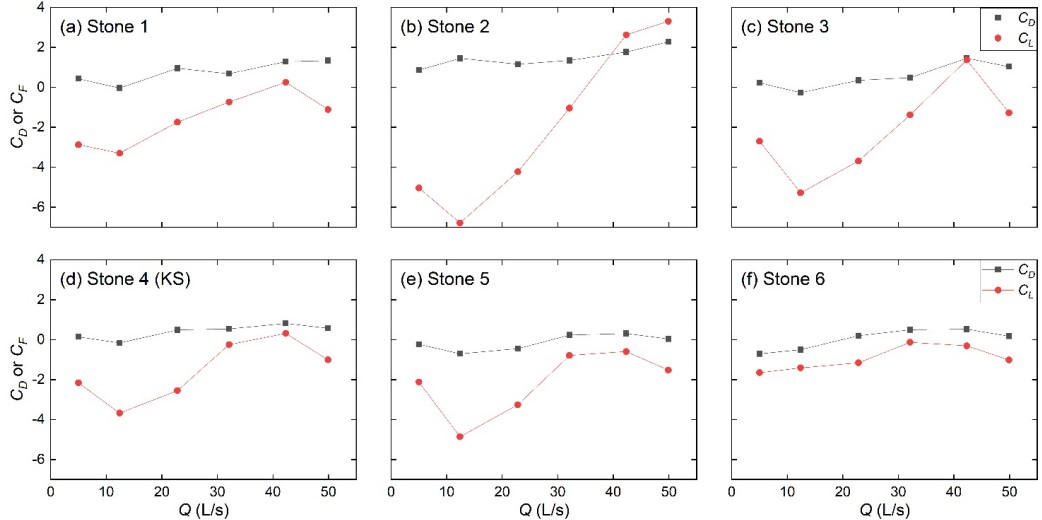

**Figure 12: Variation of drag ($C_D$) and lift ($C_L$) coefficient of the step stones along with flow rate. Stone numbers are consistent with those in Fig. 8-9. KS is short for keystone. The negative values of $C_D$ correspond to the drag forces towards the upstream while the negative values of $C_L$ correspond to lift forces pointing downwards.**

## 4 Discussion

### 4.1 Three-dimensionality of flow characteristics

Using the hybrid approach, we provided detailed description of the 3D flow properties at a millimeter-resolution around a step-pool unit made of natural gravels for the first time. Based on the results of this study, distinguished three-dimensionality of the flow structures in the pool is revealed: the wake vortexes below the step are discrete streaky structures, different from the surface jump as an integrated flow structure covering the entire flume width (Fig. 8 and Fig. A12). Natural grains used to build the step-pool unit with randomness and irregularity in size, shape, and orientation result in transverse inconsistencies of topography for the step (Fig. 2a). Our results show that the emergence of vortexes at the step toe is related to the lower elevations in the step while the higher elevations of step crests will be followed by the jet with enough kinetic energy to hit the bed surface directly. Wilcox et al. (2011) has noticed the possible influence of the variability in step architecture on the distribution of hydraulics and turbulence and the flow resistance of a step-pool sequence. Our results further reveal that the transverse configuration of a boulder step influences the flow characteristics of the downstream pool in a significant way.

The jet regime in which the flow eventually hits the bed is defined as an impinging jet, while it is defined as a surface jet if it remains at the water surface after plunging (Wu and Rajaratnam, 1998). The general jet regime for the whole step structure was recognized as an impinging jet in the CIFR T2 run (Zhang et al., 2020) based mainly on the water depths measured near



the flume walls. However, the 3D flow structures exhibit that both impinging jet and surface jet regimes coexist in the pool (Fig. 4-5). This inconsistency mainly stems from the limitation of measurements at the flume walls. In Zhang et al.'s (2018,

2020) experiments, the impinging jet was only visualized by particle tracing velocimetry near the right flume wall. The hybrid model reproduced this observation near the right wall and shows that the jet is deviated by the wake vortex and does not impinge the bed at the downstream of the right bank stone (Fig. 4 and 8), about 2-3 cm away from the right sidewall of the flume. Therefore, our results highlight the great advantage of the hybrid model in presenting fully resolved 3D hydraulic information which is necessary to achieve a comprehensive view of the flow structures over complex topography.

Our results also illustrate the segmentation of flow regimes in the pool area: hydraulic jumps at the water surface, streaky wake vortexes at the bottom, and high-speed jets in between. This segmentation of flow regimes remains in the pool until the flow reaches the negative slope (Fig. 4-5). The jet decelerates to a large degree after plunging into the pool, but still holds a much higher flow velocity magnitude than the jump and wake vortex (Fig. 4). The strong relative movement between the jet and the vortexes at the water surface and wake results in mid-profile shear that generates high level of TKE (Fig. 6). In this sense, the

3D simulated results illustrate the context of the non-logarithmic vertical profiles of flow velocity and turbulence below steps measured in the field which show higher flow velocity and turbulence in the middle (Wohl and Thompson, 2000; Li et al., 2014).

## 4.2 Energy dissipation mechanism

Energy dissipation of the flow for a step-pool unit has been reported to mainly occur in the pool area (Wohl and Thompson,

2000; Li et al., 2014; Zhang et al., 2020). With the distribution of flow velocity and kinetic energy, turbulent kinetic energy, and turbulent dissipation presented in detail by the hybrid model (Fig. 5-7), we further visualize the energy dissipation mechanisms in the pool. Both the distributions of *TKE* and turbulence dissipation in the pool exhibit high non-uniformity (Fig. 7). It is noteworthy that the energy transformation and dissipation is concentrated in the area at the upstream of the negative slope in the pool. Both the surface jump and wake vortexes show much higher *TKE* and turbulent dissipation than the high-

speed jets (Fig. 6), suggesting that two energy dissipators, i.e., the jump and wake, co-exist in this area. The surface jump has been recognized as the energy dissipator for a step-pool owing to its appealingly fluctuating appearance at the water surface (Church and Zimmermann, 2007; Wyrick and Pasternack, 2008; Wang et al., 2012; Zhang et al., 2018). However, little attention has been paid to the dissipation properties of the wake turbulence in the pool as most measurements would be blocked by the jump regime. The hybrid modelling makes it possible to compare the level of *TKE* in these two dissipators quantitatively.

We calculated the section integral and averaged (section integral values divided by the areas taken by jump or wake in the cross section) *TKE* for each dissipator before these two dissipators get mixed, as shown in Fig. 13.





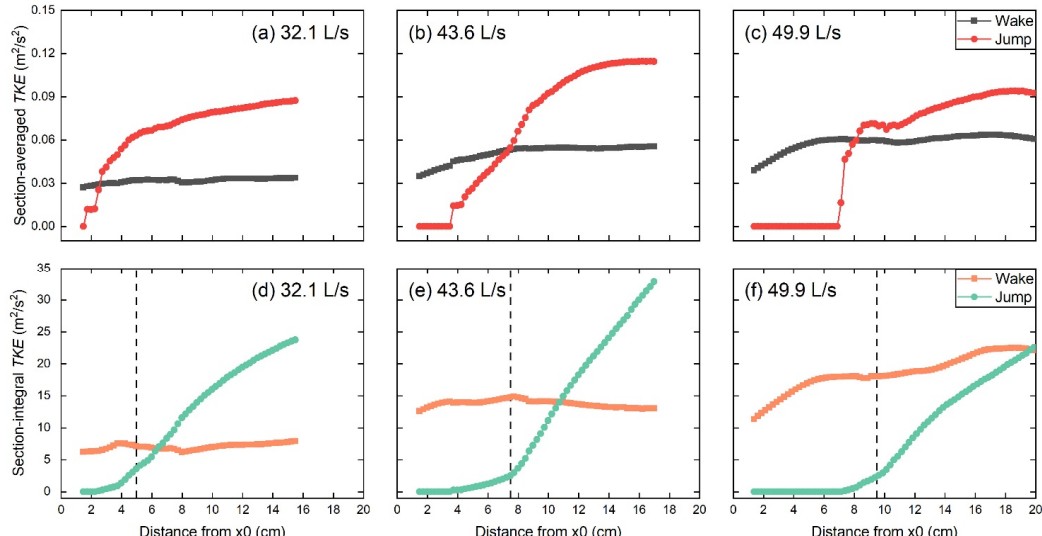

**Figure 13: Longitudinal distributions of section-averaged and -integral turbulent kinetic energy (*TKE*) for the jump and wake vortexes at the largest three discharges. The flow direction is from left to right in all the plots. The general locations of the contraction**
**sections under the three flow rates are marked by dashed lines in figures (d) to (f).**

At the downstream of the contraction section, the section-averaged *TKE* in the jump soon exceeds the value in the wake vortexes for all the three flow rates (Fig. 13a to c). In contrast, the section-integral *TKE* in the jump is higher than that of the wake for most cross sections at $Q$ = 32.1 and 43.6 L/s (Fig. 13d and e) but lower than that of the wake in almost all the cross

sections examined (Fig. 13f). After the jump regime fully develops, no difference of order is found in the section averaged or integral *TKE* between the jump and the wake for the streamwise length examined here. This indicates that the jump and wake are comparable contributors for energy dissipation in the pool. Worth noting that the *TKE* in the jump decreases when the discharge is increased to 49.9 L/s from 43.6 L/s whereas the *TKE* level in the wake vortexes sees a further increase. The suppression of *TKE* in the surface jump may be related to the higher submergence below the step and transition of jump regimes

at higher flow conditions (Pasternack et al., 2006; Wyrick and Pasternack, 2008; Zhang et al., 2020). The intensification of TKE in the wake zone of the step is associated with the development of pool scour as the flow increases (Zhang et al., 2020). This contrast suggests that the contribution of the wake vortexes to the total energy dissipation in the pool is enlarged with flow increases and pool development.

The step-pool morphology has been reported to show a higher capacity of flow energy dissipation than a vertical drop with the

same height (Zhang et al., 2020). The new understanding towards the mechanism of energy dissipation for step-pool features may provide two explanations for this phenomenon. First, the 3D natural step structure leads to 3D configurations of vortexes



downstream, which enlarges the interface between energy dissipators with the jet. As the interfaces are where high *TKE* concentrates (Fig. 6), the energy loss of the flow in the 3D wake turbulence at the step toe may surpass that of the 2D recirculation vortexes below a drop. Second, the pool geometry in a step-pool unit is normally more complex than an artificial

pool, and the local scour is intensified with flow increases until the step structure collapses (Comiti et al., 2005; Church and Zimmermann, 2007; Zhang et al., 2018, 2020). This morphological evolution maintains the co-existence of two energy dissipators for a step-pool unit and enlarges the energy capacity of the wake vortexes with increases in flow. In contrast, the fixed rectangular shape of a drop and the pool at the downstream results in significant suppression of the surface jump and limited space for the recirculation vortex to expand especially at skimming flow conditions (Chanson, 2001).

### 485 4.3 Interaction between hydraulics and morphological evolution

The distributions of flow velocity (Fig. 4-5), turbulence (Fig. 6-7) and coherent structure (Fig. 8) in the pool have visualized the expansion of both the jump and wake vortexes with the development of pool scour and flow increase. The expansion of the jump volume presented by the hybrid model is generally consistent with the experimental observations (Zhang et al., 2020). The results of this study further illustrate that both the geometric dimensions and *TKE* of the surface jump decrease when the

discharge is increased to 49.9 L/s from 43.6 L/s (Fig. 4-6, and 13) indicating that the increase of the submergence in the pool would suppress the surface jump regime at high flows. In contrast, the wake vortexes show an increase of geometric dimensions and *TKE* with this flow increase (Fig. 5-6, 8, and 13). This difference in variation patterns to the discharge increase to 49.9 L/s result from the change in jet penetration angle due to the increase of water depth (Fig. 4). The decrease of jet penetration angle at $Q = 49.9$ L/s also leads to the moving downstream of the pool bottom, which leaves space for the wake vortexes to expand.

The expansion of the wake vortexes together with the relatively low flow velocity and turbulence within the wake vortexes may explain the increased deposition of fine sediment at the step toe at $Q = 49.9$ L/s (Zhang et al., 2020). It is noteworthy that the number and location of wake vortexes remains almost unchanged during this process which is related to the stable architecture of the step structure before the step collapses. This suggests that the step architecture determines the shape and distribution of the wake vortexes at the downstream while pool scour influences the dimensions of these vortexes significantly.

Apart from the pool scour, the development of micro-bedforms in the form of grain clusters which are mainly distributed at the pool bottom and on the negative slope (Fig. 9-10) is another noteworthy morphological variation for the step-pool feature (Zhang et al., 2020). The high spatial resolution outputs of the hybrid model allow us to inspect the interaction between the grain clusters on the surrounding hydraulics in the pool. The grain clusters at the pool bottom mainly appear in the area impinged by the jet (Fig. 10 and Fig. A14) but have very limited disturbance on the surrounding flow field (see details in

Appendix C). The grain clusters on the negative slope where the jet, jump and wake vortexes get mixed (Fig. 4-6) do not show any distribution patterns but increase the flow velocity and turbulence above them significantly (Fig. A15-A16). These grain clusters also have clearer coherent structures in their wake zones than those located at the pool bottom, and these small-scaled coherent structures expand as the pool scour develops (Fig. 8). On balance, the distribution of micro-bedforms at the pool



bottom is affected by the jet regime and the micro-bedforms on the negative slope have strong interference on the surrounding
hydraulics where both the surface jump and wake vortexes fade away.

### 4.4 Insights for resistance, stability, and failure of step-pool features

The distribution of dynamic pressure and shear stress show that the step structure bears the lowest dynamic pressure but highest
shear stress in the step-pool unit, and that the distributions of water forces on the step stones are significantly affected by the
stone sizes and shapes (Fig. 9-10). The magnitude of shear stress on the step is generally two orders of magnitude smaller than
the dynamic pressure at all flow conditions, suggesting that the form drag due to pressure differences is much more prominent
than the skin friction drag acting on the grain surface. The form and skin drag are the basis for the form and grain resistance
in larger spatial scales (Comiti et al., 2009; Zimmermann, 2010). Hence, our results provide support to the finding at reach-
scale that the grain resistance only takes up a small portion of the total resistance (e.g., Aberle and Smart, 2003; Comiti et al.,
2009). Zimmermann (2010) argues that resistance partitioning into grain and form components is difficult for well-structured
beds as the grains protruding into the flow are responsible for some of the form resistance as well as the grain resistance. The
detailed distribution of pressure and shear stress in our results however indirectly quantifies the magnitudes of form and skin
drag on a step-pool unit. Given the difference of orders in magnitude between the pressure and shear stress, the suggestion to
abandon partitioning of resistance in step-pool reach by Zimmermann (2010) is reasonable.

The drag force on the step stones generally increases with flow rate except when flow is increased to 49.9 L/s from 43.6 L/s
(Fig. 11). The step structure collapsed owing to the movement of the KS soon after the flow discharge was further enhanced
from 49.9 L/s in the flume experiment (Fig. 2b, Zhang et al., 2020). This implies that triggers for the movement of the KS
apart from the increase of drag force (Lenzi, 2002; Weichert, 2005) may also exist. The lift coefficient of the step stones shows
a much larger variation range compared to the drag coefficient (Fig. 12), and the magnitude of lift force is also larger than the
drag force generally (Fig. 11). This might partly be the result of the setting that only the protruding part of each step stone was
used in force analysis (Fig. 3d), but also implies that the vertical component of flow force might play an important role in the
mobility of step stones. Considering that the gravity of the step stones does not change, the variation of lift forces will lead to
the variation in the forces on the step stones from contacting coarse grains in bed materials (Zhang et al., 2016) before the step-
pool failure. This sudden variation of the reactive forces might result in subtle changes in the internal structures of the bed
material grains beneath step stones, e.g., configuration of gaps between coarse particles and distribution of fine sediment in
the gaps (Gibson et al., 2011). The internal structure has been found to be closely related to the structural deformation and the
final failure of the step (Zhang et al., 2018). Therefore, we infer that the variation of lift force on step stones and surrounding
grains might also affect step stability and is worthy of further investigation in future research. We also admit that the data for
step-pool failure is very limited in this study and solid conclusions related to failure mechanism can only be reached with
further inspection on the flow forces at more step-pool failures.





**4.5 Limitations of the hybrid modeling**

Although the hybrid modeling shows great advantages in obtaining high-resolution information of the 3D flow properties for a step-pool unit, this approach also has limitations which merit consideration in future research.

(1) The bed surface is set to be impermeable in the model. This setting results mainly in two inconsistencies with reality. First, the hyporheic flow in a step-pool unit has been neglected. Hyporheic flow beneath the step-pool unit has been reported to exit

the bed near the step toe (Hassan et al., 2015), which may affect the wake vortexes to some degree. Second, the upstream sides of step stones beneath the bed surface are also submerged by water owing to high porosity of bed materials (Zhang et al., 2016, 2018) and hence also tolerate water pressure. Without considering this static pressure in our model, the drag force on the entire step stones would be heavily biased, i.e., pointing upstream in most cases. Consequently, only parts of step stones higher than the upstream bed surface of the step was analyzed in the hybrid model (Fig. 3d). When further information of the 3D internal

structure beneath the bed surface is accessible, hyporheic models (e.g., Dudunake et al., 2021) could be added to the hybrid model to resolve this limitation.

(2) No consideration for air entrainment in the jump regime which was observed during the flume experiment (Zhang et al., 2018, 2020) is taken in the hybrid model. Aeration has been reported to affect the flow velocity and turbulence properties at the downstream of a natural step (Vallé and Pasternack, 2006). Neglecting the air entrainment may be the reason for the

mismatch between the simulated results and hydraulic measurements around the jump (Fig. A4-A6) as high air concentration has been found to increase the jump volume (Lenzi et al., 2003). However, no measurement of air concentration in the jet and jump was collected in the flume experiment for us to set parameters for the aeration module which could be coupled to the hybrid model. Also, the limitation of computing capacity obstructs adding an aeration module to the hybrid modelling in our case.

(3) The topographic models of the bed surfaces contain limited areas at the upstream of the step, owing to the measuring difficulty that the frames and beams of the flume in this area restricted the movement of the digital camera. This limitation might result in the underestimation of turbulence development at the upstream of the step-pool model. However, considering that the bed-generated turbulence is greatly suppressed at the upstream of step structure at high flows (e.g., Fig. 6; Wohl and Thompson, 2000), the errors caused by the limited area are acceptable.

(4) No direct measurement of flow forces acting on the step stones are available to directly verify the outcome of flow forces from the hybrid model.

**5 Conclusions**

In this study, we developed a hybrid model, which combines flume measurements and RANS-VOF numerical approach, to resolve the detailed 3D flow characteristics for a step-pool unit made of natural stones. Main findings of this study are as

follows. First, the most prominent feature of hydraulics in the pool is the segmentation of flow regimes at the upstream area





of the negative slope as the jump at water surface in an integral form, streaky wake vortexes near bed, and high-speed jets in between. The transverse configuration of a boulder step significantly affects the flow characteristics at the downstream. Second, the distribution of flow energy and energy dissipation in the pool is highly non-uniform, with the concentration of flow energy transformation and dissipation at the upstream of the negative slope in the pool. Both the surface jump and wake vortexes are

the main energy dissipators for a step-pool unit with well-defined pool configuration. Third, the development of pool scour and flow increase result in the expansion of volume and increase of turbulence energy in the jump and wake vortexes before the surface jump is suppressed at the highest flow. The interference of the micro-bedforms on the surrounding hydraulics is restrained where the wake vortexes and jets dominate in the pool but is enhanced on the negative slope. Finally, the dynamic pressure is generally 1-2 orders larger than the shear stress acting on the step stones and thus the form drag is the overwhelming

component of the drag force on the step. The drag force on the step stones generally increase as the flow rate increases but decreases when the discharge is further increased to the critical value to destabilize the step structure. The lift force on step stones shows a larger magnitude and much wider varying range with an increase in flow compared with the drag force.

The hybrid model, despite its intrinsic limitations (e.g., using impermeable bed surface in the model), has shown great advantages in capturing the fully resolved 3D hydraulic information over merely using flume experiments. The advanced

hydraulic information obtained by the hybrid model helps in achieving a comprehensive understanding to the interaction between hydraulics and morphology and mechanisms of energy dissipation and stability for step-pool features.

**Appendix A: Model verification**

Two methods were taken to verify the hybrid model: (i) the grid independence test; and (ii) comparison between simulated and experimental results.

A series of simulations under the discharge of 43.6 L/s were used to test the grid convergence, with various mesh sizes but identical settings of computational domain (transverse range of Y = -24.5 to 24.5 cm) and boundary conditions. We tested six mesh sizes, i.e., 0.50 cm, 0.375 cm, 0.30 cm, 0.27 cm, 0.25 cm and 0.24 cm, and the corresponding cell numbers of the main mesh block which covered the step-pool unit were 0.89 million, 2.11 million, 4.12 million, 5.61 million, 7.15 million, and 8.08 million. The comparisons of water surface at three longitudinal sections (left boundary, Y = 24.5 cm; middle section, Y = 0.3

cm; right boundary, Y = -24.5 cm) are exhibited in Fig. A1 and the distributions of flow velocity at the middle section with the variation of mesh size are shown in Fig. A2. The variations of both the water surface and flow velocity distribution become insignificant after the mesh size is reduced to below 0.3 cm, though fluctuations exist around the contraction section (Fig. 2e). This result illustrates that the grid size of 0.25 cm which was finally chosen for all the simulations in this study satisfies the requirement of grid independence.






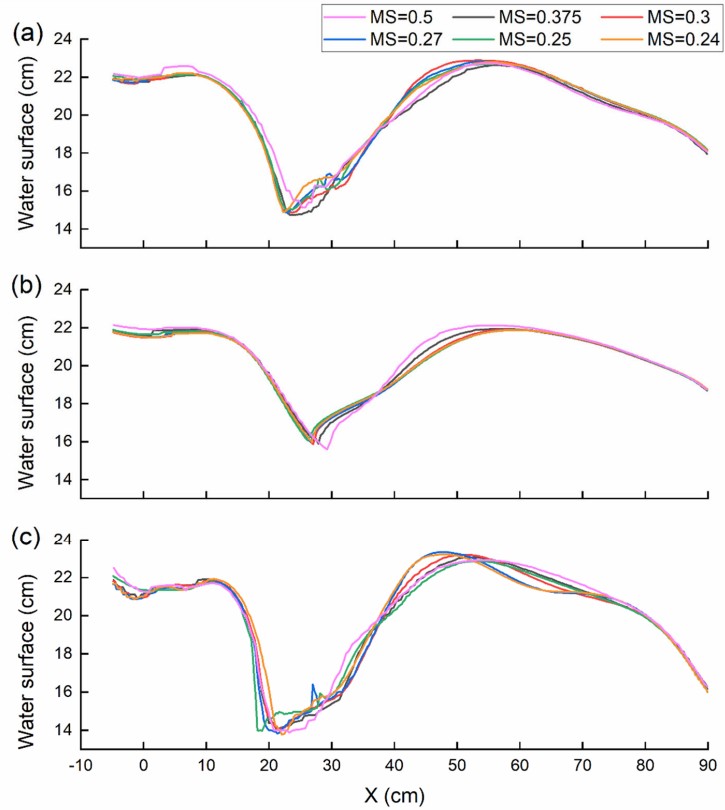

**Figure A1: Water surface profiles of the simulations with different mesh sizes at the discharge of 43.6 L/s at the longitudinal sections at: (a) Y = 24.5 cm (left boundary); (b) Y = 0.3 cm (middle section); (c) Y = -24.5 cm (right boundary). MS is short for mesh size. The flow direction is from left to right in each plot.**

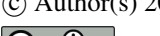

Earth **Surface**
**Dynamics**
Discussions

EGU

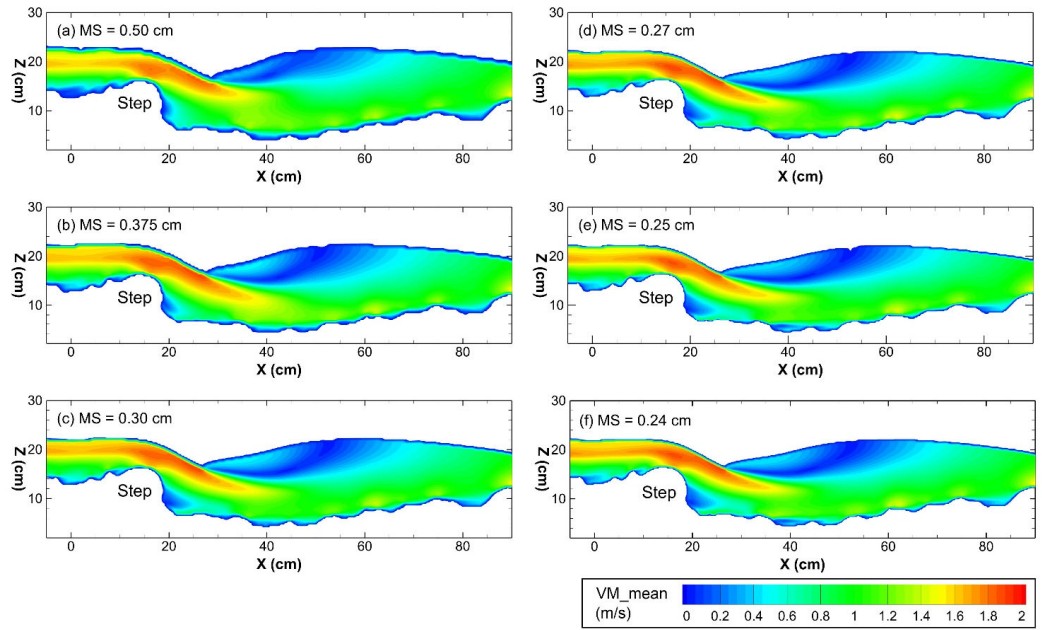

**Figure A2: Contours of velocity magnitude in the longitudinal section at Y = 0 cm at different mesh sizes (MSs) under the flow condition with the discharge of 43.6 L/s: (a) 0.50 cm; (b) 0.375 cm; (c) 0.30 cm; (d) 0.27 cm; (e) 0.25 cm; (f) 0.24 cm. The flow direction is from left to right.**

Two measurements in the previous flume experiments (Zhang et al., 2018, 2020) were used to validate the numerical models: (i) longitudinal water surface profiles extracted from the side cameras (Fig. A3a); and (ii) water surface regime recorded in pictures by the top camera (Fig. A3b).





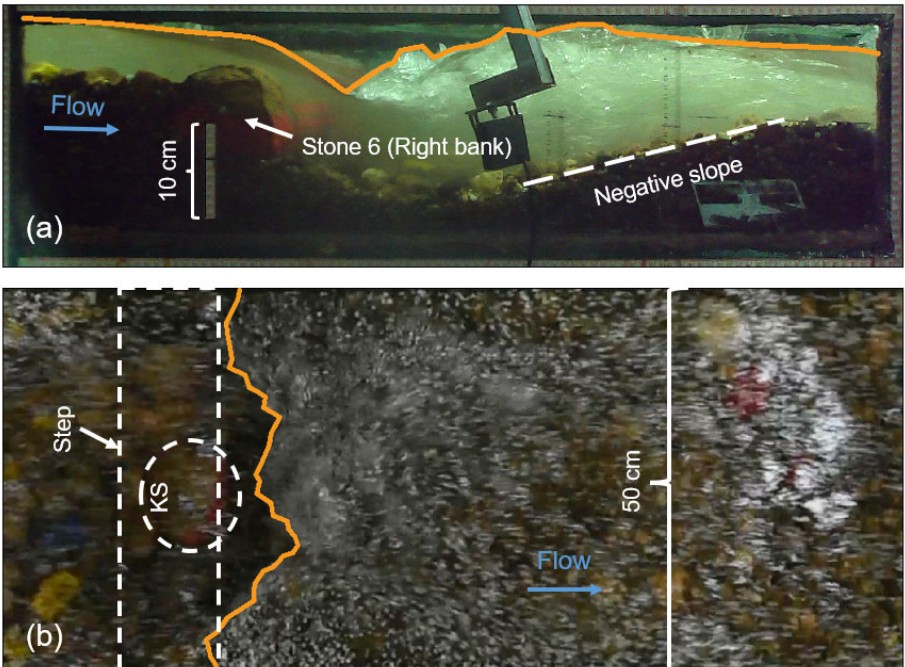

**Figure A3: Measurements of water surfaces (orange lines) used in model verification: (a) water surface profiles from both sides of the flume; (b) upstream edge of the jump regime from top view. KS refers to keystone in figure (b).**


Figures A4-A6 demonstrate the comparisons of water surface between the experiment measurements and numerical modelling results at the flow rate of 32.1, 43.6, and 49.9 L/s at both sides of the flume. The comparisons illustrate that the simulated water surface profiles are generally comparable with the experimental measurements, even at the highest flow condition tested in the experiment with fluctuating water surface. The simulated water surfaces at the upstream and downstream of the hydraulic

jump in the pool match well with the measurements. However, clear deviations of the simulations from the measured water surfaces appear at the hydraulic jump regimes where intense air entrainment occurs. The air entrainment was not considered in the CFD model in order to reduce model complexity and the requirement for computation resources. This simplification might neglect the volume expansion of the fluids at the hydraulic jump and hence, underestimate the elevation of free water surface.

Earth **Surface**
**Dynamics**
Discussions

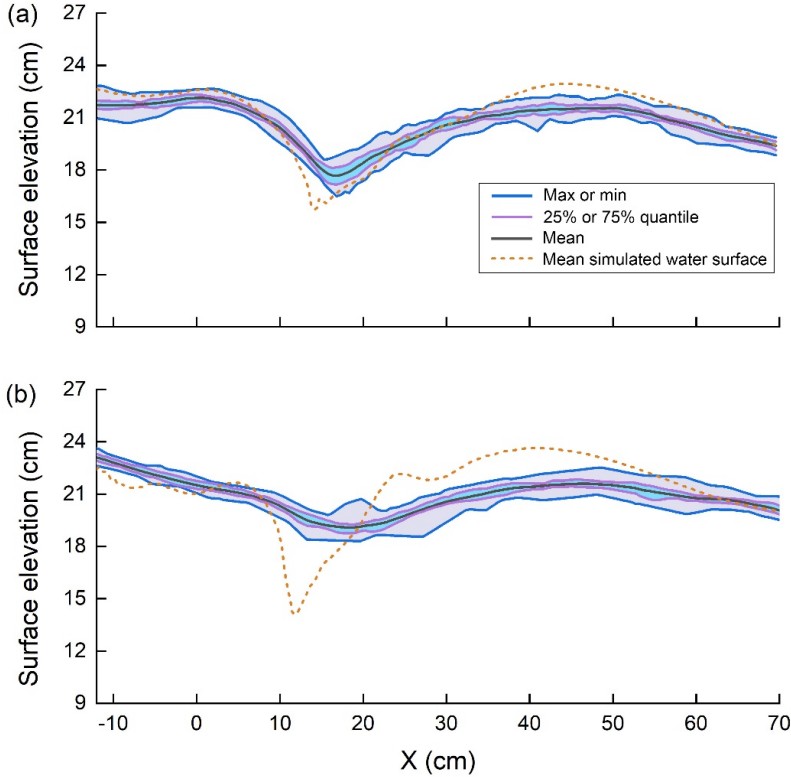

**Figure A4: Comparison of water surface between the measurement and simulation under the discharge of 32.1 L/s at (a) left side, and (b) right side of the flume. The max, 75% quantile, mean, 25% quantile and min of the measured water surfaces are presented in solid lines. The flow goes from left to right in each plot.**




Earth **Surface**
**Dynamics**
Discussions

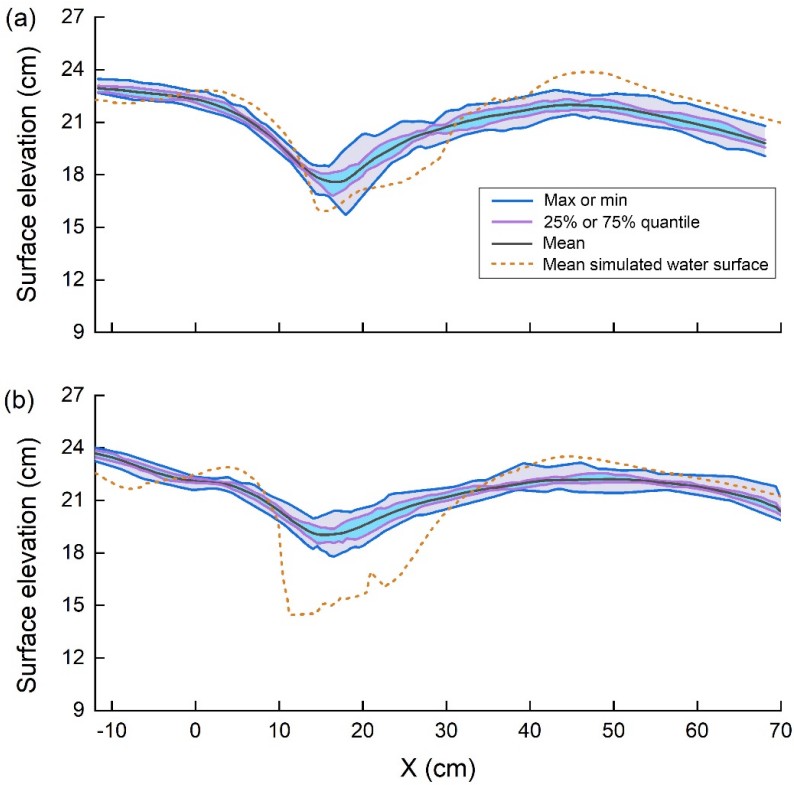

Figure A5: Comparison of water surface between the measurement and simulation under the discharge of 43.6 L/s at (a) left side, and (b) right side of the flume. The max, 75% quantile, mean, 25% quantile and min of the measured water surfaces are presented in solid lines. The flow goes from left to right in each plot.

Earth **Surface**
**Dynamics** Open Access
Discussions

EGU

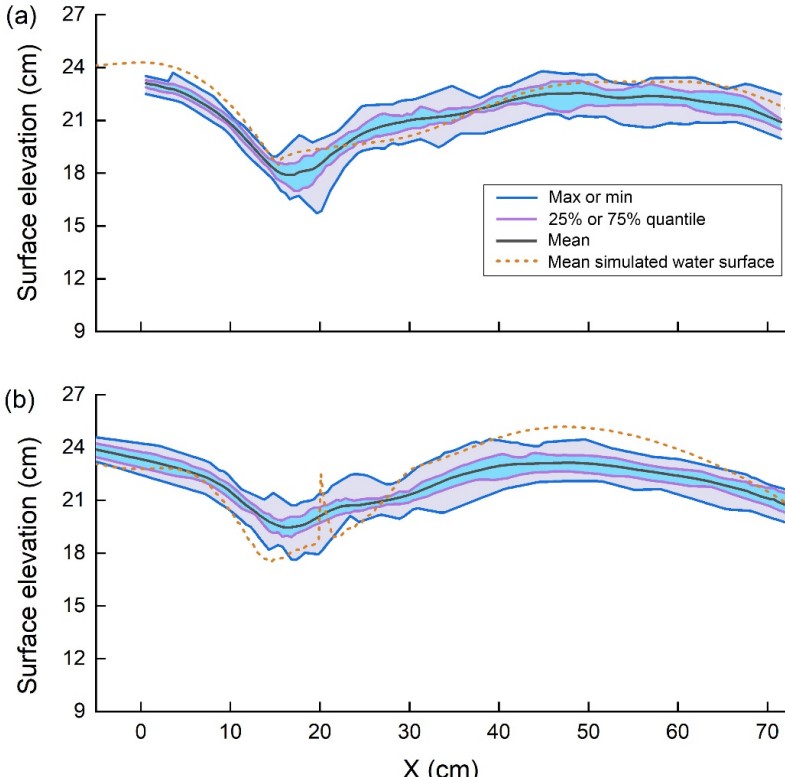

**Figure A6: Comparison of water surface between the measurement and simulation under the discharge of 49.9 L/s at (a) left side, and (b) right side of the flume. The max, 75% quantile, mean, 25% quantile and min of the measured water surfaces are presented in solid lines. The flow goes from left to right in each plot.**


Figure A7 and Table A1 exhibit the validation of the boundary which separates the jet and the jump at water surface from the topview. All the boundaries were extracted manually for the experimental and numerical results, based on the distinct contrast

of flow velocity in the two flow regimes (Fig. A3b). The simulated boundary generally locates in the range of measured boundaries (Fig. A7) and the deviations of the simulation under all the tested discharges are acceptable (Table A1). These results further verify the feasibility of our hybrid model to simulate the complex surface flow regimes over a step-pool unit. Both the comparisons of water surface from sideview and topview show that the hybrid model succeeded in capturing the flow characteristics for a step-pool feature built in the physical experiment.



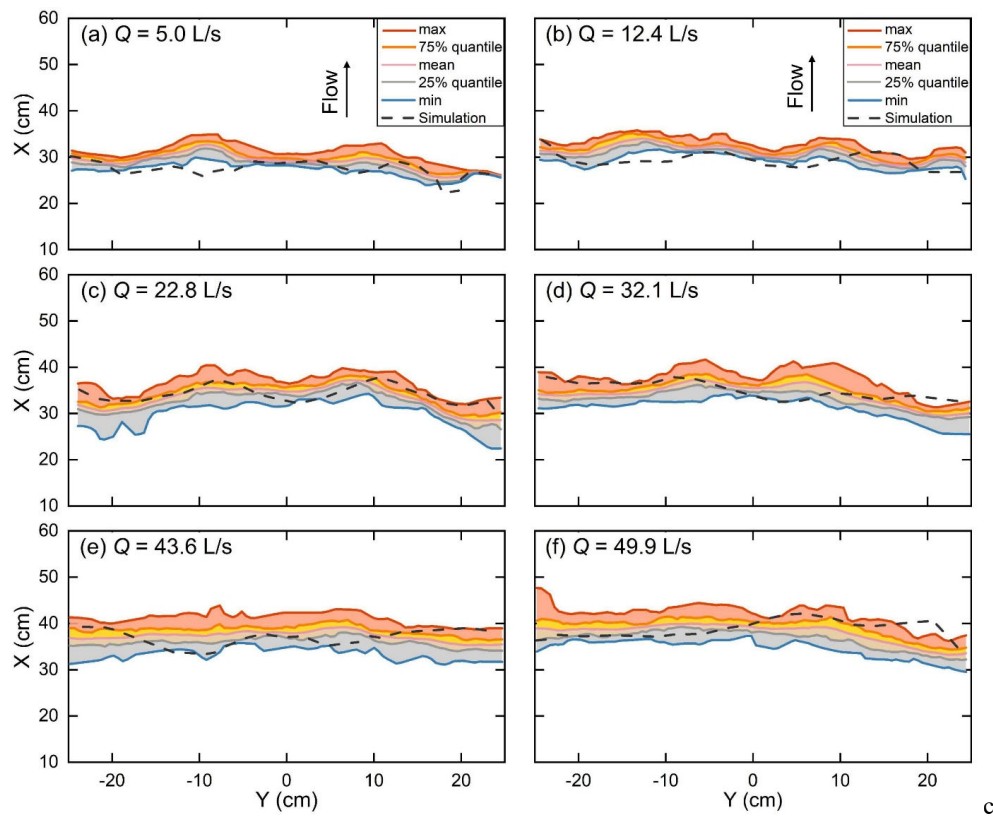

c

**Figure A7: The extracted boundaries between the jump and jet at water surface from simulated results (in dots) and experimental measurements at all the tested discharges. The max, 75% quantile, mean, 25% quantile and min X values of the measured boundaries are presented in solid lines while the mean simulated boundaries are plotted in dashed lines.**

**Table A1: Error indices of the simulated upstream edges of jump regimes from the top view**

| Q (L/s) | ME (cm) | MAE (cm) | MSE (cm) | RMSE (cm) | SDE (cm) |
|---|---|---|---|---|---|
| 5 | 1.54 | 1.71 | 5.71 | 2.39 | 0.99 |
| 12.4 | 1.82 | 2.40 | 7.16 | 2.68 | 0.73 |
| 22.8 | -0.76 | 1.75 | 3.90 | 1.97 | 1.66 |
| 32.1 | -0.71 | 2.02 | 5.44 | 2.33 | 2.11 |
| 43.6 | 0.46 | 2.21 | 6.28 | 2.51 | 2.42 |
| 49.9 | -0.92 | 2.45 | 8.13 | 2.85 | 2.54 |




## Appendix B: Supplemental figures for flow properties and forces

Figure A8 presents the longitudinal distribution of Froude number in section Y=0. Figures A9-A14 provide supplementary information of flow properties and flow forces at the discharges of 5.0, 12.4, 22.8 and 32.1 L/s.

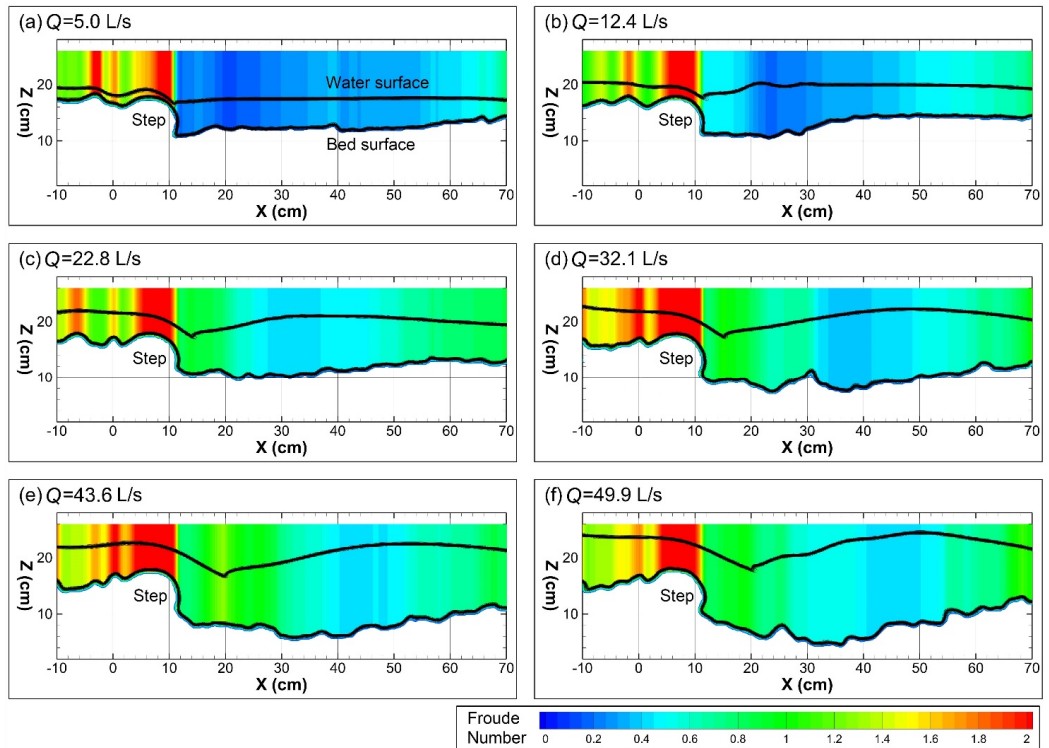

**Figure A8: Distribution of time-averaged Froude number in the longitudinal section Y = 0 cm for all flow rates. *Q* refers to the discharge at the inlet of the computational domain.**

Earth **Surface**
**Dynamics**
Discussions

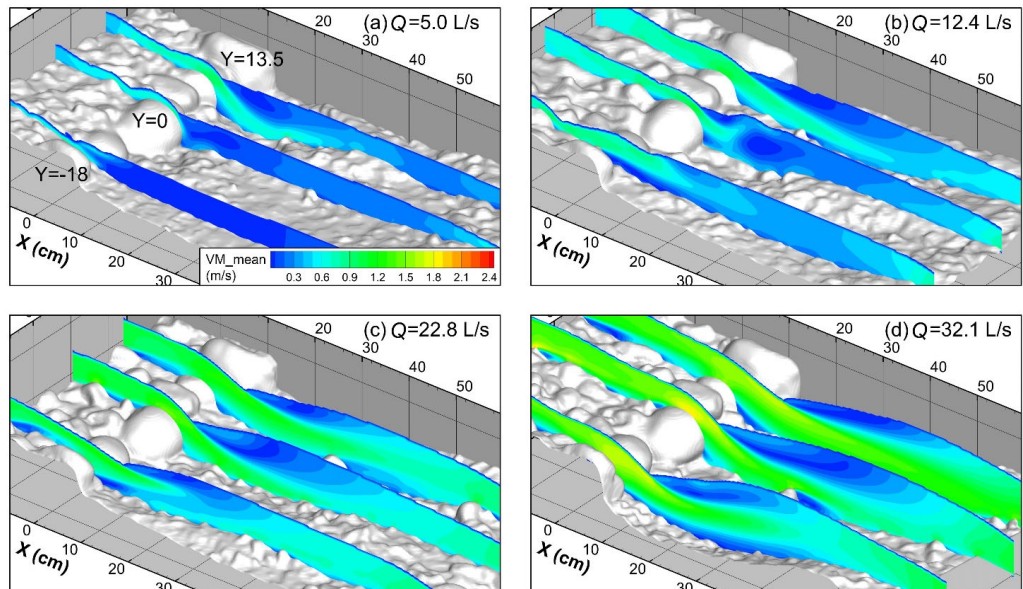

**Figure A9: Distribution of time-averaged velocity magnitude (VM_mean) in three longitudinal sections (Y = -18, 0 and 13.5 cm, marked in figure (a)). $Q$ refers to the discharge at the inlet of the computational domain. The spacings for X, Y, and Z axes are all 660 10 cm in the plots.**





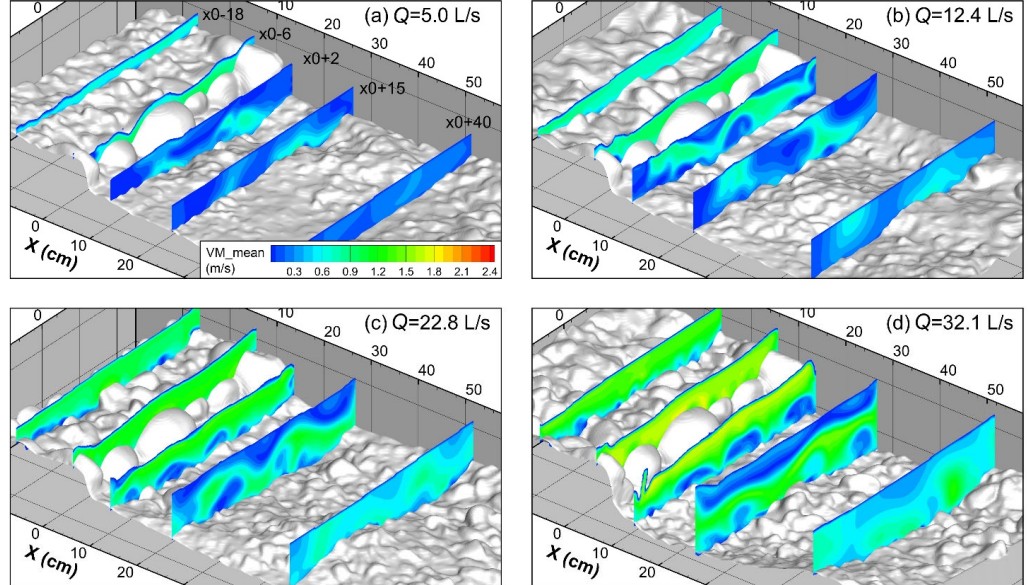

**Figure A10: Distribution of time-averaged flow velocity at five cross sections relative to the reference cross section x0. The reference cross section x0 is located at the downstream end of the keystone (KS). The five sections are marked in figure (a). _Q_ refers to the discharge at the inlet of the computational domain. The spacings for X, Y, and Z axes are all 10 cm in the plots.**



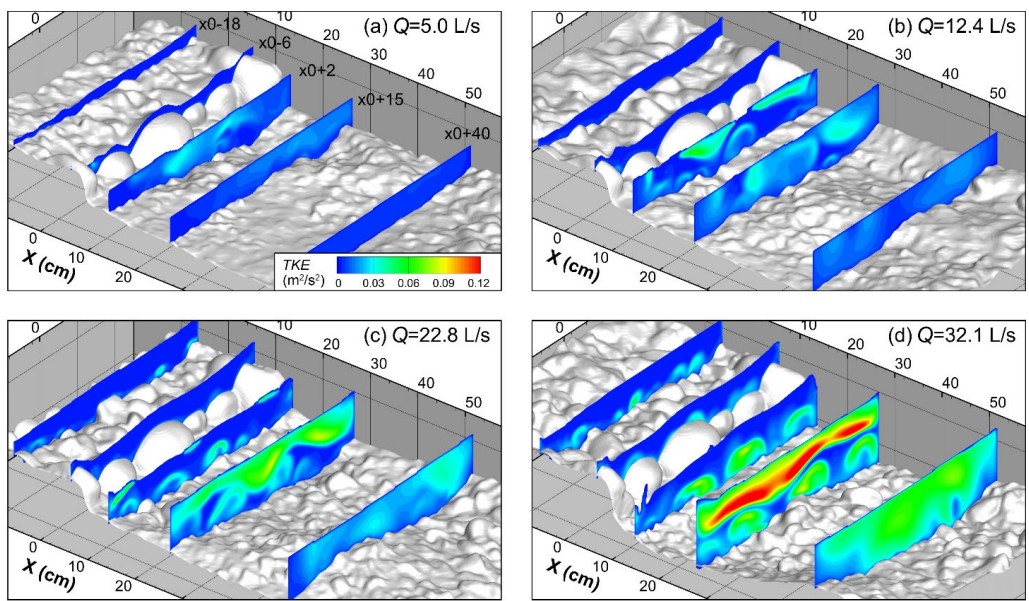


**Figure A11: Distribution of the time-averaged turbulence kinetic energy (*TKE*) in the five cross sections same with Fig. A10. *Q* refers to the discharge at the inlet of the computational domain. The spacings for X, Y, and Z axes are all 10 cm in the plots.**





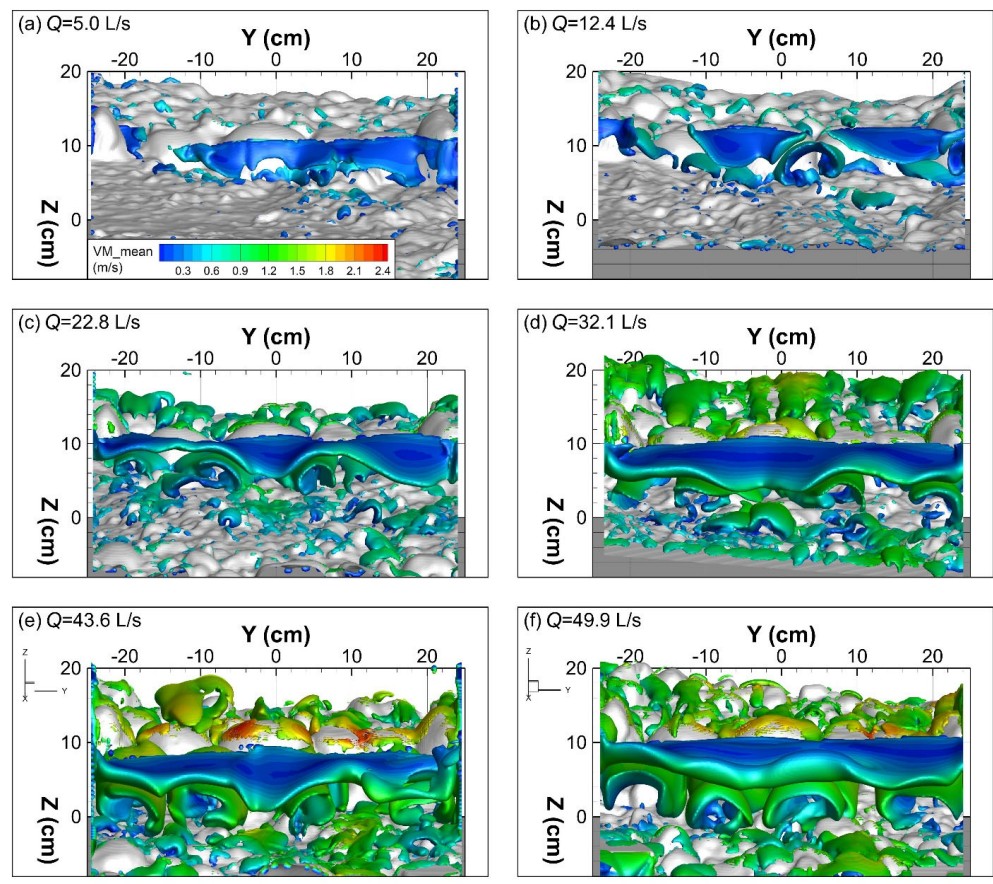

**Figure A12: Instantaneous flow structures extracted using the *Q*-criterion (*Q*<sub>criterion</sub>=1200) and colored by the magnitude of flow velocity. This figure plots the same coherent structures with Fig. 7 but in a different view.**



Earth **Surface**
Dynamics
Discussions

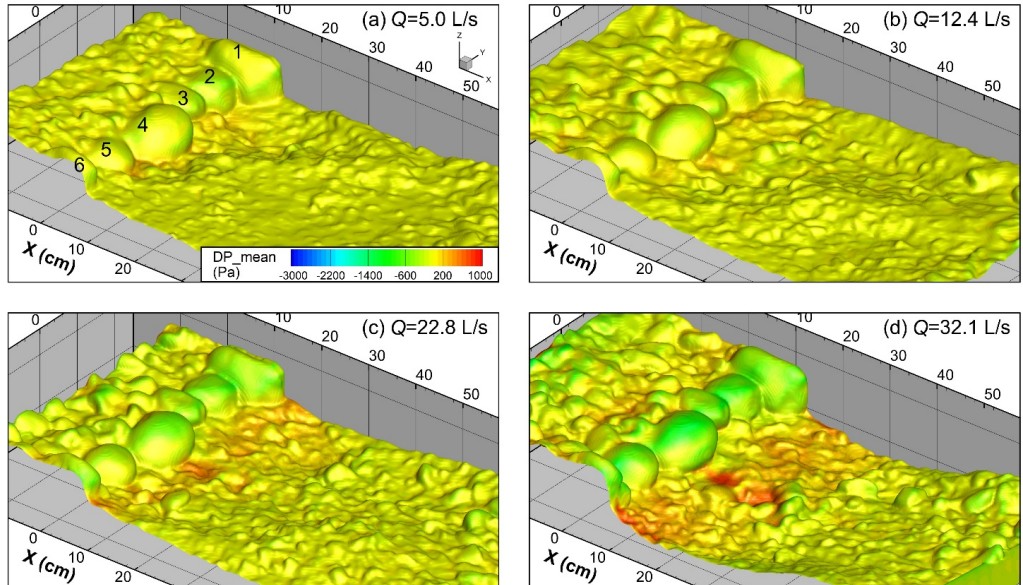

**Figure A13: Distributions of time-averaged dynamic pressure (DP_mean) on the bed surface of the step-pool unit under four flow rates. The numbers of step stones are marked in all the plots. The negative values in the plots result from the setting of standard atmospheric pressure = 0 Pa, whose absolute value is $1.013 \times 10^5$ Pa.**



Earth **Surface**
**Dynamics**
Discussions


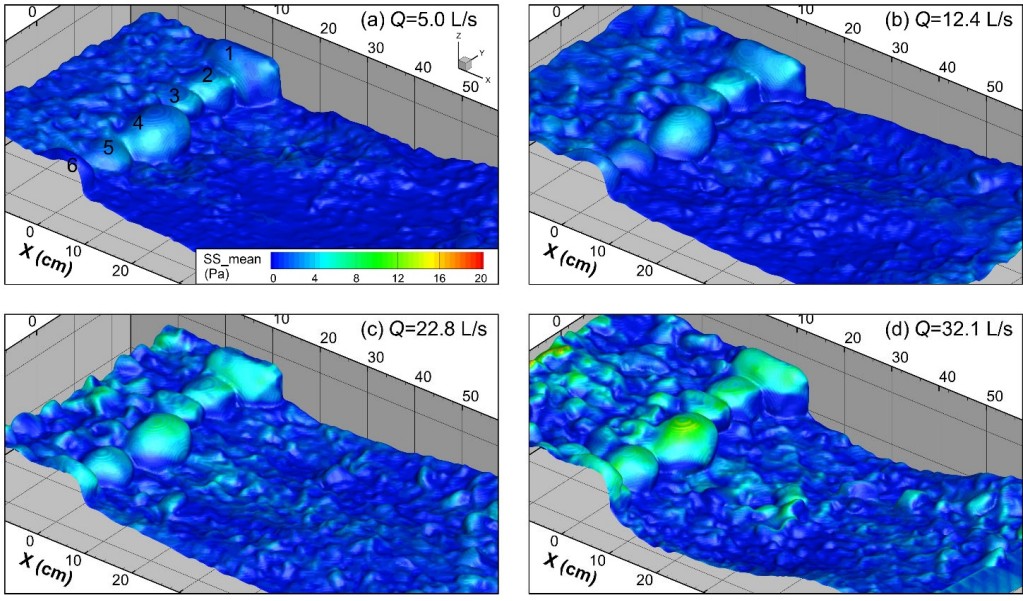

**Figure A14: Distributions of time-averaged shear stress (SS_mean) on the bed surface of the step-pool unit under four flow rates. The numbers of step stones are marked in all the plots. The standard atmospheric pressure is set as 0 Pa.**

**Appendix C: Influence of micro-bedforms in the pool on surrounding hydraulics**

To illustrate the effect of the micro-bedforms as grain clusters on the surrounding hydraulics, we take the scenario at $Q = 49.9$

L/s as an instance, shown in Fig. A15-16. The four wake vortexes show intact configurations in the cross section at x0+8, which locates at the upstream of all the micro-bedforms in the pool and hence, is used as a reference section. When a protruding grain/cluster is located within a vortex, it has almost no disturbance on the flow field or *TKE* nearby (e.g., G1 and G3 in Fig. A15c to d and Fig. A16b to c). In contrast, if a cluster is located in the gap between two vortexes (e.g., G2 and G4 in Fig. A15c to d and Fig. A16b to c), both the flow velocity and *TKE* increase near the cluster but the increase is limited in a thin layer

(with thickness < 1 cm) above the grain surface. The wake vortexes nearby show almost no deformation. These results suggest that the grain clusters have very limited influence on the surrounding hydraulics at the pool bottom, where the alternation of jets and wake vortexes dominates the flow structures near the bed surface. The interference of the grain clusters at the pool bottom on local hydraulics keeps being suppressed during the development of pool scour. In contrast, the grain clusters on the negative slope increase the flow velocity and turbulence above them and the area affected is largely expanded compared with

those at the pool bottom (Fig. A15e and A16d).

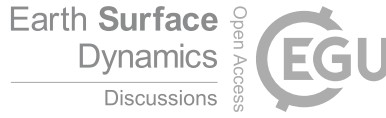

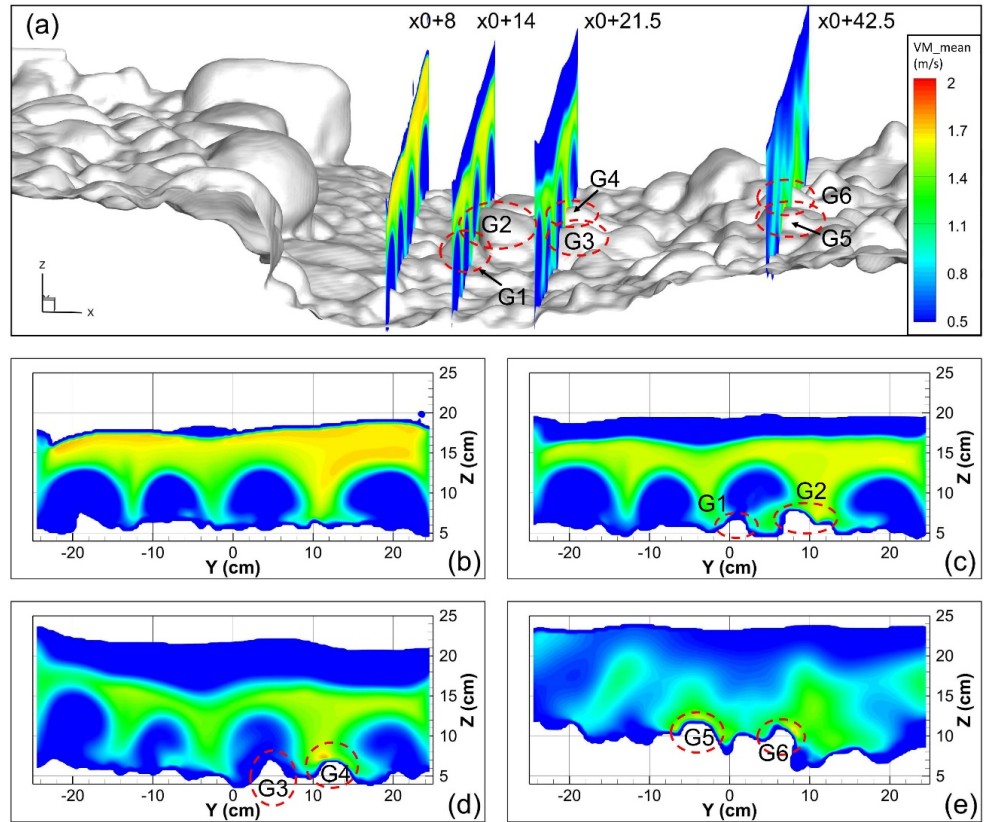

**Figure A15.** Figure (a) shows the locations of the cross sections and target coarse grains at $Q$ = 49.9 L/s. Figures (b) to (e) show the distribution of velocity magnitude (VM_mean) in the four chosen cross sections: (a) x0+8.0; (b) x0+14.0; (c) x0+21.5; (d) x0+42.5. G1 to G6 refer to 6 protruding grains in the micro-bedforms in the pool.

Earth **Surface**
**Dynamics**
Discussions

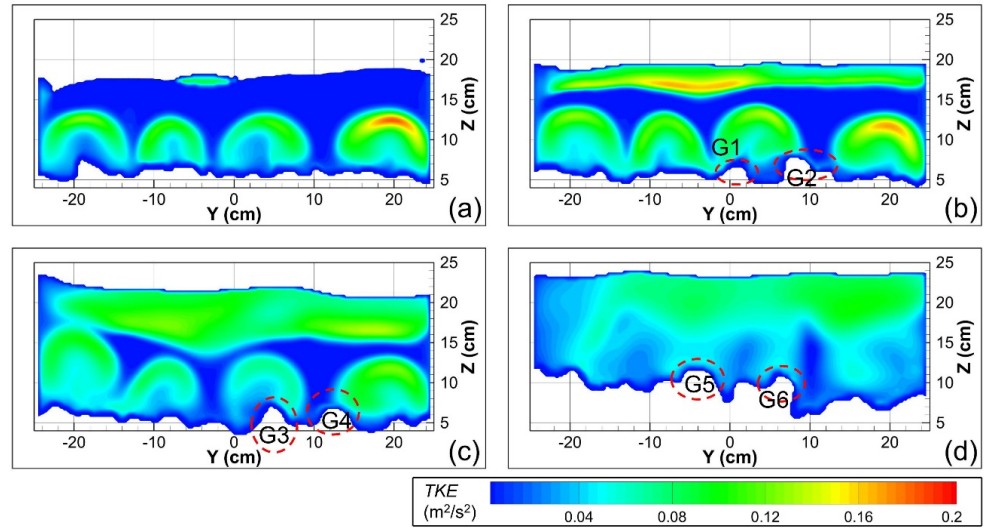


**Figure A16. The distribution of turbulent kinetic energy (*TKE*) in the same cross sections as in figure S15: (a) x0+8.0; (b) x0+14.0; (c) x0+21.5; (d) x0+42.5.**

### Data availability

Topographic models of the step-pool unit recognized in the CFD models and key settings of the CFD models can be found at
https://doi.org/10.5281/zenodo.5840753 (Zhang, 2021).

### Author contributions

CZ conceptualized and designed the research, processed the measurements of flume experiments, performed the numerical simulations, analyzed the data, wrote the manuscript, prepared the figures, and contributed to funding acquisition. YX contributed significantly to the conceptualization of the work, provided key advice in the numerical simulations, and reviewed
several versions of the manuscript. MAH contributed to the design of the research, reviewed and edited several versions of the manuscript, and contributed significantly to the interpretation and contextualization of the results. MX contributed greatly to funding acquisition and arrangement of resources, provided advice in research design and reviewed the manuscript. PH contributed in performing numerical simulations, model verification, analyzing the data, and preparing the figures and manuscript.



**Competing interests**

The authors declare that they have no conflict of interest.

**Acknowledgements**

Yingjun Liu from Tsinghua University is kindly acknowledged for his assistance in processing topographic models of bed surface and establishing the early versions of the CFD models.

**Financial support**

This study is supported by the National Natural Science Foundation of China (No. 51779120, 52009062, 41790434), Chinese Academy of Sciences (XDA23090401), China Postdoctoral Science Foundation (2018M641369), and Open Research Fund Program of State key Laboratory of Hydroscience and Engineering, Tsinghua University (sklhse-2021-B-03).

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
