# Peer review of "A combined approach of experimental and numerical modelling on 3D hydraulic features of a step-pool unit"

_Earth Surface Dynamics, 2022_

## Referee Comment (RC2)

**Review of Zhang et al. "Hybrid Modeling on 3D hydraulic features of a step-pool unit."**

**Keith Richardson**

This paper presents results from a novel application of CFD modelling to a step-pool unit. This study is innovative and the results greatly enhance our understanding of the detailed 3D hydraulic characteristics of step-pools, which present a challenging environment for direct measurement. The results will also advance our understanding of the role of hydraulics in the formation and stability of step-pools and the effect of step-pools on flow resistance. The identification of two discrete hydraulic structures that function as energy dissipators in a step-pool unit, and the discussion of the energy dissipation of 2D compared with 3D hydraulic structures, are of particular interest. I would like to congratulate the authors on this study.

However, I have two main concerns about the way the results are explained, described and interpreted. On one level, these concerns are not particularly serious and can be relatively easily addressed in the paper to make it worthy of publication, but on a more fundamental level they show that there is some misunderstanding of the main underlying topics of the study.

Firstly, in the description of the results, there is misunderstanding of the coherent and persistent hydraulic structures in the step-pool, specifically in relation to the terms "wake", "hydraulic jump", "jet" and "regime". The authors in effect are using their own working versions of these terms. I shall discuss the use of each of these terms in turn.

The region at the toe of the step is referred to as a "wake" or "wake zone" (lines 258 and 261) and the flow recirculation cell at the toe of the step is referred to throughout the paper as a "wake vortex". The term "wake" usually refers to the region immediately downstream of an object or roughness element in flow whose dimensions are much larger than those of the object or roughness element. The term "wake" cannot be applied to flow immediately downstream of a channel feature whose dimensions are similar to those of the flow, and that acts as channel topography rather than roughness. The so-called "wake" is merely the region at the toe of the step and the "wake vortex" is a transverse flow recirculation cell attached to the toe of the step.

There is indeed a hydraulic jump in the pool, but the hydraulic jump (or more usually, "jump") is repeatedly referred to as a "surface jump" and described as existing only at the surface. A hydraulic jump is a feature that occupies the entire water column; the surface feature referred to by the authors of this paper as a "jump" is in fact the flow recirculation cell of the hydraulic jump, in which flow spills backwards down the adverse surface slope generated as flow decelerates to a subcritical condition from a supercritical one.

The authors refer to a "jet" or "jets" that exist(s) above the "wake vortex" and below the "jump" that is a feature separate to the "wake vortex" and "jump". For example, lines 435-436 talk about the "segmentation" of these features. However, the impinging jet of the step and its associated recirculation cell, and the flow within the hydraulic jump below the recirculation cell of the jump, are unified and single features rather than distinct and separate features. This is especially the case with the hydraulic jump, which as mentioned previously, extends through the entire water column; the so-called "jet" is merely the fastest downstream-directed flow within the hydraulic jump.

All three features described above ("wake"/"wake vortex", "jump" and "jet") are referred to as "regimes" by the authors, when "structures", "features" or "regions" would be more appropriate terms. A regime is a region of the flow that extends across the full width (or most of the width) and depth of the flow, and over a longitudinal distance usually several times the flow width. For

example, in terms of the Froude domain, the flow regime can be described as supercritical, subcritical, or transcritical. In this respect, the flow regime in the authors' experiment and simulation is transcritical, and this describes the flow over the step-pool unit, or a number of step-pool units.

In general, the identification and description of the hydraulic coherent structures need correcting and clarifying, including the locations of these features. There are further examples in the detailed comments below.

Secondly, the discussion of shear stress, dynamic pressure, form drag, skin friction, form and grain flow resistance and flow resistance partitioning at 514-523 is both confused and confusing, and illustrates an almost total lack of understanding of these subjects. I don't know where to begin suggesting editing this section, and I can only describe this section as nonsense. In any case, this section is oxymoronic. It partitions form and grain resistance before stating that attempting such partitioning is invalid. I suggest deleting it. It is possible that conclusions regarding the relative magnitude of grain resistance and form resistance at the reach or step-pool unit scale and regarding flow resistance partitioning could be drawn from the authors' results, but it would require significant analysis further to that presented in this paper. Specifically, it would require integrating the boundary shear stress over the step-pool unit and comparing it with the total shear stress due to the downstream component of the weight of water in the step-pool unit, and making any necessary adjustments for flow non-uniformity between the inlet and outlet sections.

Detailed comments are given below, with relevant line numbers.

Line 1 replace "on" with "of" in title.

Line 16 and throughout paper. Plural of vortex is vortices.

Lines 28-29 Abrahams et al. is not an appropriate reference here. Abrahams et al. is about the relationship between step-pool geometry and flow resistance. It does not directly address the effect of step-pools on channel stability.

Lines 32-33 insert "and" between "hydraulics" and "stability". Replace "these dimensions" with "them".

Line 42-45 replace "different from" with "unlike". Replace "which result in" with "resulting from". Replace "oscillation" with "alternation". Also relevant here is the fact that formative flows of step-pools are very high discharge, typically c. 50 yr return interval, making them impractical to measure. Delete "Salt or rhodamine dilution and".

Line 56 replace "at" with "to".

Lines 85-87 and 95-96 are repetitive of lines 77-80.

Section 2.1 There are missing experimental details. What were the stepstone dimensions, initial and final step height, and step spacing?

Lines 102-103 why was Froude scaling employed as opposed to some other form of scaling? Without giving justification, any scaling method used is arbitrary.

Lines 105-106 "We did not manually build any pool features…" What does this mean, and what was the initial constructed longitudinal profile?

Line 111 What are "T runs"? Delete "designed". Replace "step by step" with "stepwise".

Line 114 What does "discharge change interval" mean?

Line 123 Replace "step-pool-step" with "step-pool".

Line 125 Replace "upstream area of" with "area upstream of". Replace "step model" with "step-pool model".

Line 127 The acronym "KS" has not been used before and needs to be explained.

Line 140 Replace "solution" with "software". Full stop after "platform". Start new sentence and replace "which" with "This software".

Lines 152-156 What effect do these added topography components have on the reliability of the CFD results?

Line 191 This is a relatively low sampling period and frequency. How were the sampling period and frequency selected? High frequency turbulent fluctuations will be missed and will not contribute to shear stress, dynamic pressure or TKE. What effect will this have on the results?

Lines 200-201 Unit of measurement missing: 3 cm? How do these RMSE values compare to key length scales such as Step stone size, step height, step spacing etc.?

Lines 222-225 This is verbose, and the separation of total pressure into dynamic and hydrostatic components is trivial and it is not necessary to describe it.

2.4 Data Processing: there are missing calculations: how were bed shear stress and the forces acting on step stones calculated?

Line 255 delete "as the main flow".

Line 258 replace "deviated" with "separated". I suggest better and further explanation here. It is true that the jet at Y =-18 cm does not impinge on the bed but the jets at all three sections are separated from the bed by a vortex (flow recirculation cell); the jet at Y = -18 cm does not impinge on the bed because the vortex here extends further downstream than that at the other two sections and then merges with the jet.

Line 260 replace "feature" with "features". Replace "limitation" with "reduction" or "contraction" or similar.

Line 269 and throughout paper. Replace "at the upstream area of" and "at the upstream of" with "upstream of"

Lines 271-272 "point of separation of the jet from the step face" would be better than "detaching point". The point of separation of the jet and the contraction section need to be explained and described because it's not clear where they are. "flow concentrates" is also confusing and inaccurate. I suggest using the phrase "high velocity regions".

Lines 272, 274 and 276 "lower top elevations" and "higher top elevations" are confusing phrases. I suggest "low/high points within the step crest".

Lines 273-275 "discrete vortexes near the bed surface" and "the gaps between the wake vortexes near the bed are filled with high speed flows"; "3D flow separation cells at the toe of the step with transverse axes separated in the transverse direction by regions of high speed flow" would be better. They expand in the longitudinal direction only with an increase in discharge. Also, the high

speed regions are centred on the contact points of the step stones and the flow separations cells are centred on the centres of the step stones, which is the reverse of your description.

Line 275 "contact points" would be better than "connecting points".

Line 277-278 replace "pool bottom" with "base of the pool". Replace "shrink" with "are less pronounced". The jet is referred to here but was not mentioned in the description of the section at X0+15; mention in description of both sections to avoid confusion.

Line 278 As mentioned in the general comments above, the hydraulic jump is not a regime, it extends throughout the water column and it has flow velocities close to zero in its flow recirculation cell close to the surface.

Lines 281-282 There is no such thing as a surface jump.

Line 283 What does "the drop of flow velocity can be found in sections X0-6 and X0+2" mean?

Line 294 TKE is not turbulence intensity. The latter is dimensionless and is normally estimated as the RMS of turbulent velocity fluctuations normalised by the mean longitudinal velocity. Replace "overlaps" with "coincides".

Lines 294-295 "high flow velocities in the upstream area of the step limits the development of turbulence". This conclusion cannot be drawn because correlation does not indicate causation, but moreover, high mean flow velocity generally coincides with low TKE, and vice versa, because where TKE is high, it has been extracted from the energy of the main flow. The two flow properties are two sides of the same coin and it is incorrect to say one causes the other.

Lines 297 and 299, etc. as mentioned in the general comments, this region is not the wake of the step stones.

Lines 299-300 etc. as mentioned in the general comments, the jump is neither at the surface nor above the jet.

Line 302 Turbulent energy dissipation is not synonymous with TKE, so do not alternate between the two phrases. The highest TKE occurs near the interfaces with the jet because this is a region of high fluid shear.

Line 304 why is there a decrease in flow velocity with an increase in discharge? Replace "lead" with "leads" and "limitation" with "reduction".

Line 311 delete "with a length of 50 cm".

Line 314 replace "as the regime of jet" with "due to the presence of the jet".

Line 316 replace "where the" with "of" and delete "shows up" and "to the".

Line 319 Insert "pool" between "negative" and "slope".

Line 332 replace "vortex" with "turbulent".

Lines 335-336 the "vortex stretched across the entire channel width near the surface is the flow recirculation cell of the hydraulic jump, and the "discrete streamwise streaky vortexes close to the bed" are flow recirculation cells attached to the toe of the step formed by the separation of the jet from the step face.

Line 340 The thickness of the hydraulic jump is the flow depth.

Line 341 delete "in the pool". The "jump regime" is the flow recirculation cell of the hydraulic jump. Replace "fades away" with "loses identity".

Lines 343-345 I suggest for final sentence "On the negative slope, coherent structures mainly follow protruding grains (micro-scale bed structures but do not show streaky features as they do upstream of the step, even though the grain sizes are similar."

Line 352 "point of flow separation of the jet from the step face" is better. However, the separation of the jet from the step face has not been properly described and needs a detailed description in section 3.1.1.

Lines 353 and 372 Replace "connection" with "contact".

Line 355 Insert "of the jet" between "Impinging point" and "in the pool".

Line 356 replace "at the locations more downstream" with "at locations further downstream".

Lines 357-359 Explain

Line 370 replace "shear stress is further concentrated" with "The highest values of shear stress occur".

Line 371 replace "top elevation" with "maximum height" and "influence" with "influences".

Line 374 Replace "configuration" with "shape".

Lines 376-377 insert "on" between "and" and "some". Delete "comparing with surrounding grains".

Line 384 replace "keeps increasing" with "increases".

Line 385 replace "enhanced" with "increased".

Line 388 replace "turns the" with "changes". How can the lift force be downwards? This does not make sense.

Line 390 replace "turns" with "changes direction to".

Line 391 insert "of magnitude" between "orders" and "smaller".

Lines 391-395 Only lift and drag forces were mentioned in the methods and they were not given as x and z components. In other words, this x, y and z coordinate system for forces has not been mentioned before. The "bank stones" have also not been mentioned before. Insert "of any component" after "the y component of flow force has the largest magnitude". Why is the transverse force component greater at the banks? Replace "enhanced" with "increased". Replace "concentrates" with "is greatest".

Line 401 How and why do $C_D$ and $C_L$ vary, and what does this mean?

Line 404 What does "keystone" mean here and why is stone 4 the keystone? Explain in methods.

Line 416 Replace "distinguished" with "the well developed".

Line 419 replace "transverse inconsistencies of" with "a 3D".

Line 421-423. This is incorrect. The vortexes (flow recirculation cells) at the step toe occur downstream of the centres of the step stone and the high speed regions between the vortices occur

downstream of the contacts between step stones. Also, whether the jet impinges on the bed or not is more related to momentum than kinetic energy.

Line 428 The jet that does not impinge on the bed at the base of the pool is not a surface jet. It still has a flow recirculation cell above it and is a classic jump. Surface jets are associated with oscillatory jumps (standing waves).

Lines 429-432 Delete.

Lines 435-438 This is incorrect. Delete up to and including "wakes vortex (Fig. 4)".

Lines 438-439 "intense mid-profile fluid shearing within the hydraulic jump and between the flow recirculation cells at the step toe and the jet plunging over the step face generates high TKE" is better.

Lines 448-449 Why is this noteworthy?

Figure 13. The symbols are illegible.

Lines 463-465 This does not makes sense. Also, insert "at 49.9 l/s" between "examined" and "(Fig. 13)".

Line 467 Insert "It is" before "Worth noting".

Line 474 insert "over a 2D step" between "vertical drop" and "with the".

Line 486 Replace "turbulence with "TKE". Replace "visualized" with "demonstrated".

Line 487 Insert "up to a discharge of 43.6 l/s" after "flow increased".

Line 488 The flow recirculation cell of the hydraulic jump expanded.

Line 492 Replace "variation patterns" with "response".

Line 493 Replace "result" with "results". The change in jet penetration angle was not mentioned in the results.

Line 494 replace "leaves" with "creates".

Line 495 This contradicts lines 296-297 which state that the "wake" vortices have high TKE.

Line 499 More accurately, the jet angle and momentum influence both vortex dimensions and pool scour.

Line 500 replace "distributed" with "located".

Line 503 replace "on the surrounding hydraulics in the pool" with "and pool hydraulics".

Line 505 replace "get mixed" with "lose their identity".

Line 508 Replace "on balance" with "in summary".

Line 512 Replace "distribution" with "distributions".

Lines 531 Delete "Considering that the gravity of the step stones does not change".

Line 547 replace "tolerate" with "experience". Is it static pressure? I would have thought dynamic water pressure would be transmitted through the pores to the bases of the stones.

Line 571 what does "in an integral form" mean?

Line 573 and onwards. Energy dissipation rate and TKE are not synonymous. I suggest sticking to TKE, which is precise because it's what you have presented.

Line 576 replace "the expansion of" with "an increase in".

Line 578 replace "restrained" with "small". Replace "enhanced" with "greater".

Lines 578-580 Dynamic pressure, shear stress and form drag: No, delete.

Lines 581-582 What are the implications of the variation in lift force? Replace "varying range" with "variation".

Line 585 replace "to" with "of".

---

## Author Comment (AC1)

**Comment on esurf-2022-5 from Anonymous Referee #1**

Dear authors,

Thanks for submitting this interesting article. I enjoyed reading and studying it. When data is challenging to measure, combining these two techniques, SFM and CFD, seems to be a promising and reliable way to expand our knowledge. I believe that until we can measure these turbulent flows, this will be the best way to obtain good spatially distributed data in such complex systems.

The article presents a "Hybrid" approach to gain insight into the characteristics of a turbulent flow in a step-pool system. The Hybrid part consists of a digital elevation model, built based on structure from motion data, which is used to define the bathymetry in a CFD 3D model, including a free-surface capturing component. By doing so, the authors show the complex flow structure at the different parts of the test channel and comment on the importance of different hydraulics variables and their effects on the stability of the structures. The base experiments are well designed and contain enough information for this exercise. Also, they account for different conditions, ultimately reaching the destruction of the step. This is key for characterizing the complete range of flow conditions.

The general structure of the article contains a description of the experiments, CFD simulations, and data analysis. A long series of appendixes and additional figures are given at the end to support the observations. After reading it several times, I felt that it was really long and could be reduced in length without losing any of the information provided.

Many thanks for the encouragement and constructive comments!

**General comments:**

1) The article is well written but too long. Lots of repetition of ideas are found through the text, while at the same time, some details are missing, as will be detailed in the "detailed comments" section. Some ideas are broken between consecutive sentences but do not affect the delivery of the message.

The length of the paper has been reduced by shortening the result section and removing repetitions throughout the paper. All the necessary details about the experimental and numerical settings suggested by the referee have been added. The sentences that did not convey consistent information have been rewritten.

2) To me is not clear why use the term "Hybrid" modeling because only one model is being used, the 3D CFD. I know that using a model of the bed surface elevation is the second model here. Still, when calling this hybrid modeling, I was expecting a dynamically linked set of models. For instance, I initially thought about a Hybrid RANS-LES type of model or something similar. This may arise from my CFD background, but it can be a little bit misleading.

"Hybrid" in this study refers to the combination of flume experiment (physical modeling) and CFD simulation (numerical modeling). We clarified this point by revising the title to "A combined approach of experimental and numerical modeling on 3D hydraulic features of a step-pool unit".

3) Some more robust validation is required for the CFD model. In general, the validation is based on one measure of the error of water surface elevation but is not clear in terms of water depth or individual velocity. For example, velocities at the pool could have been provided and compared to the model. Zhang's 2020 paper described some PIV measurements.

We agree that water depth should be provided to show the relative error of our approach and has shown

the maximum measured water depth in the pool in Table 1. As for the PIV measurements, we did not use them because of their low accuracy from the fact that the highly non-uniform flow characteristics of step-pool features led to uneven distribution of tracer particles. The low density of tracing particles led to significant underestimation of flow velocity, e.g., low flow velocity measured at the jet or in the flow translating to jet where flow velocity should be among the highest in the surveyed area. In Zhang et al., (2020), the measurements by PIV were only used to show the strong contrast of surface flow velocity at the jet and jump regimes. The contrast was strong enough (time-averaged surface flow velocity close to 0 in the flow recirculation cell of the jump) that the data with low accuracy could still work.

4) All CFD model descriptions are based on the developer's descriptions, in this case, Flow Science, but more information from peer-reviewed or independent tests are required.

Further information for the CFD model in FLOW3D has been added based on peer-reviewed journal papers (e.g., Bayon et al., 2016; Chiu et al., 2016; Morovati et al., 2021). The corresponding citations have also been added.

5) After reading the article is still unclear to me what the "insights for the stability and failure of step-pool units are also provided." The article seems to focus more on describing the results of the modes instead of describing a mechanism of stability and failure.

The reviewer is right that stability and failure mechanism was not the primary objective for this study. But with the new information on the hydraulics and flow forces, some implications could still be obtained and were discussed in Section 4.4. This section has been revised to clarify the two main points on stability and failure. First, the role of drag force in the entrainment of step stones might not be as large as expected in previous studies because it did not increase with discharge and showed smaller magnitude and variation compared to the lift force. Second, the variation of lift force was enlarged with flow increase and was worth of further attention as it may be related to the deformation of step structure (Zhang et al., 2018).

6) In general, references are too few. The use of "e.g.," in several cases throughout the text reduces the list of essential studies used. For example, CFD simulation with irregular channel boundaries (line 62) only cites one study. There are dozens of relevant studies here. I am not saying that you should cite all of them. But using, e.g., should not reduce the number of citations to just one or two examples.

Accepted, we added references based on the detailed comments.

Based on these general comments and what is provided below, I decided that probably after a major revision the article will be suitable for publication. I genuinely believe that there is immense potential when using the approach that the authors show here and that this article, once refined, will be highly cited and a reference in the field.

**Detailed comments:**

The title could be misleading. Maybe in my case, for a particular background, the "hybrid" part of the title sounds like a hybrid CFD turbulence model. If the authors have another understanding, please discard this comment.

See our reply to the Point 2 of the general comments. The title has been revised to clarify the methodology.

Is the @163.com a good email for future contact with the author? What about a couple of years from now? Maybe one from an institution provides a better way to connect with the author.

Yes. The first author has been using this @163.com email address over 12 years and used this email as the contact for the publications of several journal papers and the ORCID. Since the first author is now working in the Institute of Geographic Sciences and Natural Resources Research (IGSNRR), Chinese Academy of Science, a new work email address (zhangchendi@igsnrr.ac.cn) has also been added.

line 34: "The high-resolution information of both topography and hydraulics for step-pool features is the key to fully reveal and describe these characteristics" It Is not clear what you mean here for these characteristics.

The sentence has been revised to "The high-resolution information of both topography and hydraulics for step-pool features is fundamental to understand such interaction".

Some references are missing. Please check that all are included. One example is Golly et al 2017, which is not listed.

Golly et al. (2017) in the text has been deleted.

line 39: " Although detailed topographic information has been available" Where is that information available? Please provide references (hereinafter ref. needed)

"Detailed topographic information available" refers to the topographic models obtained by SfM photogrammetry introduced in the two previous sentences. This sentence has been revised to "Although detailed topographic information has been made available through SfM photogrammetry". References (Eltner et al., 2016; Tmušić et al., 2020) have been added to the previous sentence.

line 54: " The PTV method managed to visualize" Is not really the method that managed to visualize but what you do with the method. I understand what you are trying to say, but it is technically incorrect. It should be something like, by using the PTV method, we can visualize. So, it is not the method that visualizes things but who provides the data.

The sentence has been revised to "The recirculation at the step toe and high-speed flow impinging at the pool bottom (the lowest area in the pool) was visualized by the PTV method for the flow near flume side walls, while the strong contrast of surface flow velocities at the step and pool areas has been illustrated based on the PIV method."

lines 62 and 63: Many more studies can be cited here. Please provide some more examples.

References (Chen et al., 2018, 2022) have been added.

line 66: The CFD approach has been applied in some numerical studies. This is redundant. Of course, that CFD has to be applied to numerical studies. Suggest to change to " The CFD approach has been applied in studies containing step-pool features " or a variation of this.

Revised to "The CFD approach has been applied in studies containing step-pool features which were conceptualized by highly simplified geometry…"

line 69: Please define what a sub-unit is. Also, previously it was used a unit-scale. Please define it too.

The definition has been given by the objects following "the sub-unit geometry including". To avoid

confusion, we revised the sentence into "it fails to characterize the sub-unit-scale morphological features such as the transverse variability…, and the grain clusters developed in the pool (Zhang et al., 2020)."

line 78: "In the flume experiment of Zhang et al" In that article there is more than one, so using "the" is incorrect here.

Removed.

line 88: This is critical when using this approach. Why use only water surface elevation (WSE) and not other variables? What about the PIV data of Zheng et al 2020? I understand how difficult it is to measure velocity in flume experiments like this one, but this velocity data seems to be available for your team.

See our reply to Point 3 in the general comments.

Figure 1 is never cited (actually, there is a wrong reference to it). RNG-VOF model is not described at the point where Figure 1 is cited.

We presented figure 1 mainly to give the readers a general impression of the workflow. More citations to it have been added in Sections 2.1-2.4.

line 101: "Two side cameras were used to capture the longitudinal profiles of the bed and water surface near the flume walls" After reading Zhang et al 2020 and Zhang et al 2018 it is still not clear how the measurements of WSE are obtained. It is not described in these two papers and I think it should be better described in this article because it is the base of the model validation.

We agree. The measuring procedures have been added in Appendix A (lines 623-625 in revision) as follows:

All the image frames taken by the side camera and top camera were calibrated according to the tape measures stuck to the side walls and the constant flume width respectively. Both the water surface from the side view and the upstream edge of the jump regime from top view were depicted by polylines in each calibrated image frame. The polylines of all the 30 image frames for 60 s were rasterized with the grid size of 0.5 cm. Then the max, 75% quantile, mean, 25% quantile and min of the water surface elevations at a streamwise location or upstream edge of the jump regime at a transverse location for all the image frames were calculated and used to compare with the time-averaged values obtained from the CFD simulations.

Details on the grain size distribution (GSD) are required in the article. Some general information about the D50, grain sizes in the pool, etc will make the article more robust.

The detailed grain size information has been presented in Zhang et al., (2018, 2020) and hence was not given in this manuscript. As suggested by the reviewer, general information with regard to grain sizes of the bed and the six step stones has been added to lines 106-107, and 112 in revision. But the grain sizes in the pool cannot be added. The pool was not built artificially but formed by local scouring (Zhang et al., 2018). The grain size of sediment filled in the downstream area of the step model as the initial bed condition for each run was not strictly controlled because the grain size in the pool affects the scour hole geometry in an insignificant way (Lenzi et al., 2002; Comiti et al., 2005).

Line 110 The terminology used at this point in the article is not clear. What is KS (I know that means keystone, but that is described several pages later)? What means T2 or a T run?

We have added "KS in (a) is short for keystone. The run index in (b) is CIFR (continually-increasing-flow-rate) T2." at the end of the caption of Figure 2. The definition of KS (i.e., the immobile/rarely mobile large

stone which facilitates step forming) has also been given in lines 110-111 in revision.

Line 113: Any comments about the effect of stopping the water circulation on the bed response would be appreciated

The tail gate was firstly to be closed to inundate the step-pool model and then the flow was closed to prevent the bed surface from significant morphological variations (Zhang et al., 2020). If no special procedures were used, although the coarse grains rarely moved during water level lowering, the fine sediments (grain size <10 mm) among the coarse particles might be transported.

line 126: What other facilities? please list them.

Revised to "...as the steel frames of the flume and the frame supporting the top camera restricted..."

line 126: DMS at different flow rates? Is that "for" different flow rates? Please re-write this sentence because it is confusing.

Revised to "The DSMs for all the tested flow rates were cropped..."

Line 128: What means relatively poor? Compared to what? Can you provide a measure of good or poor?

There were clear distorted reconstructions of the transparent glass walls which were not flat in the raw DSMs because the reflections in the glass led to varying features recorded in the images. After cropping the walls and some marginal areas connected to the walls in the reconstruction, the cropped DSMs had a width slightly smaller (about 1.5-2 cm narrower) than the flume width.

The sentence has been revised to "The reconstruction of the transparent glass walls in the DSMs showed clear distortion because the reflections in the glass led to varying features recorded in the images that could hardly be matched correctly in SfM processing."

Line 128: What is KS? please define.

See our reply to the comments for line 110.

Lines 132 and 133: If these WSE were captured every 2 s, what is then provided? Is it an average over X seconds? Please describe it in detail. Also, how are longitudinal profiles of the bed used in this study?

In model validation, the measurements of WSE during 60 s (i.e., 30 surface profiles extracted from photographs) were used. The max, 75% quantile, mean, 25% quantile and min of the measured water surface elevations at each streamwise location were presented in Figure A4-A6. This information has been added in Section 2.3 and Appendix A. The longitudinal profiles of bed surface close to the flume walls were used to obtain the water depth measurements near the side walls. This point has been added in lines 143-144.

Line 137: This "high enough " is the 3.3 - 3.9 mm? Can you provide a measure to understand why is high enough? For example, the D50 was 1 cm, therefore, with 3.3 mm all grains are captured.

Yes. It refers to the mesh size of 3.3-3.9 mm. The step stones had the grain sizes of 76-104 mm (in line 106 in revision) and the diameters of the guardian step stones ranged from 64-108 mm (in line 112). So, the grid size was small enough to characterize the step stones. Our focus for the pool area was mainly on the micro-bedforms. The development of the micro-bedforms were normally involved with coarse grains larger than 25 mm. The grid setting was able to characterize the geometries of these coarse grains and the micro-bedforms. As for the pool areas where no distinct bedforms formed, the local bed surface was relatively flat

and hence could also be captured by the gridding setting.

The information of grain sizes for the step stones have been added (see our reply to the comment for line 101-102) and the micro-bedforms in the pool were mentioned as an example for the topographic characteristic of the step-pool model in the following sentence.

Line 137: "geometric feature" Please list them. It is not clear what geometric feature means in this context.

Mainly the topographic characteristics of the bed surface. We changed "geometric feature" to "topographic characteristics".

Line 138: "reduced the requirements for computing resources of the numerical simulation " How does the elevation model help reduce the computing resources used? That most likely is the only function of the numerical mesh, but not on the DSM. Please explain. This could be helpful for other researchers.

The FAVOR$^{TM}$ technique assumes straight-line connections between intersection points within the face of each mesh grid. The straight-line assumption introduces a small error when the geometry boundary is curved inside the cell. But the error can be reduced if smaller mesh size is used. To maintain as many geometric details of the extruded DSM in the favorized geometry as possible, the mesh size for the main computational domain was kept smaller than the DSM grid in FLOW3D. Hence, if smaller grid size was used to remesh the DSM, smaller mesh size should also be applied in the computational domain, which would enhance the requirements for computing resources. This point has been reflected in lines 191-193.

Line 140: Commercial solution is difficult to understand. Could it be commercial software or commercial model, or a commercial solver?

Revised to "commercial software".

Line 141: This part is critical. What do we understand as "shown good performance"? Please provide more information here, some peer-reviewed examples using flow3D.

References (Bayon et al., 2016; Chiu et al., 2016; Morovati et al., 2021) have been added.

Line 142: Please provide references for the TruVOF. The article of Hirt and Nichols is well known and one of the fundamental studies in VOF, but it is not about TruVOF.

The reviewer is right. Hirt and Nichols (1981) is for the VOF method in the sentence. Reference (Flow science, 2016; Bayon et al., 2018) has been added specially for the TruVOF technique.

Line 143: Same for FAVOR, please provide more information here, some peer-reviewed examples using FAVOR

We used the References (Hirt and Sicilian, 1985; Flow science, 2016) where FAVOR appears in the paper for the first time and added more references (e.g., Chiu et al., 2016; Morovati et al., 2021) in the following sentence to highlight the advantage of this method.

Line 146: Please provide more references to justify the model selection. Using the vendor's information is not enough. I believe that it is a good model, but using Flow science, 2016 as a reference is suitable just for description.

Line 146 was "…of complex geometric shapes (Flow science, 2016). 3D solid entities rather than 3D surfaces are required to build the terrain…" We do not see any issue related to model selection. If the reviewer

referred to the selection of turbulence model, please see our reply to line 168 below. If the reviewer referred to the FAVOR™ technique, references have been added (see details in our reply to line 143).

Line 148: How was the FAVOR technique tested here?
It should be "previewed by the FAVOR™ technique". Revised.

Line 154: What does it mean by "leaking" - " When leaks emerged between the DSMs of bed surface ..."
It is better to call them 'gaps'. As described in Section 2.1 (also the reply to line 128), marginal areas of the DSMs close to the flume walls were cut, but the cuts were not always smooth as the cutting plane might go through some triangular mesh cells. This leads to the irregular margins after the DSMs were extruded (Fig. 3b). The cropped DSMs were generally 1.5-2 cm smaller than the flume width but at some spots the DSM was even narrower. While the mesh block width was uniform for the whole computational domain, gaps appeared where the width of the cropped DSM was smaller than the width of computational domain. The bed would leak water at these gaps so we added solid rectangular columns to block the gaps. The information above has been added in lines 166-168.

Line 168: Please describe in more detail the Renormalized Group (RNG) $k$-$\varepsilon$ turbulence model and its implementation in the solver. No references to the turbulence modes are given. Please provide references to literature and now the software developer.
The Renormalized Group (RNG) model in FLOW3D is based on methods developed by Yakhot et al. (1986, 1992). All the turbulence models including the RNG model have been modified slightly by FLOW3D to include the influence of the fractional areas/volumes of the FAVOR™ method and to generalize the turbulence production (or decay) associated with buoyancy forces (e.g., buoyancy effects associated with non-inertial accelerations). The implementation has been added in lines 184-186.

Line 169: "The VOF technique ..." This has been said already.
The sentence has been removed.

Lines 171 to 175: Why there is always ranges of mesh properties? 2-3 structured mesh blocks, 24-37 cm, 6.5-9.4 million units, etc. Are those different for different flows?
Yes. The models at different discharges had different mesh settings. This stems from the morphological variation of the step-pool unit with flow increase. The more developed pool scour and increased water depth resulted in an increase of mesh block height. The different lengths of DSMs led to the variation in the mesh block length. Consequently, the mesh cell number also varied. The reason has been added to lines 193-195.

Line 176: It is unclear how uniform and non-uniform mesh differ when using them in the model.
The uniform mesh had the constant mesh size of 2.5 mm in all XYZ directions while the non-uniform mesh had larger mesh sizes in two directions and hence, had fewer mesh cells. (i.e., still 2.5 mm in Y direction but 2.5-5 mm in X direction and 5 mm in Z direction). The more detailed settings have been added in lines 196-198 in revision.

**General comments on the simulations:**

Before resuming the line-by-line analysis, I will summarize some missing numerical details in the article. These are critical because they give the reader an idea about how the model was configured. After analyzing

the results and looking at the figures, I think first order numerical schemes were used. I can't know this for sure, but it is my impression because velocity profiles and wse are super smooth.

Yes, the reviewer's guess is correct. First order momentum advection was used to ensure the computational stability for the flow over the highly complex bed surface of a step-pool unit. This setting has been discussed as one of the limitations for this study in Point 4 in Section 4.5.

- Please provide details about how each term was treated (Numerical schemes) For example, did you use MULES for the water phase? If so, was it limited? with artificial compression? second-order in velocity? etc.

To our knowledge, MULES was not used in FLOW3D. Instead, a donor cell advection method was used for the VOF setting in the cases "Two fluids with sharp interface" and "One fluid, free surface" in FLOW3D. In the former case the donor cell can be on either side of the interface while the donor cell is always on the volume fraction = 1 side of the interface in the latter. Unfortunately, FLOW3D is commercial, so the detailed algorithm is unknown.

- What convergence criteria were used for Vel and Pressure? What about other variables?

FLOW3D employs the projection-based generalized minimum residual method (GMRES, Saad, 1996) as a pressure-velocity solver (Flow Science, 2016). The linear equation systems were numerically solved by GMRES method with a Krylov subspace dimension of 15 following Valero et al. (2018) and Morovati et al. (2021). The GMRES solver uses a combination of baseline absolute, steady-state absolute and relative convergence limits to decide on a good convergence limit, which is quantified by the "multiplier for dynamically adjusted convergence criterion" parameter in FLOW3D (Flow Science, 2016). This parameter value was set as 1 by default in this study.

- The results we are seeing, are they time-averaged? Over how long?

They are time-averaged results based on the exported data at a frequency of 2 Hz for 30 seconds (added in lines 219 and 271).

- What are the boundary conditions for k and epsilon?

How does the model treat the variables when it has a discontinuity in permeability (bed/water transition)? I am wondering if wall boundary conditions are required.

For all the boundary conditions used in this study, k and epsilon can only be set for the specific velocity boundary at the inlet and specific pressure boundary at the upper faces of the mesh blocks. The user does not have to set k and epsilon values at the inflow boundary if LES model is not used. These two parameters will be resolved during the calculating iterations by the RNG model (Flow Science, 2016). Air was regarded as void region rather than a fluid in our simulations (added in line 181), so there was no need to set k and epsilon values in the specific pressure boundaries.

The reviewer is right that wall boundary condition is not necessary for the solid geometry imported to FLOW3D. The surfaces of the solid geometry recognized by FAVOR$^{TM}$ method will be used as rigid walls automatically. All advective and diffusive fluxes are automatically zero and velocity normal to the boundary is also zero at rigid walls where the fractional open areas vanish. The wall boundary condition at the lower mesh planes was unnecessary as these mesh planes were lower than the solid bed surface in a DSM. But this setting had no influence on the simulation and a boundary condition has to be chosen for each mesh block face in FLOW3D.

- Why this particular turbulence model was used? Changes in pressure would suggest using a k-omega sst type of closure. I don't think it is wrong to use k-e, but it must be justified.

The Renormalized Group (RNG) $k$-$\varepsilon$ turbulence model was applied based on three concerns. First, the RNG model has been used in hydraulic structures like vertical drop pool (Chiu et al., 2016) and stepped spillway (Morovati et al., 2021) which also show jet and jump regimes as step-pool does. Second, the RNG model shows affordable computational cost for our workstation, and relatively high computational stability for the complex geometries of the DSMs used in this study. Third, FLOW3D does not provide a k-omega sst model. Only a standard k-omega model is implemented in the software. The k-w model requires smaller cell size close to solid boundaries and hence would increase the computational cost significantly. These reasons have been added in lines 182-189 in Section 2.2.

- In the upstream end, the flow enters the study reach. The images show that distance is too short to be a fully developed flow. Is there any consideration used to ensure that the numerical flow is a good representation of the actual flow once it reaches the step?

We admit that the short distance at the upstream of the step was a limitation for this study and has discussed it in Section 4.5. However, the possible errors caused by this limitation are acceptable for two reasons.

First, the hydraulics in the pool was the main focus for the 3D hydraulic features of a step-pool unit. The flow in the pool was mainly affected by the jet separated from the step surface. The jet would not be significantly influenced by the turbulence development upstream of the step because turbulence would be reduced at the transition flow with high speed (Wohl and Thompson, 2000). This has been tested by changing the inlet location in the simulation at 43.6 L/s (moving 2 cm upstream and 5 cm downstream) and no significant difference in flow structures was observed at the downstream of the step. The flow structures upstream of the step were slightly different from the original simulation for the paper as expected because the bed surface at the inlet cross section also became different when the inlet location was changed.

Second, the distance for the flow turbulence to be fully developed was not very long for a steep rough bed surface with relatively shallow water depth. This is supported by the fact that the streaky coherent structures already formed at the downstream of protruding grains upstream of the step in this study (Fig. 9).

- How was the flow initialized? There is a mention that the pool had water initially, but no other details are provided.

That was the only setting for the initial condition. With the bed surface submerged in most areas in the pool, the flow came from the inlet cross section.

- How was defined the location of the WSE. In VOF, the fractions are a continuum, so depending on the flow type, some studies use 0.5 (others 0.1 or 0.9) to define the boundary between water and air.

We used the volume fraction of 0.5 as the criterion to visualize water surface, following the default setting in FLOW3D. This has been added to line 220.

- How much space was left from the WSE to the upper boundary. What are boundary conditions used to model the air entering/leaving the domain?

The smallest vertical distance between the WSE and the upper boundaries of the mesh blocks at the inlet (the water level was among the highest in the water surface profiles in a simulation) was kept larger than 5 cm in all the simulations. This information has been added in lines 194-195.

Specific pressure boundary using the standard atmospheric pressure was used for the upper planes of all

the mesh blocks to keep air phase here. The continuative and outflow boundary conditions both allow air exchange in FLOW3D, added in lines 204-205.

- Was the velocity specified at the inlet uniform? a logarithmic profile?

Uniform distribution of flow velocity was used at the inlet for two reasons. First, it was the simplest way to set the specific velocity boundary condition for the inlet section, with only water level and velocity magnitude at three directions to be determined. Second, no measurement for the vertical profile of flow velocity was accessible in the experiments of Zhang et al. (2018, 2020). Only the cross section-averaged flow velocity was available, calculated from the measured discharge and water depths at the location of the inlet from the measurements at the side walls.
* * *
All these comments point towards a better understanding of the model configuration.
Now I will continue with line-by-line comments.

**line-by-line comments**

- Line 180: For the no-slip condition, what is the cell size on the walls? What about the other variables, for example, pressure, k or epsilon?

We used a structural mesh block with uniform grid size of 2.5 mm to cover the step-pool component. So the cell size on the walls was also 2.5 cm. In FLOW3D the wall boundary is no-slip and has a zero-velocity condition normal to the boundary. The variables mentioned by the reviewer cannot be predetermined in boundary condition in the software.

- Line 185: ' which efficiently accelerated " Compared to what? compared to starting with an empty (only air) domain? How do you know that it was faster? This could be helpful for other researchers because when using VOF obtaining good initial conditions is a time-consuming process.

Yes, compared to starting with dry bed surface in the pool. We did not use the ponded pool condition at first and the time step automatically determined by the software decreased sharply when the flow impinged at the bed surface in the pool. Sometimes the computation divergence occurred before the pool was ponded. This was related to the very complex flow regimes after the flow reached the irregular bed surface in the pool (Fig. 3a). Splashes might appear when the high-speed jet first hit the protruding grains in the pool. This point has been explained in lines 207-209.

- Line 190: "were collected after the solution was steady, with the variation from the mean less than 0.5% at each flow rate" This is an essential aspect of the numerical experiment design. In a turbulent environment like the one used in this case, we would expect high fluctuations in every single variable, this can be seen in the images of Zheng et al 2018 and 2020. A variation of 0.5% from the mean seems to indicate the use of a highly diffusive numerical scheme. The good thing is that it is stable for this complex simulation but not really accurate. Please provide more details. Maybe a plot of velocity in time at some locations could be helpful.

Since RNG $k$-$\varepsilon$ turbulence model was used in this study for the complex geometry, the high fluctuations due to turbulence could not be reproduced by this Reynolds-averaged Navier-Stokes (RANS) model. This has been added as one of the limitations of this study (Point 4 in section 4.5). The spatial distribution of hydraulic features instead of temporal distribution characteristics was the focus of this study. This is why the

export frequency of the simulation was low (2 Hz) and the exported results were further time-averaged for the analysis.

- Line 191: A frequency of 2 Hz could be too distant in time to capture some important turbulence properties. This is only 2 observations every second. For example, you mention later that you use velocity fluctuations for some calculations. Please comment.

The kinetic energy, turbulent kinetic energy and turbulent dissipation were all derived from the solver. The actual frequency of the data used to calculate these variables was related to the time step rather than the export frequency (2 Hz). We revised the text around Eq. 1-2 to make this clear.

- Line 195: Please explain how it was decided that grid independence was reached. Please provide a metric.

We used longitudinal water surface profiles (Fig. A1) and flow velocity distribution in the streamwise section (Fig. A2) located in the middle of the flume to determine whether the grid independence was reached. If the decrease of mesh size did not cause significant variation in the two metrics mentioned, the grid independence was recognized. It can be seen that the water surface profiles overlapped for the results using mesh cell size smaller than 0.3 cm. The flow velocity distributions became almost the same after the mesh cell size was reduced to below 0.3 cm. These test results show that the cell size of 0.25 met the requirement of grid independence.

To better correspond the result of water surface elevations to the flow velocity, only the results for the longitudinal section at Y=0.3 cm is presented in Fig. A1 in revision.

- Line 200: Although there are several metrics for evaluating the error in WSE, a comparison to water depth will also be helpful. It is difficult to judge if 2-3 cm is a small quantity or not because there is no information related to water depth.

We agree. Considering the water depth varied significantly from the upstream to the downstream in the step-pool unit, we added the maximum flow depth in the pool in Table 1 as a measure for the errors.

- Line 214: How are velocity fluctuations evaluated?

The time-averaged results were analyzed in this study without examining the velocity fluctuations. See details for the calculations on turbulence in our reply to line 191.

Line 219: Are all these terms in equation 3 calculated by post-processing the information or given as a result of the model? Do you have to calculate all the gradients or there is a function within Flow3D that does the job for the user?

Qcriterion was calculated as a result of the model by FLOW3D. This has been clarified in line 244.

- Line 221: Then, obtaining all these results, what is the typical time step in the simulations? Any CFL criteria were used?

Automatic time-step control provided by FLOW3D was used in this study and the time step varied with step-pool morphology, incoming flow conditions and flow regimes. After a simulation was relatively steady, the time step was: $3.5\text{-}4.6 \times 10^{-4}$ s for $Q$ =5 L/s; $2.5\text{-}2.6 \times 10^{-4}$ s for $Q$ =12.4 L/s; $1.15\text{-}2.12 \times 10^{-4}$ s for $Q$ =22.8 L/s; $1.0\text{-}1.35 \times 10^{-4}$ s for 32.1-49.9 L/s. The general information of time step has been added in lines 212-215.

The default automatic time-step option allows FLOW3D to automatically adjust the time step size to be as large as possible without exceeding any of the stability limits, affecting accuracy, or unduly increasing the

effort required to enforce the continuity condition by the pressure solver (Flow science, 2016). Several stability conditions are used in FLOW3D, including that fluid must not be permitted to flow across more than one computational cell in one time step. In the mode of automatic time-step selection, CFL is controlled to be smaller than 0.85.

- Line 221: How is the shear stress calculated? If it is a result from the solver, what equation is used?

Shear stress is obtained from the solver and calculated by the friction velocity, which is computed iteratively from the log-law formula. This method is very necessary because the FAVOR method does not precisely locate wall boundaries within a cell.

- Line 222: Please explain why the dynamic pressure is the one analyzed here.

The static pressure was only affected by water depth and would take up the majority of the total pressure in the pool area where water depth was large but flow velocity was low. This will lead to a similar spatial distribution of the total pressure and the water depth in the pool area, which cannot reflect the spatial distribution of flow kinetic energy. As a result, the jet impinging at the bed surface in the pool cannot be visualized. To remove the influence of water depth distribution, we used the dynamic pressure. The reason has been added to lines 250-251.

- Line 225: Please explain how it is obtained in a given model. Is a result of manually post-processed from VOF?

The dynamic pressure was calculated based on the total pressure and water depth at a horizontal location which were both obtained from the solver. Added in line 249.

- Line 225: Was water density equal to 1000 kg/m3? or any correction for temperature was used? What about air density and viscosity?

We applied the fluid properties for water at 20°Cpreset in FLOW3D: density = 1 g/cm$^3$ and viscosity = 0.01 g/cm/s. Water temperature was not measured during the flume experiments, but should be around 20°C because tap water was used and was around 20°C during summer in the city where the experiments were conducted. Water density has been added in line 254.

Since only water was regarded as fluid, only the air pressure was needed at the upper boundary of each computational mesh, set as specific pressure boundary in FLOW3D. Standard atmospheric pressure (i.e., 1.01325*10$^5$ Pa) was then used.

- Line 231, Is Using measured at the inlet?

No. It was the cross sectional-averaged flow velocity at the inlet of the sampling volume. The following sentence clarifies this point.

- Line 234: Please describe the threshold method.

The threshold method has been introduced in the following sentences (lines 263-266). This sentence has been revised to reinforce the link to the following sentences.

- In general, the results are well described. They could be reduced a little bit, but they are well explained. Good job!

Thanks! The result section has been revised to be more concise.

- Line 297: What is here a contraction? This term may be used with different meanings in different fields. Please provide a definition.

The definition has been given in the caption for figure 3 where it appears for the first time in the paper. The definition is further revised as "cross section where the hydraulic jump starts to appear was referred to as the contraction section." The definition has also been shown in lines 220-221.

- Figure 6. At the end it says, Figure 3. Probably it is Figure 5.

Yes, it should be Figure 5 and has been revised.

- Section 3.2.2 What is the role of shear stress in this type of structure?

The shear stress is the basis for skin drag of the step stones and grain resistance of the step structure at larger spatial scale. Our results now reveal that the magnitude of shear stress in a step-pool unit is generally two orders of magnitude smaller than the dynamic pressure at all flow conditions. This suggests that the skin drag may be much smaller than the form drag in a step-pool unit. However, as suggested by the other referee, the conclusion can only be drawn after extensive work is done on the quantification of skin and form drag, and grain resistance.

- In the discussion section, I believe some paragraphs connecting the observations from the results are required. How is everything connected to stability?

We have gone through the discussion section to strengthen the links to the results. Section 4.1-4.3 did not refer to step-pool stability. Only in section 4.4 we discussed the stability and failure issues. Attention was paid to stability because the step model was destabilized in CIFR T2 run at last. So we tried to look at the step stability issue with the newly obtained information for the forces acting on the step stones in this study.

- Line 553 - Air entrainment is possible to include, at least partially, in VOF. Especially in this type of model when there is a jump. Please comment on why it is not considered. I said partially because it depends on the grid size. Required more post-processing because air and water fractions or their boundaries are more difficult to isolate, but it is possible. Please check the literature on bubbles (OpenFOAM has been used widely for this).

Yes, it is possible to consider air entrainment in the simulation by activating the air entrainment model in FLOW3D. But we did not include air entrainment mainly for three reasons. First, air concentration was not measured in the experiments of Zhang et al. (2020). So the simulated air fraction in water flow is impossible to be validated. Knowledge or measurements on the air entrainment characteristics for step-pool features were also very limited. As a result, we had problems in determining the parameters needed for the air entrainment model (e.g., entrainment rate coefficient, drag coefficient for air bubbles, average air bubble diameters). Second, activating the air entrainment model will increase the computational cost because the interaction between the air and water needs to be addressed and the air phase needs to be regarded as a fluid rather than void. This might lead to unaffordable computation cost for us. Third, as said by the reviewer, the computation stability would be reduced by coupling the air entrainment model with the RNG-VOF model because the entrained air would lead to the variation of fluid density (Flow Science, 2016). We have stated these concerns in Point 2 limitation in Section 4.5. Nevertheless, we believe that air entrainment may be coupled to numerical simulations for step-pool features in future when further knowledge is available based on observations with advanced experimental techniques (e.g., Hohermuth et al., 2021).

- As a limitation, please mention all those that may arise by selecting the numerical schemes and model configuration, also, by using a k-e turbulence model.

A new paragraph has been added as Point 4 of limitations in Section 4.5. The original Point 4 was changed to be Point 5.

**Comment on esurf-2022-5 from Referee #2 Keith Richardson**

This paper presents results from a novel application of CFD modelling to a step-pool unit. This study is innovative and the results greatly enhance our understanding of the detailed 3D hydraulic characteristics of step-pools, which present a challenging environment for direct measurement. The results will also advance our understanding of the role of hydraulics in the formation and stability of step-pools and the effect of step-pools on flow resistance. The identification of two discrete hydraulic structures that function as energy dissipators in a step-pool unit, and the discussion of the energy dissipation of 2D compared with 3D hydraulic structures, are of particular interest. I would like to congratulate the authors on this study.

Thanks a lot for the encouragement! The following detailed and constructive comments are greatly appreciated by the authors.

However, I have two main concerns about the way the results are explained, described and interpreted. On one level, these concerns are not particularly serious and can be relatively easily addressed in the paper to make it worthy of publication, but on a more fundamental level they show that there is some misunderstanding of the main underlying topics of the study.

Firstly, in the description of the results, there is misunderstanding of the coherent and persistent hydraulic structures in the step-pool, specifically in relation to the terms "wake", "hydraulic jump", "jet" and "regime". The authors in effect are using their own working versions of these terms. I shall discuss the use of each of these terms in turn.

The misuse of the hydraulic terms has been recognized and revised throughout the draft.

According to the classic definitions of jump (e.g., Finnemore and Franzini, 2002), jump regime starts from the contraction section while the distance between the point of separation at the step and the contraction section is occupied by jet regime. However, the contraction section is not a rigid boundary of the jet and jump regimes. Both the high-speed flow and the vortices formed at the step toe extend to the downstream area of the contraction area. We attempted to describe such hydraulic segmentation but used the wrong words that should be used for larger spatial scale.

The region at the toe of the step is referred to as a "wake" or "wake zone" (lines 258 and 261) and the flow recirculation cell at the toe of the step is referred to throughout the paper as a "wake vortex". The term "wake" usually refers to the region immediately downstream of an object or roughness element in flow whose dimensions are much larger than those of the object or roughness element. The term "wake" cannot be applied to flow immediately downstream of a channel feature whose dimensions are similar to those of the flow, and that acts as channel topography rather than roughness. The so-called "wake" is merely the region at the toe of the step and the "wake vortex" is a transverse flow recirculation cell attached to the toe of the step.

The reviewer is right. If we regard the step as a roughness unit, then the wake zone would be the entire pool area downstream of the step. However, 'wake zone' only refers to the area from the step toe to the negative slope near the bed. So it is inappropriate to use this term in the draft. We used "step toe/the toe of

the step" as suggested by the reviewer. "Flow recirculation cell" is used as well as vortex/vortices to describe the roller eddies located at the step toe near bed surface.

There is indeed a hydraulic jump in the pool, but the hydraulic jump (or more usually, "jump") is repeatedly referred to as a "surface jump" and described as existing only at the surface. A hydraulic jump is a feature that occupies the entire water column; the surface feature referred to by the authors of this paper as a "jump" is in fact the flow recirculation cell of the hydraulic jump, in which flow spills backwards down the adverse surface slope generated as flow decelerates to a subcritical condition from a supercritical one.

The jump regime should occupy the entire water depth (Finnemore and Franzini, 2002) so it starts from the contraction section. What we describe here is mainly the flow recirculation cell near the water surface. The term has been corrected throughout the paper.

The authors refer to a "jet" or "jets" that exist(s) above the "wake vortex" and below the "jump" that is a feature separate to the "wake vortex" and "jump". For example, lines 435-436 talk about the "segmentation" of these features. However, the impinging jet of the step and its associated recirculation cell, and the flow within the hydraulic jump below the recirculation cell of the jump, are unified and single features rather than distinct and separate features. This is especially the case with the hydraulic jump, which as mentioned previously, extends through the entire water column; the so-called "jet" is merely the fastest downstream-directed flow within the hydraulic jump.

The jet was meant for the high-speed flow after the flow plunged into the pool. But it is not the precise word for the area downstream of the contraction section where the hydraulic jump occupies, as the reviewer said. We agree to replace jet with high-speed flow when describing the segmentation of flow velocity and turbulence in the pool. As for the impinging jet and surface flow regimes (revised from surface jet) mentioned in section 4.1, jet is kept to follow the definition by Wu and Rajaratnam (1998).

All three features described above ("wake"/"wake vortex", "jump" and "jet") are referred to as "regimes" by the authors, when "structures", "features" or "regions" would be more appropriate terms. A regime is a region of the flow that extends across the full width (or most of the width) and depth of the flow, and over a longitudinal distance usually several times the flow width. For example, in terms of the Froude domain, the flow regime can be described as supercritical, subcritical, or transcritical. In this respect, the flow regime in the authors' experiment and simulation is transcritical, and this describes the flow over the step-pool unit, or a number of step-pool units.

We have examined throughout the draft to revise "regime" to the terms suggested by the reviewer.

In general, the identification and description of the hydraulic coherent structures need correcting and clarifying, including the locations of these features. There are further examples in the detailed comments below.

All the terms mentioned here and in the detailed comments have been carefully checked and corrected.

Secondly, the discussion of shear stress, dynamic pressure, form drag, skin friction, form and grain flow resistance and flow resistance partitioning at 514-523 is both confused and confusing, and illustrates an almost total lack of understanding of these subjects. I don't know where to begin suggesting editing this section, and I can only describe this section as nonsense. In any case, this section is oxymoronic. It partitions form and grain resistance before stating that attempting such partitioning is invalid. I suggest deleting it. It is

possible that conclusions regarding the relative magnitude of grain resistance and form resistance at the reach or step-pool unit scale and regarding flow resistance partitioning could be drawn from the authors' results, but it would require significant analysis further to that presented in this paper. Specifically, it would require integrating the boundary shear stress over the step-pool unit and comparing it with the total shear stress due to the downstream component of the weight of water in the step-pool unit, and making any necessary adjustments for flow non-uniformity between the inlet and outlet sections.

We agree that more quantified results on the resistance over a step-pool unit can only be achieved by extensive work based on further processing of the CFD outputs. Unfortunately, this paper has been too long to include further details on resistance. We meant to preliminarily discuss the implications of our results for resistance partition in a step-pool unit here, but it turns out to be confusing. As a result, this part (lines 514-523 in the first submission) has been removed, as suggested by the reviewer.

**Detailed comments are given below, with relevant line numbers.**

Line 1 replace "on" with "of" in title.

The title has been revised to "A combined approach of experimental and numerical modeling on 3D hydraulic features of a step-pool unit", according to the comment of referee #1.

Line 16 and throughout paper. Plural of vortex is vortices.

Revised throughout the paper.

Lines 28-29 Abrahams et al. is not an appropriate reference here. Abrahams et al. is about the relationship between step-pool geometry and flow resistance. It does not directly address the effect
of step-pools on channel stability.

This reference has been removed and Lenzi (2002) is used here.

Lines 32-33 insert "and" between "hydraulics" and "stability". Replace "these dimensions" with "them".

Done.

Line 42-45 replace "different from" with "unlike". Replace "which result in" with "resulting from". Replace "oscillation" with "alternation".

Done.

Also relevant here is the fact that formative flows of step-pools are very high discharge, typically c. 50 yr return interval, making them impractical to measure.

A sentence has been added as "Also, the formative flows of step-pools are exceptional floods with a return period of about 50 years (Lenzi, 2001; Turowski et al., 2009), making the hydraulic measurement impractical in the field."

Delete "Salt or rhodamine dilution and".

Done.

Line 56 replace "at" with "to".

Done.

Lines 85-87 and 95-96 are repetitive of lines 77-80.

Lines 77-80 have been shortened as "we established a combined approach of experimental and numerical modeling on the 3D hydraulics of a step-pool unit and analyzed the 3D distribution of hydraulics and flow forces." Lines 95-96 were removed.

Section 2.1 There are missing experimental details. What were the stepstone dimensions, initial and final step height, and step spacing?

Grain sizes of the step stones have been added in Lines 106-108.

The step heights measured at the right flume wall at 5 L/s and 49.9 L/s have been added in line 130 as 7.2 cm and 15.4 cm respectively.

The spacing between the step and guardian step was kept constant as 0.7 m throughout the experiment. This information has been added to this sentence.

Lines 102-103 why was Froude scaling employed as opposed to some other form of scaling? Without giving justification, any scaling method used is arbitrary.

Froude scaling was used to follow the gravity similarity criterion as gravity was the main acting force to the flow in mountain channels. This has been added in line 108.

Lines 105-106 "We did not manually build any pool features…" What does this mean, and what was the initial constructed longitudinal profile?

It means that the pool feature was not created artificially.

The sentence has been rewritten as "The area between the step and guardian step was filled with sediment mix to cover the red paint on the step stones, and local scouring on this sediment mix by the flow formed the pool morphology during each run."

Line 111 What are "T runs"? Delete "designed". Replace "step by step" with "stepwise".

"T runs" was not used alone. It followed "CIFR" and the run was called as a "CIFR T run". T was short for topography, meaning the topographic data was collected during the run.

Done.

Line 114 What does "discharge change interval" mean?

The lasting time between two neighboring discharge changes. It was marked in Fig. 2b in the revision.

Line 123 Replace "step-pool-step" with "step-pool".

Done.

Line 125 Replace "upstream area of" with "area upstream of". Replace "step model" with "step-pool model".

Done.

Line 127 The acronym "KS" has not been used before and needs to be explained.

A sentence has been added in Lines 107-108 to introduce the keystone.

Line 140 Replace "solution" with "software". Full stop after "platform". Start new sentence and replace

"which" with "This software".

Done.

Lines 152-156 What effect do these added topography components have on the reliability of the CFD results?

After adding these components, the simulated water surface around the pool outlet matched well with the measurement, with the backwater effect removed. This means the model reliability was improved by such a setting.

Line 191 This is a relatively low sampling period and frequency. How were the sampling period and frequency selected? High frequency turbulent fluctuations will be missed and will not contribute to shear stress, dynamic pressure or TKE. What effect will this have on the results?

The RNG $k$-$\varepsilon$ turbulence model is a highly computational stable approach and the result did not show large variations after becoming steady. Also see details in the reply to the comment of Referee #1 on Line 190. So a period of 30 s was long enough to get the time-averaged values of hydraulics.

The calculation result was exported every 2 s but was calculated based on the time step. The information of time step has been added to lines 214-215 and at the level of $10^{-4}$ s, which was small enough for simulation of shear stress, dynamic pressure or *TKE*.

Lines 200-201 Unit of measurement missing: 3 cm? How do these RMSE values compare to key length scales such as Step stone size, step height, step spacing etc.?

It should be 3 cm. Revised.

The information of step stone size and the maximum water depth has been added (also based on the comment from Referee #1 on this point) and Table 1 respectively to show that the simulation errors were acceptable.

Lines 222-225 This is verbose, and the separation of total pressure into dynamic and hydrostatic components is trivial and it is not necessary to describe it.

The sentence to describe the separation of pressure has been removed.

2.4 Data Processing: there are missing calculations: how were bed shear stress and the forces acting on step stones calculated?

Shear stress and forces on step stones were calculated in the turbulence solver, as described in lines 248 and 255-256 respectively.

Line 255 delete "as the main flow".

Done.

Line 258 replace "deviated" with "separated". I suggest better and further explanation here. It is true that the jet at Y =-18 cm does not impinge on the bed but the jets at all three sections are separated from the bed by a vortex (flow recirculation cell); the jet at Y = -18 cm does not impinge on the bed because the vortex here extends further downstream than that at the other two sections and then merges with the jet.

Done. The sentence has been revised to "…in the section Y = -18 cm, in which the vortex at the step toe extends further downstream than that in the other two sections and then merges with the jet on the negative

slope.", as suggested by the reviewer.

Line 260 replace "feature" with "features". Replace "limitation" with "reduction" or "contraction" or similar.
   Done.

Line 269 and throughout paper. Replace "at the upstream area of" and "at the upstream of" with "upstream of"
   The replacement has been done throughout the paper.

Lines 271-272 "point of separation of the jet from the step face" would be better than "detaching point". The point of separation of the jet and the contraction section need to be explained and described because it's not clear where they are. "flow concentrates" is also confusing and inaccurate. I suggest using the phrase "high velocity regions".
   The suggestions have been accepted.

Lines 272, 274 and 276 "lower top elevations" and "higher top elevations" are confusing phrases. I suggest "low/high points within the step crest".
   "lower top elevations" has been revised to "low points within the step crest" while "higher top elevations" is changed to "high points of the crests of the four step stones".

Lines 273-275 "discrete vortexes near the bed surface" and "the gaps between the wake vortexes near the bed are filled with high speed flows"; "3D flow separation cells at the toe of the step with transverse axes separated in the transverse direction by regions of high speed flow" would be better. They expand in the longitudinal direction only with an increase in discharge. Also, the high speed regions are centred on the contact points of the step stones and the flow separations cells are centred on the centres of the step stones, which is the reverse of your description.
   The sentence has been revised to "…shows the existence of vortex cells at the step toe with transverse axes separated by regions of high-speed flows."
   The vortices also expand in the transverse and vertical directions with flow increase. See sections x0+2 in Fig. 5 and flow structures in Fig. 8. The high-speed regions are located downstream of the high points of the step crests. The view in Fig. 5 may cause misunderstanding to the relative locations of the vortices and step stones. See Fig. S12 for a clearer view.

Line 275 "contact points" would be better than "connecting points".
   Accepted.

Line 277-278 replace "pool bottom" with "base of the pool". Replace "shrink" with "are less pronounced". The jet is referred to here but was not mentioned in the description of the section at X0+15; mention in description of both sections to avoid confusion.
   We would like to keep "pool bottom" and have added definition of "the lowest area in the pool" where the term first appears in line 61. "shrink" is revised as suggested by the reviewer and "jet" is revised to "high-speed flow".

Line 278 As mentioned in the general comments above, the hydraulic jump is not a regime, it extends

throughout the water column and it has flow velocities close to zero in its flow recirculation cell close to the surface.

Revised to "the flow recirculation cell near the water surface".

Lines 281-282 There is no such thing as a surface jump.

Revised to "the recirculation cell near the water surface".

Line 283 What does "the drop of flow velocity can be found in sections X0-6 and X0+2" mean?

Revised to "the flow velocity decreases in the sections x0-6 and x0+2".

Line 294 TKE is not turbulence intensity. The latter is dimensionless and is normally estimated as the RMS of turbulent velocity fluctuations normalized by the mean longitudinal velocity. Replace "overlaps" with "coincides".

The "turbulent intensity" has been revised to TKE. Done.

Lines 294-295 "high flow velocities in the upstream area of the step limits the development of turbulence". This conclusion cannot be drawn because correlation does not indicate causation, but moreover, high mean flow velocity generally coincides with low TKE, and vice versa, because where TKE is high, it has been extracted from the energy of the main flow. The two flow properties are two sides of the same coin and it is incorrect to say one causes the other.

The sentence has been removed.

Lines 297 and 299, etc. as mentioned in the general comments, this region is not the wake of the step stones.

Revised to "step toe/toe of the step".

Lines 299-300 etc. as mentioned in the general comments, the jump is neither at the surface nor above the jet.

Revised to "recirculation cell".

Line 302 Turbulent energy dissipation is not synonymous with TKE, so do not alternate between the two phrases. The highest TKE occurs near the interfaces with the jet because this is a region of high fluid shear.

"Turbulent energy dissipation" has been revised to *TKE* and the reason of high fluid shear has been added.

Line 304 why is there a decrease in flow velocity with an increase in discharge? Replace "lead" with "leads" and "limitation" with "reduction".

Because the water depth increased to a large degree according to the experimental observation. Done.

Line 311 delete "with a length of 50 cm".

Done.

Line 314 replace "as the regime of jet" with "due to the presence of the jet".

Done.

Line 316 replace "where the" with "of" and delete "shows up" and "to the".

Done.

Line 319 Insert "pool" between "negative" and "slope".
    Done.

Line 332 replace "vortex" with "turbulent".
    Done.

Lines 335-336 the "vortex stretched across the entire channel width near the surface is the flow recirculation cell of the hydraulic jump, and the "discrete streamwise streaky vortexes close to the bed" are flow recirculation cells attached to the toe of the step formed by the separation of the jet from the step face.
    Revised to "…as a combination of flow recirculation cell of the jump that stretches across the entire channel width near the water surface and discrete streaky vortexes attached to the step toe close to the bed.".

Line 340 The thickness of the hydraulic jump is the flow depth. Line 341 delete "in the pool". The "jump regime" is the flow recirculation cell of the hydraulic jump. Replace "fades away" with "loses identity".
    "The thickness of the hydraulic jump" is revised to "The thickness of the recirculation cell at water surface". The rest suggestions have been accepted.

Lines 343-345 I suggest for final sentence "On the negative slope, coherent structures mainly follow protruding grains (micro-scale bed structures but do not show streaky features as they do upstream of the step, even though the grain sizes are similar."
    Accepted.

Line 352 "point of flow separation of the jet from the step face" is better. However, the separation of the jet from the step face has not been properly described and needs a detailed description in section 3.1.1.
    Accepted. We have added "The step crests are also the points of separation of the jet from the step face in the three sections." In lines 282-283 in section 3.1.1.

Lines 353 and 372 Replace "connection" with "contact".
    Done.

Line 355 Insert "of the jet" between "Impinging point" and "in the pool".
    Done.

Line 356 replace "at the locations more downstream" with "at locations further downstream".
    Done.

Lines 357-359 Explain
    We added "This is related to the lower water depth in the pool but higher flow velocity of the jet at $Q =$ 43.6 L/s (Fig. 4-5)."

Line 370 replace "shear stress is further concentrated" with "The highest values of shear stress occur".
    Done.

Line 371 replace "top elevation" with "maximum height" and "influence" with "influences".

Done.

Line 374 Replace "configuration" with "shape".

Done.

Lines 376-377 insert "on" between "and" and "some". Delete "comparing with surrounding grains".

Done.

Line 384 replace "keeps increasing" with "increases".

Done.

Line 385 replace "enhanced" with "increased".

Done.

Line 388 replace "turns the" with "changes". How can the lift force be downwards? This does not make sense.

Done. Because only the protruding part of the step stones were sampled for the lift force (force component in Z direction) and gravity of the flow is counted in the lift force.

Line 390 replace "turns" with "changes direction to".

Done.

Line 391 insert "of magnitude" between "orders" and "smaller".

Done.

Lines 391-395 Only lift and drag forces were mentioned in the methods and they were not given as x and z components. In other words, this x, y and z coordinate system for forces has not been mentioned before. The "bank stones" have also not been mentioned before. Insert "of any component" after "the y component of flow force has the largest magnitude". Why is the transverse force component greater at the banks? Replace "enhanced" with "increased". Replace "concentrates" with "is greatest".

The definition that drag and lift forces were the components of flow forces in X and Z directions has been added in lines 255-256.

The bank stones are defined in lines 111-112. The greater transverse flow force at the banks was related to the stone shapes. No. 2 to 5 stones were more bilateral symmetry than No. 1 and 6 stones, leading to smaller resultant force in the transverse direction at the stones between the bank stones. The editorial suggestions have been accepted.

Line 401 How and why do C D and C L vary, and what does this mean?

The variation has been described in the paragraph following this line. The main finding is that $C_D$ varied in a more limited range than $C_L$, which also showed larger magnitude. The variation of $C_D$ and $C_L$ results from the variation of the object's resistance to the local flow in streamwise and vertical directions respectively (Eqs. 5-6). Both the dynamic pressure and shear stress on the step stones vary during flow increase, which might lead to the variation of drag/lift force (Fig. 9 and 10). Flow velocity also varies when the flow

approaches to the step (Fig. 4 and A9). This difference between $C_D$ and $C_L$ of the step stones indicates that the combined effect of fluid force and velocity is more sensitive to the flow increase in vertical direction than the streamwise direction for the step stones.

Line 404 What does "keystone" mean here and why is stone 4 the keystone? Explain in methods.
    The information has been added in lines 110-111 (see our reply to the comment on Line 110 of Referee #1).

Line 416 Replace "distinguished" with "the well developed".
    Done.

Line 419 replace "transverse inconsistencies of" with "a 3D".
    Done.

Line 421-423. This is incorrect. The vortexes (flow recirculation cells) at the step toe occur downstream of the centres of the step stone and the high speed regions between the vortices occur downstream of the contacts between step stones. Also, whether the jet impinges on the bed or not is more related to momentum than kinetic energy.
    The description here is correct. See Fig. S12 and our reply to your comment for Lines 273-275. "Kinetic energy" has been replaced by "momentum" as the latter is a vector that includes the direction of flow velocity.

Line 428 The jet that does not impinge on the bed at the base of the pool is not a surface jet. It still has a flow recirculation cell above it and is a classic jump. Surface jets are associated with oscillatory jumps (standing waves).
    The "surface jet" here was revised to "surface flow". We followed the definition of surface flow regimes, including breaking surface wave, surface wave and surface jet, by Wu and Rajaratnam (1998).

Lines 429-432 Delete.
    Done.

Lines 435-438 This is incorrect. Delete up to and including "wakes vortex (Fig. 4)".
    We are aware of the misuse of hydraulic concepts here and have revised the description as the segmentation of flow velocity.

Lines 438-439 "intense mid-profile fluid shearing within the hydraulic jump and between the flow recirculation cells at the step toe and the jet plunging over the step face generates high TKE" is better.
    Accepted.

Lines 448-449 Why is this noteworthy?
    The specific location of the highest level of energy transformation and dissipation has been illustrated.

Figure 13. The symbols are illegible.
    Revised.

Lines 463-465 This does not make sense. Also, insert "at 49.9 l/s" between "examined" and "(Fig. 13)".

Removed and revised.

Line 467 Insert "It is" before "Worth noting".

Done.

Line 474 insert "over a 2D step" between "vertical drop" and "with the".

This would be unnecessary as "step-pool" appears at the beginning of the sentence.

Line 486 Replace "turbulence with "TKE". Replace "visualized" with "demonstrated".

Done.

Line 487 Insert "up to a discharge of 43.6 l/s" after "flow increased".

Done.

Line 488 The flow recirculation cell of the hydraulic jump expanded.

"jump regime" has been revised to "flow recirculation cell of the hydraulic jump".

Line 492 Replace "variation patterns" with "response".

Done.

Line 493 Replace "result" with "results". The change in jet penetration angle was not mentioned in the results.

Done. A sentence "The jet penetration angles into the pool decrease in all the three sections as the flow rate and water depth increase at 49.9 L/s from 43.6 L/s." has been added in lines 294-295.

Line 494 replace "leaves" with "creates".

Done.

Line 495 This contradicts lines 296-297 which state that the "wake" vortices have high TKE.

Revised to "low flow velocity and high turbulence".

Line 499 More accurately, the jet angle and momentum influence both vortex dimensions and pool scour.

The point here is mainly on the relation between step architecture and the number and locations of the vortices attached to the toe of the step. The last sentence of this paragraph has been revised to "This is related to the stable architecture of the step structure, which determines the distribution of jet angle and momentum at the step crest."

Line 500 replace "distributed" with "located".

Done.

Line 503 replace "on the surrounding hydraulics in the pool" with "and pool hydraulics".

Done.

Line 505 replace "get mixed" with "lose their identity".

Done.

Line 508 Replace "on balance" with "in summary".
   Done.

Line 512 Replace "distribution" with "distributions".
   Done.

Lines 531 Delete "Considering that the gravity of the step stones does not change".
   Done.

Line 547 replace "tolerate" with "experience". Is it static pressure? I would have thought dynamic water pressure would be transmitted through the pores to the bases of the stones.
   Done.
   The majority of water pressure should be static pressure as the flow velocity through the pores is much lower than that above the bed surface. To be more accurate, we use "water pressure" here and has revised "static pressure" to "pressure" in this sentence.

Line 571 what does "in an integral form" mean?
   The phrase has been removed. The flow recirculation cell near the water surface shows the coherent structure as a whole comparing to the recirculation cells formed at the step toe that are separated by high-speed flows (Fig. 8 and S12). But the point here is on the segmentation of hydraulics in the pool, so this phrase can be removed.

Line 573 and onwards. Energy dissipation rate and TKE are not synonymous. I suggest sticking to TKE, which is precise because it's what you have presented.
   The sentence has been revised to "Second, the distribution of *TKE* in the pool is highly non-uniform, with the concentration of flow energy transformation and dissipation upstream of the negative slope in the pool." The flow energy transformation and dissipation refer to the results in Fig. 7

Line 576 replace "the expansion of" with "an increase in".
   Done.

Line 578 replace "restrained" with "small". Replace "enhanced" with "greater".
   Done.

Lines 578-580 Dynamic pressure, shear stress and form drag: No, delete.
   The original sentence has been replaced by the sentence: "the step experiences the lowest dynamic pressure but highest shear stress in the step-pool unit."

Lines 581-582 What are the implications of the variation in lift force? Replace "varying range" with "variation".
   The implication is described in lines 539-547. As it is an assumption that needs to be further verified in future study, the implication is not written in the conclusions.

Done.

Line 585 replace "to" with "of".

Done.

**References for replies**

Bayon, A., Valero, D., García-Bartual, R., & López-Jiménez, P. A. (2016). Performance assessment of OpenFOAM and FLOW-3D in the numerical modeling of a low Reynolds number hydraulic jump. *Environmental Modelling & Software, 80*, 322-335.

Bayon, A., Toro, J. P., Bombardelli, F. A., Matos, J., & López-Jiménez, P. A. (2018). Influence of VOF technique, turbulence model and discretization scheme on the numerical simulation of the non-aerated, skimming flow in stepped spillways. *Journal of hydro-environment research*, *19*, 137-149.

Chen, Y., Liu, X., Gulley, J. D., & Mankoff, K. D. (2018). Subglacial conduit roughness: Insights from computational fluid dynamics models. *Geophysical Research Letters, 45*(20), 11-206.

Chen, Y., Bao, J., Fang, Y., Perkins, W. A., Ren, H., Song, X., ... & Scheibe, T. D. (2022). Modeling of streamflow in a 30 km long reach spanning 5 years using OpenFOAM 5. x. Geoscientific Model Development, 15(7), 2917-2947.

Chiu, C. L., Fan, C. M., & Tsung, S. C. (2017). Numerical modeling for periodic oscillation of free overfall in a vertical drop pool. *Journal of Hydraulic Engineering, 143*(1), 04016077.

Comiti, F., Andreoli, A., & Lenzi, M. A. (2005). Morphological effects of local scouring in step-pool streams. *Earth Surface Processes and Landforms*, *30*(12), 1567-1581.

Eltner, A., Kaiser, A., Castillo, C., Rock, G., Neugirg, F., & Abellán, A. Image-based surface reconstruction in geomorphometry-merits, limits and developments. *Earth Surface Dynamics, 4*(2), 359-389, 2016.

Finnemore, E. J., & Franzini, J. B. (2002). *Fluid mechanics with engineering applications*. McGraw-Hill Education.

Hohermuth, B., Kramer, M., Felder, S., & Valero, D. (2021). Velocity bias in intrusive gas-liquid flow measurements. *Nature communications, 12*(1), 1-9.

Lenzi, M. A. Marion, A., Comiti, F., & Gaudio, R. 2002. Local scouring in low and high gradient streams at bed sills. *Journal of Hydraulic Research*, *40*(6), 731-739.

Morovati, K., Homer, C., Tian, F., & Hu, H. (2021). Opening configuration design effects on pooled stepped chutes. *Journal of Hydraulic Engineering, 147*(9), 06021011.

Saad. Y. (1996). Iterative Methods for Sparse Linear Systems. The PWS Series in Computer Science. PWS Publishing Company, ISBN 9780534947767. URL: http://books.google.com/books?id=jLtiQgAACAAJ.

Tmušić, G., Manfreda, S., Aasen, H., James, M. R., Gonçalves, G., Ben-Dor, E., ... & McCabe, M. F. Current practices in UAS-based environmental monitoring. *Remote Sensing, 12*(6), 1001, 2020.

Valero, D., Bung, D. B., & Crookston, B. M. (2018). Energy dissipation of a Type III basin under design and adverse conditions for stepped and smooth spillways. *Journal of Hydraulic Engineering*, *144*(7), 04018036.

Wohl, E. E., & Thompson, D. M. (2000). Velocity characteristics along a small step–pool channel. *Earth Surface Processes and Landforms*, *25*(4), 353-367.

Wu, S., & Rajaratnam, N. (1998). Impinging jet and surface flow regimes at drop. *Journal of Hydraulic Research*, *36*(1), 69-74.

Yakhot, V., & Orszag, S. A. (1986). Renormalization group analysis of turbulence. I. Basic theory. *Journal of scientific computing, 1*(1), 3-51.

Yakhot, V., & Smith, L. M. (1992). The renormalization group, the ε-expansion and derivation of turbulence models. *Journal of scientific computing, 7*(1), 35-61.

Zhang, C., Xu, M., Hassan, M. A., Chartrand, S. M., & Wang, Z. (2018). Experimental study on the stability and failure of individual step-pool. *Geomorphology*, *311*, 51-62.

Zhang, C., Xu M., Hassan, M., Chartrand, S. M., Wang, Z., & Ma, Z. (2020). Experiment on morphological and hydraulic adjustments of step-pool unit to flow increase. *Earth Surface Processes and Landforms*. DOI: https://doi.org/10.1002/esp.4722.

Zimmermann A. E., Salleti M., Zhang C., Hassan M. A.: Step-pool Channel Features, in: Treatise on Geomorphology (2nd Edition), vol. 9, Fluvial Geomorphology, edited by: Shroder, J. (Editor in Chief), Wohl, E. (Ed.), Elsevier, Amsterdam, Netherlands, https://doi.org/10.1016/B978-0-12-818234-5.00004-3, 2020.

---

## Referee Report (RR1)

**Review of Zhang et al. "Hybrid Modeling on 3D hydraulic features of a step-pool unit." – Version 2**

**Keith Richardson**

This revision is a great improvement on the initial submission. The entire paper is more clearly written, more technically accurate and more authoritative. I would like to congratulate the authors again on this paper. This paper contains some novel and excellent data and some important conclusions and it deserves and needs to be published. I would also like to thank the authors for responding positively and constructively to my comments. However, that said, I still have some concerns over terminology and the description of results – although this aspect shows a distinct improvement on the first draft – and it needs further revision, although much more minor than the first revision. I shall discuss this concern in detail first, and then list general comments.

The word "regime" is misused at several places. Line 230; the "water surface regime" probably did not change during the experiment (it would always have been the plunging jet regime), so it's not clear what was measured here. Line 235; the jet and the jump are flow structures or components or regions of flow, not flow regimes. Line 284; "Plunging jet" (Wu & Rajaratnam 1998) is a flow regime and refers to a type of flow over a vertical drop, consisting of a jet plunging into a hydraulic jump. This regime contrasts with the surface flow regime over a drop in which the jet travels horizontally and enters an oscillatory jump. However a jet is not a flow regime so it is incorrect to say that "flow accelerates... before plunging into the pool as the jet regime."

Lines 284-286. This is a misinterpretation. This description gives the impression that the region of high flow velocity moves from the upper part of the flow to the lower part. In fact, the position of the region of high flow velocity remains unchanged in the upper part of the flow and the step stones protrude upwards into it.

Line 293. The word "wake" is still misused. The region at the toe of the step cannot be described as a wake. See my comments on the use of "wake" in my review on the first draft.

Lines 308-309. It is not evident from Figure 5 that high velocity regions occur at the low points in the step crest.

Line 339. Why does flow velocity decrease with an increase in discharge? This is counterintuitive.

Line 366 and Figure 8. Surely these are time-averaged turbulent coherent structures, not instantaneous ones? Although the authors state that the structures illustrated in Figure 8 are instantaneous, the description that follows in the text is of time-averaged structures, and certainly it is only the persistent, time-averaged structures that are relevant to the study.

Line 367. What does "streamwise coherent structures" mean? This is vague. What type of structure are they?

Lines 23, 369 and 444. The vortices at the step toe are described as "streaky". This adjective is applied to structures that are highly elongated in the streamwise direction, which these vortices are not.

Lines 373-375. This description of curvature in the recirculation cell of the hydraulic jump correlating with the positions of step toe vortices is not convincing. If you decide to stick with it, I recommend replacing "upper bends" and "downward bends" with "convex-upwards curvature" and "convex-downwards" curvature respectively.

Lines 400-404. I suggest that, rather than the shape and maximum height of the step stones influencing the distribution of shear stress, it is the presence of the step toe vortices. The downstream curvature of the high shear stress zone you mention on the upper faces of the step stones matches the boundary of the step toe vortices as shown in Figure 8.

Lines 429-435. Is it necessary to present drag and lift coefficients as well as drag and lift forces? It does not seem to add anything to the interpretation of step failure and stability.

Line 455. This is a misinterpretation of the surface flow regime. These experiments did not generate the surface flow regime because there was always a hydraulic jump across the entire flume width. However, I agree that the fact that in some longitudinal sections the jet did not impinge on the pool bottom indicates that (1) the Q=49.9 l/s discharge is probably transitional between the impinging jet and surface flow regimes, and (2) this indicates that the flow regime over a step is not necessarily binary (i.e. either impinging jet or surface flow).

Lines 486-487. This statement is not apparent from Figure 13. Also, why do the values for the step toe vortex not drop to zero downstream of the end of the vortex?

I shall now list general comments.

Line 20. Replace "by" with "from".

Line 24-25. Replace "with comparable capacity" with "of comparable magnitude".

Line 26. Replace "increase" with "expansion".

Line 27. Delete "as". Place "grain clusters" in parentheses.

Lines 34 and 37. "Benefits" and "advantages" are subjective terms.

Line 45. What type of products?

Line 56 Semi-colon rather than comma required after "morphology".

Line 65. Delete "(the lowest area in the pool)".

Line 89. Replace "experimental" with "physical".

Line 107. Is the superscript "2" required on "$px^2$"?

Line 111. It would be helpful to report $D_{100}$ and $D_{84}$ as well.

Line 112. What discharge is the Froude scaling based on, and why was Froude scaling (rather than some other scaling method) chosen?

Line 117. Change "the step" to "the model step".

Figure 2. "L/s" should be "l/s".

Line 125. What are T runs?

Line 133. Insert "for discharges < 56.1 l/s" after "CIFR T2".

Lines 152-153. Insert "all" after "characterize". Replace "e.g." with "including for example".

Lines 184-185. "The gravity model was activated…" Is this sentence necessary?

Line 223. 2 Hz and 30 s are a low sampling frequency and period respectively. This sampling strategy cannot capture high frequency velocity fluctuations. What are the implications of this for the results?

Line 244. Replace "in three directions" with "and the subscripts denote the respective coordinate axis".

Line 246. Delete "in three directions".

Lines 247-248. "Q-criterion" notation should be consistent. Replace "calculate and visualize" with "identify". How was the threshold value of 1200 selected?

Line 253. Replace $P_s$ with $P_d$.

Line 267 and 270. Replace "taken" with "occupied".

Lines 281. Insert "mean" before "flow velocity" in title.

Line 289-290. Change "the" to "a" in "the recirculation cell". Insert "associated with a hydraulic jump" after "water surface". Insert "associated with an attached vortex" after "step toe". Delete "sliding".

Line 296. Replace "reduction" with "contraction".

Line 308. Replace "locate" with "are located".

Lines 319-321. Confusing sentence.

Line 329. Replace "with" with "as those in".

Line 330. Replace "are" with "is". Replace "at a much lower level if compared with" with "lower than in".

Line 331. I suggest replacing this sentence with "These areas of low TKE coincide with areas of high mean flow velocity". Indeed, would it be simpler to state that Mean flow velocity and TKE are inversely correlated in general?

Line 332. Insert "Like mean flow velocity," at the start of this sentence.

Line 333. Replace "above the bed surface" with "within the attached vortices".

Line 334-335. Move "both" to after "recirculation cells". Insert "at the" after ""water surface and".

Line 335-336. Replace "and much higher TKE is contained…" with "although that in the recirculation cell above the jet is much higher".

Line 346, 353, 368. Replace "at the downstream area of the" with "downstream of".

Line 366. Replace "In the upstream area of" with "upstream of".

Line 368. Replace "as a combination of" with "in both the".

Line 369. Insert "the" after "water surface and".

Line 372-373. Replace "A near bed vortex starts" with "The vortices at the step toe start". Replace "its" with "their". Delete "to the" and "direction". Replace "vortex vanishes" with "vortices vanish".

Line 376-377. Replace "do not show streaky features as they do" with "are not elongated in the streamwise direction as they are".

Line 387. Replace "flow" with "jet".

Line 399. Delete "The step stones bear the highest level of shear stress in the step-pool unit". This sentence is basically repeated in the next sentence.

Lines 403-404. Replace "The edges of the" with "There is also a". Replace "in the back sides" with "on the downstream faces". Replace "show clear" with "with a".

Line 405. Replace "flow" with "jet".

Line 445. Replace "different" with "as distinct".

Line 448. Replace "will be followed by the" with "generate".

Line 452. Delete "eventually".

Line 465. Insert "of the profile" after "middle".

Line 467. Replace "of the flow for" with "within".

Line 470. Move "both" to after "distributions of".

Line 474. "Appealingly" is not appropriate.

Line 478. Insert "section" before "-averaged".

Line 486. Insert "section" before "integral".

Line 489. Insert "cell" after "recirculation".

Line 490. Replace "one" with "vortex". Insert "at 43.6 l/s" after "near the water surface".

Line 491. Replace "below" with "of".

Line 492. Replace "intensification of" with "increase in".

Line 494. Replace "is enlarged" with "increases". Insert "at high discharges" after "pool development".

Line 499. Replace "concentrates" with "occurs".

Line 501. Replace "local" with "pool".

Line 504. Replace "a drop as well as the pool at the downstream" with "an artificial 2D drop"

Line 513-514. Replace "in the pool" with "of the step".

Line 520-522. Is this noteworthy?

Line 526. "The grain clusters at the pool bottom…". This sentence is a tautology.

Line 531. How is the distribution of micro-bedforms at the pool bottom affected by the jet?

Line 548. The variation in the direction of the lift force is unlikely to be "sudden".

---

## Referee Report (RR2)

Dear authors,

        Thanks for the detailed responses to my comments and observations. The article
        is better than the first version, especially in terms of writing and
        clarifications. I enjoyed reading it and analyzing the results.

I still think some aspects of the CFD implementation and description in the article are
still missing or need more clarification in the text (some are well explained in the
responses). Also, some comments were not addressed, but the response said they were. For
example,

line 78: "In the flume experiment of Zhang et al" In that article there is more than
one, so using "the" is incorrect here.
Removed.

but (new) line 96 says:
"in the flume experiment of Zhang et al. (2020)"

This is just an example, but this happens in other parts too.

My primary concerns are related to the CFD implementation and the impacts that it may
have on the results, especially in the magnitudes of the variables. The following three
points summarize this:

1) The complete paper is constructed around the results of highly diffusive numerical
schemes. They are first order in all cases and impact the magnitude of every single
variable, especially those related to forces and turbulence. The authors tried to
justify this in line 578 saying: "The RNG k-ε turbulence model and first-order momentum
advection were applied in the CFD simulation. Such settings ensured the computational
stability for the flow over the highly complex bed surface of a step-pool unit but could
only provide time-averaged results"
While it is true that the configuration will be more stable, the results are impacted by
this setup. This should be acknowledged in the paper. As it is now, it seems to be an
advantage rather than a loss in accuracy. For CFD studies, we want second order accuracy
in any simulation.
The problem with first-order accuracy is that we don't know if the magnitudes are under
or over-estimated (most likely underestimated because velocity fluctuations almost
disappear).

2) As expressed in the first review, the distance between the inlet and the first step
is (based on the figures) 10 to 20 cm. Boundary conditions are critical in a CFD
simulation. A short distance with a uniform velocity profile does not represent the
inlet of a step-pool unit. The authors justify this by mentioning the work of Wohl and
Thompson (2000), but they had developed turbulence when working in the field. Also,
adding 2 to 5 cm is still not enough. I mentioned this because I have experience
simulating step-pool sequences using LES and noticed that the flow variables in the
first unit are different than the 2nd and 3rd. Actually, the first unit may not be used
to calculate average properties because it is the one that helps in developing the flow
structure in the subsequent units. Then the authors said that "This is supported by the
fact that the streaky coherent structures already formed at the downstream of protruding
grains upstream of the step in this study". This is not an accurate statement because it
is a result of the model. You will always have some flow structure, but you can only
determine if it is valid if you have measurements.

3) There is only one step-pool unit in the experiment. This is not representative of
reality because they are sequences most of the time.

So, when considering the cumulative effects of the different experimental
configurations, 1st order + boundary conditions + single step-pool unit,  I don't know
if the results are a good representation of what was happening in the actual experiment.

I believe all these three points must be acknowledged and explained earlier in the
article and not leave them for a small discussion at the end of the text. I would place
them in section 2. This is a good study and will certainly be a reference for future
studies, so these simplifications and decisions must be highlighted. Subsequent studies
can identify these gaps and improve upon them. There are no problem by saying that
simplifications have been done, actually that would be an advantage because they can be
clearly identified.

Finally, some responses are very useful but were only included in the line by line responses and not in the actual article. For example, the comment about convergence criteria, boundary conditions for k and epsilon, etc. Make sure that the answers are included in the text too. My comments are intended for the general audience.

---

## Author Response (AR2)

**Responses to associate editor and reviewers**

**Associate Editor**

Comments to the author:
Dear authors,

thanks for the revised paper and apologies that it took some time to secure the second review.

I have now received the assessment of both original reviewers, and they are very happy about how you responded to their comments. Reviewer #1 (Richardson) is happy with the content, and makes a large number of editorial and language suggestions. Reviewer #2 makes three important points. Although these points have been addressed in the discussion, the reviewer comments that they contain fundamental caveats. Essentially, you have to pathways to deal with that: (1) either, you address them by improving the numerical scheme and set up, or (2) be up-front with these caveats and reason them. The former would be quite a lot of work, and the latter has been explicitly suggested by the reviewer. If you want to take approach 2, please add some material to the method description of the CFD, early in the paper. Highlight the potential caveats and provide a reasoning as to why you choose your approach (this could be, for example, a statement that a high order approach is numerically inhibitive). In the discussion, you could add a few sentences on potential future improvements.

I have decided on major revisions, as I would like to have the option to send the revised paper to reviewer #2 again.

Please provide a detailed rebuttal to the comments.

Thanks a lot and I am looking forward to seeing your revised manuscript.

With best wishes,
Jens Turowski

Dear Prof. Jens Turowski,
Thanks a lot for the valuable comments and suggestions!

All the remarks of both yours and the reviewers have been addressed and replied in this document. We agree to take the second approach suggested by you. Notations for the potential caveats have been added in Section 2 (e.g., in Lines 195-196). Potential future improvements have also been added to Section 4.5 in the discussion. Besides, the appendices have been moved to the supplement file in order to reduce the length of the manuscript. This may make the paper easier to follow for the readers.

All the best,
Chendi Zhang
On behalf of all co-authors

**Report #1**

Keith Richardson

This revision is a great improvement on the initial submission. The entire paper is more clearly written, more technically accurate and more authoritative. I would like to congratulate the authors again on this paper. This paper contains some novel and excellent data and some important conclusions and it deserves and needs to be published. I would also like to thank the authors for responding positively and constructively to my comments. However, that said, I still have some concerns over terminology and the description of results – although this aspect shows a distinct improvement on the first draft – and it needs further revision, although much more minor than the first revision. I shall discuss this concern in detail first, and then list general comments.

Thanks again for the detailed comments and suggestions! They contribute greatly to improve the clarity of the manuscript.

The word "regime" is misused at several places. Line 230; the "water surface regime" probably did not change during the experiment (it would always have been the plunging jet regime), so it's not clear what was measured here. Line 235; the jet and the jump are flow structures or components or regions of flow, not flow regimes. Line 284; "Plunging jet" (Wu & Rajaratnam 1998) is a flow regime and refers to a type of flow over a vertical drop, consisting of a jet plunging into a hydraulic jump. This regime contrasts with the surface flow regime over a drop in which the jet travels horizontally and enters an oscillatory jump. However, a jet is not a flow regime so it is incorrect to say that "flow accelerates… before plunging into the pool as the jet regime."

The misuse of the word "regime" has been checked and revised throughout, not limited to the three places listed in this comment.

For line 230, "water surface regime" has been revised to "the boundary between the jet and recirculation cell at the water surface". This boundary was extracted from the pictures taken by the top camera during the experiment and compared with the simulated results. Accordingly, the caption of Fig. S3 has been revised to "…(b) upstream edge of the recirculation cell at the water surface from the top view".

Line 235 has been revised to "…the boundaries between the jet and jump components…"

In line 284, "as the jet regime" has been revised to "as a jet feature".

Lines 284-286. This is a misinterpretation. This description gives the impression that the region of high flow velocity moves from the upper part of the flow to the lower part. In fact, the position of the region of high flow velocity remains unchanged in the upper part of the flow and the step stones protrude upwards into it.

The sentence has been revised to "The highest flow velocity in the vertical profile at the crests of step stones mainly exists near the stone surface (Fig. 4), different from the vertical profile of flow velocity upstream of the step where the flow velocity is higher near the water surface (at 43.6 L/s) or shows relatively uniform distribution (at 49.9 L/s)." Our point is to illustrate the difference in vertical distribution of flow velocity.

Line 293. The word "wake" is still misused. The region at the toe of the step cannot be described as a wake. See my comments on the use of "wake" in my review on the first draft.

"as a result of wake turbulence" has been deleted.

Lines 308-309. It is not evident from Figure 5 that high velocity regions occur at the low points in the step

crest.

The sentence has been revised to "The x0-6 section, which is located at the step crest, shows that the highest flow velocity is located above the No. 2 and 3 step stones (Fig. 3d) which are among the lowest points within the step crest."

Line 339. Why does flow velocity decrease with an increase in discharge? This is counterintuitive.

The water depth also increased. It may not be appropriate to say that the flow velocity decrease as the section-averaged flow velocity was not calculated. The point is the highest flow velocity decreases. To be more accurate, the sentence has been revised to "The increase of flow rate from 43.6 L/s to 49.9 L/s leads to…"

Line 366 and Figure 8. Surely these are time-averaged turbulent coherent structures, not instantaneous ones? Although the authors state that the structures illustrated in Figure 8 are instantaneous, the description that follows in the text is of time-averaged structures, and certainly it is only the persistent, time-averaged structures that are relevant to the study.

Based on the turbulence mode (k-epsilon RNG) used in this study, the numerical simulation was based on Reynolds-averaged Navier–Stokes equations, which means that the turbulence modeling is based on time-averaged scheme. However, transient solver was used in the simulation, so the time derivative term was solved in the N-S equations. In this case, the flow structures based on $Q_{criterion}$ are instantaneous for advection part of the flow, but time-averaged in terms of turbulence modeling. If time-averaged velocity is used to compute the turbulent structures, many small-scaled vortices due to the rough elements may not be displayed. Thus, the results shown in Fig. 8 are instantaneous ones at one moment rather than time-averaged ones. We did not calculate the time-averaged $Q_{criterion}$ of the exported data from the solver.

Line 367. What does "streamwise coherent structures" mean? This is vague. What type of structure are they?

They refer to the coherent structures following the protruding grains. The point here is to illustrate the general orientation of these coherent structures. The sentence has been revised to "…coherent structures are mainly streamwise and located near the bed, particularly downstream of protruding grains at the flow rate higher than 12.4 L/s."

Lines 23, 369 and 444. The vortices at the step toe are described as "streaky". This adjective is applied to structures that are highly elongated in the streamwise direction, which these vortices are not.

Revised to "the discrete vortices extending along the streamwise direction attached to the step toe close to the bed" in Line 369. "streaky structures" in Line 444 is changed to "structures along the streamwise direction". The "streaky" in Line 23 is replaced by "streamwise".

Lines 373-375. This description of curvature in the recirculation cell of the hydraulic jump correlating with the positions of step toe vortices is not convincing. If you decide to stick with it, I recommend replacing "upper bends" and "downward bends" with "convex-upwards curvature" and "convex-downwards" curvature respectively.

The reviewer is right that the correlation is not that evident. The sentence has been removed.

Lines 400-404. I suggest that, rather than the shape and maximum height of the step stones influencing the distribution of shear stress, it is the presence of the step toe vortices. The downstream curvature of the high shear stress zone you mention on the upper faces of the step stones matches the boundary of the step toe

vortices as shown in Figure 8.

We agree that the downstream curvatures of the high shear stress zones on the step stones match well with the shapes of the step toe vortices. This point has been added in Lines 416-417. However, the shapes of the vortices are also influenced by the shape and maximum height of the step stones, which affect the separation of jet from the step stones. So the description linking the shape and maximum height of the step stones and the distribution of shear stress is kept.

Lines 429-435. Is it necessary to present drag and lift coefficients as well as drag and lift forces? It does not seem to add anything to the interpretation of step failure and stability.

Indeed, the drag and lift forces show similar trend with drag and lift coefficients. The figure for drag and lift coefficients has been moved to Section S2 in the supplement as Fig. S14 and the text has been shortened in Lines 437-440.

Line 455. This is a misinterpretation of the surface flow regime. These experiments did not generate the surface flow regime because there was always a hydraulic jump across the entire flume width. However, I agree that the fact that in some longitudinal sections the jet did not impinge on the pool bottom indicates that (1) the Q=49.9 l/s discharge is probably transitional between the impinging jet and surface flow regimes, and (2) this indicates that the flow regime over a step is not necessarily binary (i.e. either impinging jet or surface flow).

The sentence has been revised to "However, the 3D flow structures exhibit that the jet does not impinge the pool bottom in some longitudinal sections (Figs. 4-5)."

We agree with the two interpretations provided by the reviewer and have added them in Lines 463-466.

Lines 486-487. This statement is not apparent from Figure 13. Also, why do the values for the step toe vortex not drop to zero downstream of the end of the vortex?

The greatest difference between the *TKE* values for the recirculation cells near the water surface and at the step toe is less than 4 times (Fig. 12d, Figure 13 in last version is Figure 12 in this version) downstream of the contraction section. So there is no difference of order between the *TKE* for the recirculation cells at different locations in the pool.

The streamwise distance shown in Fig. 12 does not cover the whole pool length. The recirculation cells at the step toe and the water surface merge on the negative slope where the algorithm used to separate them does not work anymore. The merge of the recirculation cells already appears in some vertical lines in the cross sections near the pool bottom. So the calculation was only conducted for the streamwise distance where the recirculation cells are clearly separated by the jet. As a result, the calculation stopped a bit upstream of the pool bottom. This is why the decrease of *TKE* for the step toe vertex was not shown in Fig. 12. This information of the calculation method has been added in Lines 281-283 in Section 2.4.

I shall now list general comments.

Line 20. Replace "by" with "from".

Done.

Line 24-25. Replace "with comparable capacity" with "of comparable magnitude".

Done.

Line 26. Replace "increase" with "expansion".
Done.

Line 27. Delete "as". Place "grain clusters" in parentheses.
Done.

Lines 34 and 37. "Benefits" and "advantages" are subjective terms.
Line 34 has been revised to "This bed structure has been reported to contribute to providing…"
"With these advantages" in line 37 has been removed.

Line 45. What type of products?
Revised to "topographic reconstructions".

Line 56 Semi-colon rather than comma required after "morphology".
Revised.

Line 65. Delete "(the lowest area in the pool)".
Done.

Line 89. Replace "experimental" with "physical".
Done.

Line 107. Is the superscript "2" required on "px 2 "?
Yes. Camera resolution needs a unit of area.

Line 111. It would be helpful to report D100 and D84 as well.
$D_{100}$ ($D_{max}$) and $D_{84}$ were 140 mm and 50 mm, respectively. Added in Line 112.

Line 112. What discharge is the Froude scaling based on, and why was Froude scaling (rather than some other scaling method) chosen?
The peak discharge for the prototype is 12.6 $m^3$/s (Zhang et al., 2020), between the peak discharges of 10.4 $m^3$/s on 14 September 1994 in Rio Cordon, Italy (Lenzi, 2001) and 14.6 $m^3$/s in the Erlenbach on 20 June 2007 in Switzerland (Turowski et al., 2009), respectively.
Froude scaling based on gravity similarity was selected because gravity played the leading role in high-gradient mountain streams rather than other forces (e.g., viscous force, pressure or surface tension).

Line 117. Change "the step" to "the model step".
Done.

Figure 2. "L/s" should be "l/s".
Liter can be written in both "L" and "l" (e.g., in the Guide for SI unit system of the National Institute of Standards and Technology, US). Furthermore, "L" is preferred sometimes to avoid the risk of confusion between "l" and "1".

Line 125. What are T runs?

"CIFR T runs" are the names used in Zhang et al. (2020). The reference has been added here. "T" is short for topography.

Line 133. Insert "for discharges < 56.1 l/s" after "CIFR T2".
Done.

Lines 152-153. Insert "all" after "characterize". Replace "e.g." with "including for example".
Done.

Lines 184-185. "The gravity model was activated…" Is this sentence necessary?
Yes. Gravity module is an independent physical module in the software.

Line 223. 2 Hz and 30 s are a low sampling frequency and period respectively. This sampling strategy cannot capture high frequency velocity fluctuations. What are the implications of this for the results?
The turbulence model (RNG $k$-$\varepsilon$) used in this study cannot capture the velocity fluctuations of high frequency. Thus, the velocity fluctuation was not reflected in the results. However, the spatial distribution rather than the temporal characteristics of hydraulics in a step-pool unit was the focus of this study. So both the low sampling frequency and short sampling period were not problems for capturing the time-averaged characteristics of hydraulics.

Line 244. Replace "in three directions" with "and the subscripts denote the respective coordinate axis".
Done.

Line 246. Delete "in three directions".
Done.

Lines 247-248. "Q-criterion" notation should be consistent. Replace "calculate and visualize" with "identify". How was the threshold value of 1200 selected?
The first "Q-criterion" refers to the visualization method while the rest refer to the variable. So they are written in different forms. "identify" has been used. The choice of threshold value was determined by trial and error to clearly present the coherent structures for the recirculation cells both at the water surface and the step toe.

Line 253. Replace Ps with Pd.
Done.

Line 267 and 270. Replace "taken" with "occupied".
Done.

Lines 281. Insert "mean" before "flow velocity" in title.
Done.

Line 289-290. Change "the" to "a" in "the recirculation cell". Insert "associated with a hydraulic jump" after

"water surface". Insert "associated with an attached vortex" after "step toe". Delete "sliding".
Done.

Line 296. Replace "reduction" with "contraction".
Done.

Line 308. Replace "locate" with "are located".
Done.

Lines 319-321. Confusing sentence.
Revised to "As the discharge increases from 43.6 L/s to 49.9 L/s, water depth increases in all the five cross sections while the areas occupied by the flows >1.8 m/s decreases in the sections x0-6 and x0+2. The vortices formed at the toe of the step expand their areas in the sections x0+2 and x0+15 at 49.9 L/s than 43.6 L/s."

Line 329. Replace "with" with "as those in".
Done.

Line 330. Replace "are" with "is". Replace "at a much lower level if compared with" with "lower than in".
Done.

Line 331. I suggest replacing this sentence with "These areas of low TKE coincide with areas of high mean flow velocity". Indeed, would it be simpler to state that Mean flow velocity and TKE are inversely correlated in general?
Good idea. Both are accepted.

Line 332. Insert "Like mean flow velocity," at the start of this sentence.
Done.

Line 333. Replace "above the bed surface" with "within the attached vortices".
Done.

Line 334-335. Move "both" to after "recirculation cells". Insert "at the" after "water surface and".
Done.

Line 335-336. Replace "and much higher TKE is contained…" with "although that in the recirculation cell above the jet is much higher".
Done.

Line 346, 353, 368. Replace "at the downstream area of the" with "downstream of".
Done.

Line 366. Replace "In the upstream area of" with "upstream of".
Done.

Line 368. Replace "as a combination of" with "in both the".
Done.

Line 369. Insert "the" after "water surface and".
Done.

Line 372-373. Replace "A near bed vortex starts" with "The vortices at the step toe start". Replace "its" with "their". Delete "to the" and "direction". Replace "vortex vanishes" with "vortices vanish".
Done.

Line 376-377. Replace "do not show streaky features as they do" with "are not elongated in the streamwise direction as they are".
Done.

Line 387. Replace "flow" with "jet".
Done.

Line 399. Delete "The step stones bear the highest level of shear stress in the step-pool unit". This sentence is basically repeated in the next sentence.
Done.

Lines 403-404. Replace "The edges of the" with "There is also a". Replace "in the back sides" with "on the downstream faces". Replace "show clear" with "with a".
Done.

Line 405. Replace "flow" with "jet".
Done.

Line 445. Replace "different" with "as distinct".
Done.

Line 448. Replace "will be followed by the" with "generate".
Done.

Line 452. Delete "eventually".
Done.

Line 465. Insert "of the profile" after "middle".
Done.

Line 467. Replace "of the flow for" with "within".
Done.

Line 470. Move "both" to after "distributions of".

Done.

Line 474. "Appealingly" is not appropriate.
Revised to "strongly".

Line 478. Insert "section" before "-averaged".
Done.

Line 486. Insert "section" before "integral".
Done.

Line 489. Insert "cell" after "recirculation".
Done.

Line 490. Replace "one" with "vortex". Insert "at 43.6 l/s" after "near the water surface".
Done.

Line 491. Replace "below" with "of".
Done.

Line 492. Replace "intensification of" with "increase in".
Done.

Line 494. Replace "is enlarged" with "increases". Insert "at high discharges" after "pool development".
Done.

Line 499. Replace "concentrates" with "occurs".
Done.

Line 501. Replace "local" with "pool".
Done.

Line 504. Replace "a drop as well as the pool at the downstream" with "an artificial 2D drop"
Done.

Line 513-514. Replace "in the pool" with "of the step".
Done.

Line 520-522. Is this noteworthy?
"It is noteworthy that" has been deleted.

Line 526. "The grain clusters at the pool bottom…". This sentence is a tautology.
The sentence has been revised into "The grain clusters at the pool bottom are mainly located where the bed is impinged by the jet, rather in the areas occupied by the recirculation cells connected to the step toe (Fig.

10 and Fig. S14 in the supplement). These grain clusters have very limited…" The point here is that the grain clusters are not distributed in the entire channel width at the pool bottom. The alternation of jet and vortices attached to step toe affects the formation of grain clusters to a large degree at the pool bottom.

Line 531. How is the distribution of micro-bedforms at the pool bottom affected by the jet?
See our reply above for line 526.

Line 548. The variation in the direction of the lift force is unlikely to be "sudden".
"sudden" has been removed.

**Report #2**

Dear authors,

Thanks for the detailed responses to my comments and observations. The article is better than the first version, especially in terms of writing and clarifications. I enjoyed reading it and analyzing the results.
Thanks for reviewing our manuscript again and the constructive comments!

I still think some aspects of the CFD implementation and description in the article are still missing or need more clarification in the text (some are well explained in the responses). Also, some comments were not addressed, but the response said they were.
For example,
line 78: "In the flume experiment of Zhang et al" In that article there is more than one, so using "the" is incorrect here.
Removed.
but (new) line 96 says: "in the flume experiment of Zhang et al. (2020)"
This is just an example, but this happens in other parts too.

The line 96 in the revision indeed has the same problem with the line 78 in the first submission and 'the' should be removed. However, line 78 was in the last paragraph of the Introduction Section while (new) line 96 is in the first paragraph of the Methods Section. They do not correspond to each other. As we replied, 'the' in line 78 was removed according to this suggestion. But the entire sentence was further removed according to the comment of the other reviewer.

My primary concerns are related to the CFD implementation and the impacts that it may have on the results, especially in the magnitudes of the variables. The following three points summarize this:
1) The complete paper is constructed around the results of highly diffusive numerical schemes. They are first order in all cases and impact the magnitude of every single variable, especially those related to forces and turbulence. The authors tried to justify this in line 578 saying: "The RNG k-ε turbulence model and first-order momentum advection were applied in the CFD simulation. Such settings ensured the computational stability for the flow over the highly complex bed surface of a step-pool unit but could only provide time-averaged results"
While it is true that the configuration will be more stable, the results are impacted by this setup. This should be acknowledged in the paper. As it is now, it seems to be an advantage rather than a loss in accuracy. For

CFD studies, we want second order accuracy in any simulation.

The problem with first-order accuracy is that we don't know if the magnitudes are under or over-estimated (most likely underestimated because velocity fluctuations almost disappear).

The loss of accuracy due to the first-order momentum advection has been added to the line mentioned by the reviewer. The future improvement has also been added to this point of limitation as "The performance of a higher order approach for step-pool features remains an open research question for future research." in Lines 594-595.

The limitation of using first-order accuracy has also been stated in Lines 195-196 as "First-order momentum advection was applied also to ensure numerical stability under the complex bed surface conditions in this study but inevitably led to the loss in accuracy."

2) As expressed in the first review, the distance between the inlet and the first step is (based on the figures) 10 to 20 cm. Boundary conditions are critical in a CFD simulation. A short distance with a uniform velocity profile does not represent the inlet of a step-pool unit. The authors justify this by mentioning the work of Wohl and Thompson (2000), but they had developed turbulence when working in the field. Also, adding 2 to 5 cm is still not enough. I mentioned this because I have experience simulating step-pool sequences using LES and noticed that the flow variables in the first unit are different than the 2nd and 3rd. Actually, the first unit may not be used to calculate average properties because it is the one that helps in developing the flow structure in the subsequent units. Then the authors said that "This is supported by the fact that the streaky coherent structures already formed at the downstream of protruding grains upstream of the step in this study". This is not an accurate statement because it is a result of the model. You will always have some flow structure, but you can only determine if it is valid if you have measurements.

We admit the relatively short distance from the inlet to the step-pool unit as a limitation to this study and has stated it in Section 4 as point 3. Mentioning the work of Wohl and Thompson (2000) was to indicate that flow turbulence would significantly decrease as the flow velocity increases over the step stones. The flow turbulence might be underestimated upstream of the step-pool unit, but such mechanism would restrict the influence of inlet setting to the hydraulics over a step-pool unit.

Thanks for sharing the experience on LES simulations for step-pools. It is important and we would pay attention to it if we get the chance to compare the performance of LES and RNG on step-pool hydraulics, especially for step-pool sequence in future.

In order to further highlight this limitation, we added "It is noteworthy that the limited distance from the inlet to the step available in the DSMs might lead to underestimated turbulence upstream of the step (see detailed discussion in Section 4.5)." in Lines 203-205. Also, a sentence "Long enough distance from the inlet to a step-pool unit is suggested for future research to better reconstruct the incoming flow turbulence for the step-pool unit." has been added in Lines 589-590.

3) There is only one step-pool unit in the experiment. This is not representative of reality because they are sequences most of the time.

It is true that most step-pool features appear as sequences in nature and that it is more physically-solid if step-pool sequences are reconstructed to study step-pool units in flume experiments. Using step-pool sequences in flume experiment would lead to great difficulty in controlling variables, especially for the replicate flume experiments that needs the same initial conditions. As a result, only one step-pool unit was inspected in the experiments of Zhang et al. (2020) in which the initial geometric conditions of the step (locations and orientations of all the step stones) needed to keep as same as possible in all the tested runs. We admit this

setting to be a limitation and has added it to Section 4.5 in the discussion Section. The flume experiment for step-pool sequence with detailed hydraulic and morphological measurements is our plan for future research when the techniques such as 3D printing to make the experiment repeatable are ready in our lab.

So, when considering the cumulative effects of the different experimental configurations, 1st order + boundary conditions + single step-pool unit, I don't know if the results are a good representation of what was happening in the actual experiment.

I believe all these three points must be acknowledged and explained earlier in the article and not leave them for a small discussion at the end of the text. I would place them in section 2. This is a good study and will certainly be a reference for future studies, so these simplifications and decisions must be highlighted. Subsequent studies can identify these gaps and improve upon them. There is no problem by saying that simplifications have been done, actually that would be an advantage because they can be clearly identified.

These points have been highlighted in Section 2 and sentences on potential future improvements corresponding to the simplifications has been added in Section 4.5.

Finally, some responses are very useful but were only included in the line-by-line responses and not in the actual article. For example, the comment about convergence criteria, boundary conditions for k and epsilon, etc. Make sure that the answers are included in the text too. My comments are intended for the general audience.

These points were not presented in the first revision to avoid increasing the length of the manuscript which was already long. As suggested by the reviewer, some responses (including all the points mentioned by the reviewer) in first revision have been incorporated into the manuscript, especially in Section 2.2. For instance, the GMRES method has been added in Lines 197-198 and the setting of k and epsilon at the inlet boundary has been presented in Lines 215-217.

**References**

Lenzi, M.A., 2001. Step-pool evolution in the Rio Cordon, Northeastern Italy. *Earth Surface Processes and Landforms, 26*, 991–1008.

Turowski, J.M., Yager, E.M., Badoux, A., Rickenmann, D., Molnar, P., 2009. The impact of exceptional events on erosion, bedload transport and channel stability in a step-pool channel. *Earth Surface Processes and Landforms, 34*(12), 1661–1673.

Wohl, E. E., & Thompson, D. M. (2000). Velocity characteristics along a small step–pool channel. *Earth Surface Processes and Landforms*, *25*(4), 353-367.

Zhang, C., Xu, M., Hassan, M. A., Chartrand, S. M., Wang, Z., & Ma, Z. (2020). Experiment on morphological and hydraulic adjustments of step-pool unit to flow increase. *Earth Surface Processes and Landforms, 45*(2), 280-294.